# Discrete Markov Probabilistic Models: An Improved Discrete Score-Based Framework with sharp convergence bounds under minimal assumptions

Le Tuyet Nhi Pham [* 1]   Dario Shariatian [* 2]   Antonio Ocello [1]   Giovanni Conforti [3]   Alain Durmus [1]

## Abstract

This paper introduces the Discrete Markov Probabilistic Model (DMPM), a novel algorithm for discrete data generation. The algorithm operates in discrete space, where the noising process is a continuous-time Markov chain that can be sampled exactly via a Poissonian clock that flips labels uniformly at random. The time-reversal process, like the forward noise process, is a jump process, with its intensity governed by a discrete analogue of the classical score function. Crucially, this intensity is proven to be the conditional expectation of a function of the forward process, strengthening its theoretical alignment with score-based generative models while ensuring robustness and efficiency. We further establish convergence bounds for the algorithm under minimal assumptions and demonstrate its effectiveness through experiments on low-dimensional Bernoulli-distributed datasets and high-dimensional binary MNIST data. The results highlight its strong performance in generating discrete structures. This work bridges theoretical foundations and practical applications, advancing the development of effective and theoretically grounded discrete generative modeling.

## Introduction

Score-based Generative Models (SGMs) have become a key reference for generating complex data, such as images (see, e.g., Rombach et al., 2022; Ramesh et al., 2022; Saharia et al., 2022), audio (Chen et al., 2020; Kong et al., 2020), and video (Ho et al., 2022; Villegas et al., 2022; Bar-Tal et al., 2024). In continuous time, this approach benefits from a strong theoretical framework, and a scalable, stable learning objective.

By contrast, *discrete* generative modeling continues to pose significant challenges. Multiple diffusion-based methods have recently been proposed for discrete spaces (Austin et al., 2021; Hoogeboom et al., 2021; Shi et al., 2024; Campbell et al., 2022; Holderrieth et al., 2024; Ren et al., 2024), or spaces of mixed type (Bertazzi et al., 2024), but there is still no consensus on which approach is theoretically sound or most practically efficient. Various formulations rely on complex forward kernels or computationally unstable ratio-based estimators for backward transitions, leading to limited convergence guarantees and high computational costs in high dimensions. Furthermore, recent analyses of discrete diffusions have introduced valuable theoretical tools (Campbell et al., 2022; Holderrieth et al., 2024; Ren et al., 2024), yet most methods remain either overly generic or require strong assumptions, making them difficult to scale or to deploy with simple, stable training objectives.

**Contributions.** In this paper, we introduce *Discrete Markov Probabilistic Models* (DMPMs), a score-based generative models for discrete data inspired by Sohl-Dickstein et al. (2015) and Austin et al. (2021) that bridges these gaps. Our framework specializes the forward noising process to a continuous-time Markov chain on the hypercube $\{0, 1\}^d$. Leveraging theoretical insights on time-reversal Markov dynamics of this process, this choice preserves the key strengths and structure of continuous SGMs, addressing the issues raised in prior work. Our main results are summarized as follows:

- **Forward-Backward Construction.** We provide a principled derivation of the noising (forward) and denoising (backward) processes.

- **Score and denoiser function and stable estimation.** Our analysis reveals that the time-reversed process inherits a score function with an explicit conditional expectation form. By formulating the learning objective as an $L^2$ projection onto this score, we obtain a simple and principled regression loss term, as opposed to the high-variance estimators based on probability ratios

---

[*]Equal contribution [1]Centre de Mathématiques Appliquées (CMAP), École Polytechnique, Institut Polytechnique de Paris, 91120 Palaiseau, France [2]Sierra lab, Inria, Paris, France [3]Department of Mathematics, University of Padova, Italy. Correspondence to: Le Tuyet Nhi Pham <le-tuyet-nhi.pham@polytechnique.edu>.

*Proceedings of the 42$^{nd}$ International Conference on Machine Learning*, Vancouver, Canada. PMLR 267, 2025. Copyright 2025 by the author(s).

used in some prior work. Furthermore, we introduce a *denoiser reparameterization* of the score function, closely paralleling the continuous setting, which provides an interpretable and practical target for model training.

- **Theoretical Guarantees.** We prove that DMPMs converge to the underlying data distribution under minimal assumptions, providing non-asymptotic error bounds that underscore the method's reliability. In contrast to many earlier works, we prove that the sampling error grows linearly rather than exponentially with respect to dimension.

- **Empirical Performance.** We demonstrate that our approach attains competitive or superior performance on discrete datasets, including binarized MNIST, frequently with fewer function evaluations compared to existing discrete diffusion frameworks (e.g., 2.89 vs 7.34 FID compared to Discrete Flow Matching, Gat et al., 2024, with 2.5x fewer network calls).

**Notation.** Given a measurable space $(\mathsf{E}, \mathcal{E})$, we denote by $\mathcal{P}(\mathsf{E})$ the set of probability measures on $\mathsf{E}$. Given two probability measures $\mu, \nu \in \mathcal{P}(\mathsf{E})$, the Kullback–Leibler divergence (also called relative entropy) of $\mu$ with respect to $\nu$ is defined as $\mathrm{KL}(\mu|\nu) := \int \log \frac{d\mu}{d\nu} d\mu$ if $\mu$ is absolutely continuous with respect to $\nu$, and $\mathrm{KL}(\mu|\nu) = +\infty$ otherwise. The total variation distance between $\mu$ and $\nu$ is defined as $\|\mu - \nu\|_{\mathrm{TV}} = \sup_{\mathsf{A} \in \mathcal{E}} |\mu(\mathsf{A}) - \nu(\mathsf{A})|$. Consider a random variable $X$, we denote by $\mathrm{Law}(X)$ the law of $X$. We denote by $\delta_x$ the Dirac mass at $x$.

# 1. Forward and backward process of DMPMs

We propose a generative modeling framework that adapts diffusion-based methods to discrete spaces. Let $(\overrightarrow{X}_t)_{t \in [0,T]}$ be a forward Markov process on $\{0,1\}^d$, initialized from the data distribution $\mu^\star$, and evolving over a fixed time horizon $T_f > 0$ toward a simple base distribution. In the continuous setting, this role is often played by the Ornstein–Uhlenbeck process converging to a Gaussian. We define the corresponding backward process $(\overleftarrow{X}_t)_{t \in [0,T_f]}$ as $\overleftarrow{X}_t := \overrightarrow{X}_{T-t}$, which reconstructs $\mu^\star$ from the base distribution. While the backward process can be Markovian too, its transition rates or drift terms are typically intractable and must be approximated from forward trajectories, commonly done via score matching in continuous domains.

To adapt this idea, we introduce a forward CTMC on $\{0,1\}^d$ where each bit flips via an independent Poisson clock. We show that its time-reversal remains a tractable CTMC with backward rates given by conditional expectations over the forward process, enabling efficient regression-based training in the discrete setting.

## 1.1. CTMCs on discrete state-spaces

A CTMC $(X_t)_{t \in [0,T_f]}$ defined over the discrete space $\mathsf{X}$ is a Markovian stochastic process which is piecewise constant with jump at random times following a non-homogeneous Poisson process. As we shall see, under mild assumption, a (non-homogeneous) CTMC is uniquely characterized by a family of rate matrices $(q_t)_{t \in [0,T_f]}$, $q_t : \mathsf{X} \times \mathsf{X} \to \mathbb{R}$, which constitutes the infinitesimal generator of the process, and should satisfy $\sum_{y \in \mathsf{X}} q_t(x,y) = 0$, for any $x \in \mathsf{X}$. In particular, this object allows to define a Markov process whose transitions are informally characterized as $h \to 0$ by:

$$\mathbb{P}(X_{t+h} = y | X_t = x) = \delta_x(y) + hq_t(x,y) + o(h) , \quad (1)$$

where $o$ is the standard little-o Landau notation.

**Definition and sampling procedure.** When a simple characterization of the transition probability matrix is not available or does not exist, it is possible to simulate the process using the rate matrices, as suggested by equation (1). Popular sampling strategies include Gillespie's algorithm or $\tau$-leaping (Gillespie, 2007). In our case, we introduce the former which uses the jump rate and jump kernel associated to the process, defined as:

$$\lambda_t(x) = \sum_{y \in \mathsf{X}} q_t(x,y) , \quad k_t(x,y) = \mathbb{1}_{x \neq y} \frac{q_t(x,y)}{\lambda_t(x)} . \quad (2)$$

Informally, the jump rate governs the frequency of the random jumps of the process, and the jump kernel the next state at these jumps. More precisely, starting from a drawn $X_0$ from $\mu_0$ and a sequence of i.i.d. random variables distributed according to the exponential distribution with parameter 1, $\{E_i : i \in \mathbb{N}\}$, we can define the jump times $(T_i)_{i \in \mathbb{N}}$ of the process and its transition by induction setting $T_0 = 0$. Given $(T_i, X_{T_i})$, we define the next jump time as $T_{i+1} = T_i + \Delta T_{i+1}$, where

$$\Delta T_{i+1} = \inf\{t \geqslant 0 : \int_0^t \lambda_{T_i+r}(X_{T_i}) dr \geqslant E_i\} . \quad (3)$$

Then, set $X_t = X_{T_i}$ for $t \in (T_i, T_{i+1} \wedge T_f)$, and finally if $T_{i+1} < T_f$, $X_{T_{i+1}} = Y$ for $Y$ distributed according to $\mathrm{Cate}(\{k_{T_{i+1}}(X_{T_i}, y)\}_{y \in \mathsf{X}})$. Note that in the case where $\lambda_t = \lambda > 0$ for any $t \in [0, T_f]$, $(T_i)_{i \in \mathbb{N}}$ simply corresponds to the jump times a simple homogeneous Poisson process with rate $\lambda$, defined as $\mathfrak{N}_t = \sum_{i>1} \mathbb{1}_{T_i \leqslant t}$. Another equivalent procedure for simulating the process is provided in the supplement (see Appendix C.1). In practice, since the integral in (3) cannot be computed exactly, a time-discretized sampling strategy must be employed. We present these methodological considerations in Section 1.5 below.

**Kolmogorov equation.** As a final remark, the time-marginal distributions of the process $(X_t)_{t \in [0,T_f]}$ starting

from $X_0$, $\nu_t^{(X_0)}(x) = \mathbb{P}(X_t = x)$ and identified as a tuple, satisfy the backward Kolmogorov equation:

$$\partial_t \nu_t^{(X_0)}(x) = [\nu_t^{(X_0)}]^\mathrm{T} q_t(x) = \sum_{y \in \mathsf{X}} \nu_t^{(X_0)}(y) q_t(y, x) \ . \tag{4}$$

Here, we identify $q_t$ as a matrix, similarly to $\nu_t^{(X_0)}$. When the rate matrix is simple (e.g., as we use in our forward process in Section 1.2), we can use the Kolmogorov equation to derive the transition probability matrix $p_t(x_0, x_t) = \nu_t^{(x_0)}(x_t) = \mathbb{P}(X_t = x_t | X_0 = x_0)$, for $x_0, x_t \in \mathsf{X}, 0 \leqslant t \leqslant T_f$, yielding a *simulation-free* procedure.

### 1.2. Simple case $\mathsf{X} = \{0, 1\}$

To introduce the key ideas, we first consider the simple case where the state space is $\mathsf{X} = \{0, 1\}$, *i.e.*, when $d = 1$. We will then extend our method to $d > 1$ by factorizing the forward process over dimensions.

**Forward process.** We define the forward process $(\overrightarrow{X}_t)_{t \in [0, T_f]}$ as the homogeneous CTMC starting from $\overrightarrow{X}_0 \sim \mu^\star$, and driven by a simple bit-flip process associated with the rate matrix defined for any $x, y \in \mathsf{X}$, as

$$\overrightarrow{q}^1(x, y) := \begin{cases} \lambda \ , & \text{if } y \neq x \ , \\ -\lambda \ , & \text{otherwise} \ . \end{cases} \tag{5}$$

In the case of a constant forward rate matrix $\overrightarrow{q}^1$, the transition probability matrix $\overrightarrow{p}_t^1$, for $0 \leqslant t \leqslant T_f$, is known to be $\overrightarrow{p}_t^1 = \exp(t \overrightarrow{q}^1)$ (Liggett, 2010), which we compute as:

$$\overrightarrow{p}_t^1(x, y) = \begin{cases} \frac{1}{2} + \frac{1}{2} \mathrm{e}^{-2\lambda t} \ , & \text{if } x = y \ , \\ \frac{1}{2} - \frac{1}{2} \mathrm{e}^{-2\lambda t} \ , & \text{otherwise} \ . \end{cases} \tag{6}$$

The detailed derivations are given in the supplementary material B.1 and rely on equation (4). We note that using a time-dependent rate $\lambda_t$ is a straightforward extension, as defining $\overrightarrow{q}_t^1 = \lambda_t \overrightarrow{q}^1$ yields the transition matrix $\exp(\overrightarrow{q}^1 \int_0^t \lambda_s \mathrm{d}s) = \overrightarrow{p}_{\int_0^t \lambda_s \mathrm{d}s}^1$, and leave it to future work.

**Backward process.** To recover the data distribution, we analyze the time-reversed process, which is denoted by $(\overleftarrow{X}_t)_{t \in [0, T_f]}$, and defined as $\overleftarrow{X}_t = \overrightarrow{X}_{T_f - t}$ for any $t \in [0, T_f]$. Conforti & Léonard (2022, Theorem 2.8) show that $(\overleftarrow{X}_t)_{t \in [0, T_f]}$ is also a non-homogeneous CTMC, associated with a family of generator matrices $(\overleftarrow{q}_t)_{t \in [0, T_f]}$ satisfying the time-reversal formula:

$$\mu_{T_f - t}(x) \overleftarrow{q}_t^1(x, y) = \mu_{T_f - t}(y) \overrightarrow{q}^1(y, x) \ , \tag{7}$$

for any $0 \leqslant t \leqslant T_f$ and $x \neq y \in \mathsf{X}$, where for any $t \in$

$[0, T_f]$, we denote by $\mu_t$ the forward marginal distribution:

$$\mu_t(x) = \mathbb{P}(\overrightarrow{X}_t = x) \ . \tag{8}$$

Since our chosen rate matrix $\overrightarrow{q}$ is symmetric (see (5)) and $\mu_{T_f - t}(x) > 0$ for all $x \in \mathsf{X}$, $t \in [0, T_f)$, we deduce that the backward generator $\overleftarrow{q}_t^1$ for $0 \leqslant t < T_f$ is given for any $x \neq y \in \mathsf{X}$ by

$$\overleftarrow{q}_t^1(x, y) = \overrightarrow{q}^1(y, x) \frac{\mu_{T_f - t}(y)}{\mu_{T_f - t}(x)} \ . \tag{9}$$

**Discrete score function.** Define $s_t : \mathsf{X} \to \mathbb{R}$ for any $x \in \mathsf{X}$ by

$$s_t(x) := \frac{\mu_{T_f - t}(x) - \mu_{T_f - t}(1 - x)}{\mu_{T_f - t}(x)} \ . \tag{10}$$

$s_t$ acts as a discrete derivative in $\mathsf{X}$ of $\log \mu_t$, and thus serves as a discrete analogue of the score function in continuous models. With this notation, $\overleftarrow{q}_t(x, y)$ (9) can be expressed, for any $x \neq y \in \mathsf{X}$, for $0 \leqslant t < T_f$, as:

$$\overleftarrow{q}_t^1(x, y) := \begin{cases} \lambda(1 - s_t(x)) \ , & \text{if } y \neq x \ , \\ -\lambda(1 - s_t(x)) \ , & \text{otherwise} \ . \end{cases} \tag{11}$$

Access to the sequence $(s_t)_{t \in [0, T_f]}$ is equivalent to having access to $(\overleftarrow{q}_t^1)_{t \in [0, T_f]}$, and therefore allows to sample from $\overleftarrow{X}_t$ for any $t \in [0, T_f]$ using the procedure described in Section 1.1. However, the score function $s$ is generally intractable, as it depends on the unknown marginal distributions $(\mu_t)_{t \in [0, T_f]}$. To address this, we derive an alternative expression for $s_t$ in terms of a conditional expectation over the forward process. For $x \in \mathsf{X}$ and $t \in [0, T_f)$,

$$s_t(x) =$$
$$\mathbb{E}\left[\frac{2\alpha_{T_f - t}}{1 + \alpha_{T_f - t}} - \frac{4\alpha_{T_f - t} \mathbb{1}_{\overrightarrow{X}_0}(\overrightarrow{X}_{T_f - t})}{1 - \alpha_{T_f - t}^2} \middle| \overrightarrow{X}_{T_f - t} = x \right], \tag{12}$$

with

$$\alpha_t := \mathrm{e}^{-2\lambda t} \ . \tag{13}$$

Indeed, the score function can be computed as

$$s_t(x) = 1 - \frac{\mu_{T_f - t}(y)}{\mu_{T_f - t}(x)} = 1 - \sum_{x_0 \in \mathsf{X}} \frac{p_{T_f - t|0}(y|x_0)}{\mu_{T_f - t}(x)} \mu^\star(x_0)$$

$$= 1 - \sum_{x_0 \in \mathsf{X}} \frac{p_{T_f - t|0}(y|x_0)}{p_{T_f - t|0}(x|x_0)} p_{0|T_f - t}(x_0|x)$$

$$= \mathbb{E}\left[1 - \frac{p_{T_f - t|0}(y|\overrightarrow{X}_0)}{p_{T_f - t|0}(x|\overrightarrow{X}_0)} \middle| \overrightarrow{X}_{T_f - t} = x \right] \ ,$$

and plugging in the expression for our forward transition

matrix given in (6) we obtain equation (12). Therefore, the function $s$ is an $L^2$-projection and its approximation boils down to a regression problem.

### 1.3. General state space $\mathsf{X} = \{0, 1\}^d$

**Forward process.** We generalize the previous results for the hypercube in $\mathbb{R}^d$, *i.e.*, the state space is $\mathsf{X} = \{0, 1\}^d$ with $d \in \mathbb{N}^*$. We consider the homogeneous CTMC $(\overrightarrow{X}_t)_{t \in [0, T_f]}$ starting from $\overrightarrow{X}_0 \sim \mu^\star$, defined with the following rate matrix:

$$\overrightarrow{q}(x, y) := \begin{cases} \lambda\,, & \text{if } \|y - x\|^2 = 1\,, \\ -\lambda d\,, & \text{if } y = x\,, \\ 0\,, & \text{otherwise}\,. \end{cases} \tag{14}$$

The corresponding sampling procedure following the one provided in Section 1.1 is given in Appendix B.2 for completeness. Similarly to the one-dimensional case, we can establish an explicit expression for the transition probability matrix $\overrightarrow{p}_t$ for $0 \leqslant t \leqslant T_f$ as

$$\overrightarrow{p}_t(x, y) = \prod_{i=1}^d \overrightarrow{p}_t^1(x^i, y^i)\,, \tag{15}$$

where $\overrightarrow{p}_t^1$ is defined in (6) and $x = (x^i)_{i=1}^d, y = (y^i)_{i=1}^d \in \mathsf{X}$. The detailed computation is given in the supplementary material B.2.1 The factorization of the transition probability in (15) is of great practical interest, as this tells us that the dynamic of the forward process simply consists in the single-bit forward dynamic applied independently to each component, as described in Section 1.2. As a consequence, the forward marginal distribution $\mu_t$ of $\overrightarrow{X}_t$ admits the formula

$$\mu_t(x) = \sum_{z \in \mathsf{X}} \mu_0(z) \prod_{i=1}^d \overrightarrow{p}_t(z, x)\,. \tag{16}$$

**Backward process and score function.** Denote by $(\overleftarrow{X}_t)_{t \in [0, T_f]}$, the time-reversal process associated with $(\overrightarrow{X}_t)_{t \in [0, T_f]}$, and defined as $\overleftarrow{X}_t = \overrightarrow{X}_{T_f - t}$ for any $t \in [0, T_f]$. As in the case $d = 1$, Conforti & Léonard (2022, Theorem 2.8) shows that $(\overleftarrow{X}_t)_{t \in [0, T_f]}$ is also a *non-homogeneous* CTMC, with backward generator matrix $(\overleftarrow{q}_t)_{t \in [0, T_f]}$ that satisfies (7) and therefore (9), proceeding as before. As in the case $d = 1$, we show that $(\overleftarrow{q}_t)_{t \in [0, T_f]}$ depends only on a discrete score function, which we now introduce.

First, note that (9) and (14) yield $\overleftarrow{q}_t(x, y) = 0$, for $x, y \in \mathsf{X}$ satisfying $\|x - y\|^2 \neq 1$ and $x \neq y$. Then, for $0 \leqslant t < T_f$, define $s_t : \mathsf{X} \to \mathbb{R}^d$ for any $x \in \mathsf{X}$, $s_t(x) = \{s_t^\ell(x)\}_{\ell=1}^d$ as

the vector in $\mathbb{R}^d$, with components $\ell \in \{1, \ldots, d\}$,

$$s_t^\ell(x) := \frac{\mu_{T_f - t}(x) - \mu_{T_f - t}(\varphi^{(\ell)}(x))}{\mu_{T_f - t}(x)}\,, \tag{17}$$

where $\varphi^{(\ell)} : \mathsf{X} \to \mathsf{X}$ is defined as $\varphi^{(\ell)}(x) = y$, with $y$ obtained by flipping the $\ell$-th bit of $x$, *i.e.*, $y^\ell = 1 - x^\ell$, and $y^i = x^i$ for $i \neq \ell$. Then, for $0 \leqslant t < T_f$, $x \neq y \in \mathsf{X}$, we can write the backward generator $\overleftarrow{q}_t(x, y)$, as given in (7), as:

$$\overleftarrow{q}_t(x, y) = \sum_{\ell=1}^d \lambda(1 - s_t^\ell(x)) \mathbb{1}_{y = \varphi^{(\ell)}(x)}\,.$$

**Score function.** Note that the function $s$ thus defined is an extension to the case $d \geqslant 1$ of the function $s$ defined for $d = 1$ in (10). As a result, $s_t$ is a conditional expectation over the forward process, where each of its components admits an expression similar to the 1d case (12).

**Proposition 1.1.** *The score function can be expressed as a conditional expectation:*

$$s_t^\ell(x) = \mathbb{E}\left[f_t^\ell(\overrightarrow{X}_0, \overrightarrow{X}_{T_f - t}) \mid \overrightarrow{X}_{T_f - t} = x\right]\,, \tag{18}$$

*where $t \in [0, T_f)$, $x \in \mathsf{X}$, $\ell = 1, \ldots, d$, $s_t^\ell$ is the $\ell$-th component of the score function $s_t$, and*

$$f_t^\ell(\overrightarrow{X}_0, \overrightarrow{X}_{T_f - t}) = \frac{2\alpha_{T_f - t}}{1 + \alpha_{T_f - t}} - \frac{4\alpha_{T_f - t}(\overrightarrow{X}_{T_f - t}^\ell - \overrightarrow{X}_0^\ell)^2}{1 - \alpha_{T_f - t}^2}\,. \tag{19}$$

The proof of this result is given in Appendix B.2.2. Similarly to the 1d case, access to the score allows to simulate the backward process following the procedure described in Section 1.1 since the the non-homogeneous jump rate $\overleftarrow{\lambda}_t$ and jump kernel $\overleftarrow{k}_t$ of the backward process are given by

$$\overleftarrow{\lambda}_t(x) = \lambda \sum_{\ell=1}^d (1 - s_t^\ell(x))\,,$$

$$\overleftarrow{k}_t(x, y) = \sum_{\ell=1}^d \mathbb{1}_{y = \varphi^{(\ell)}(x)} \cdot \lambda(1 - s_t^\ell(x))/\overleftarrow{\lambda}_t(x)\,, \tag{20}$$

for $x \neq y \in \mathsf{X}$ and $t \in [0, T_f)$.

### 1.4. Approximating the score function

In this section, we derive the training objective to estimate the score function $(s_t)_{t \in [0, T_f)}$ defined in (17), which governs the backward rate matrix. As in standard diffusion models, our goal is to sample from the time-reversed process, requiring approximation of the (intractable) score. Leveraging its conditional expectation structure (Proposi-

tion 1.1), we approximate $s_t$ using a parameterized family $\left\{(t, x) \mapsto s_t^\theta(x)\right\}_{\theta \in \Theta}$, where $\theta$ is optimized via an adapted score-matching objective, defined as the function

$$\mathfrak{L}_{\mathrm{L}^2} : \theta \mapsto \int_0^{T_f} \mathbb{E}\left[\|s_{T_f-t}^\theta(\overrightarrow{X}_t) - f_{T_f-t}(\overrightarrow{X}_0, \overrightarrow{X}_t)\|^2\right] \mathrm{d}t \,. \tag{21}$$

Another objective function to fit $\theta$ can be derived by using the fact that for any $x \in \mathsf{X}$, $t \in [0, T_f), \ell \in \{1, \ldots, d\}$, $1 - s_t^\ell(x)$ is non-negative. Thus, we introduce the following entropy-based term:

$$\theta \mapsto \int_0^{T_f} \mathbb{E}\left[\sum_{\ell=1}^d (1 - s_{T_f-t}^{\theta,\ell}) h\left(\frac{1 - s_{T_f-t}^\ell}{1 - s_{T_f-t}^{\theta,\ell}}\right) (\overrightarrow{X}_t)\right] \mathrm{d}t \,,$$

where $h(a) = a \log(a) - (a-1)$. Minimizing this function is equivalent to minimizing:

$$\mathfrak{L}_{\mathrm{e}} : \theta \mapsto \int_0^{T_f} \mathbb{E}\left[\sum_{\ell=1}^d \left( - s_{T_f-t}^{\theta,\ell}(\overrightarrow{X}_t) \right.\right.$$
$$\left.\left. + (s_{T_f-t}^\ell(\overrightarrow{X}_t) - 1) \log(1 - s_{T_f-t}^{\theta,\ell}(\overrightarrow{X}_t)) \right) \right] \mathrm{d}t \,. \tag{22}$$

We further derive a discrete denoiser structure in Appendix C.3, rewriting the score function as

$$s_t^\ell(x) = \frac{2\alpha_{T_f-t}}{1 + \alpha_{T_f-t}} - \frac{4\alpha_{T_f-t} d_t^\ell(x)}{1 - \alpha_{T_f-t}^2} \,, \tag{23}$$

where $d_t^\ell(x) = \mathbb{P}(\overrightarrow{X}_0^\ell \neq x^\ell | \overrightarrow{X}_{T_f-t} = x)$ serves as a classifier referred to as a *discrete denoiser*. We leverage this structure by reparameterizing our score model as:

$$s_t^{\theta,\ell}(x) = \frac{2\alpha_{T_f-t}}{1 + \alpha_{T_f-t}} - \frac{4\alpha_{T_f-t} d_t^{\theta,\ell}(x)}{1 - \alpha_{T_f-t}^2} \,. \tag{24}$$

As a result, we modify our objective $\mathfrak{L}_{\mathrm{L}^2}$ to $\mathfrak{L}_{\mathrm{L}^2}^{\mathrm{den}}$ to fit the conditional expectations $(d_t)_{t \in [0,T_f)}$ instead of the score functions $(s_t)_{t \in [0,T_f)}$, as follows:

$$\mathfrak{L}_{\mathrm{L}^2}^{\mathrm{den}} : \theta \mapsto \int_0^{T_f} \mathbb{E}\left[\|\sum_{\ell=1}^d d_{T_f-t}^{\theta,\ell}(\overrightarrow{X}_t) - \mathbb{1}_{\overrightarrow{X}_0^\ell}(\overrightarrow{X}_t^\ell)\|^2\right] \mathrm{d}t \,, \tag{25}$$

see Appendix C.4 for more details. This reparameterization moves the approximation from a space of ratios into probability space, which is smoother and more amenable to learning, mitigating the instability of direct score or rate estimation as reported in (Lou et al., 2024). To fit $d_t^\theta(x)$ to

$d_t(x)$, we introduce an additional cross-entropy loss $\mathfrak{L}_{\mathrm{CE}}$:

$$\mathfrak{L}_{\mathrm{CE}} : \theta \mapsto \int_0^{T_f} \mathbb{E}\left[\sum_{\ell=1}^d -\mathbb{1}_{\overrightarrow{X}_0^\ell \neq \overrightarrow{X}_{T_f-t}^\ell} \log d_t^{\theta,\ell}(\overrightarrow{X}_{T_f-t}^\ell)\right.$$
$$\left. + (\mathbb{1}_{\overrightarrow{X}_0^\ell \neq \overrightarrow{X}_{T_f-t}^\ell} - 1) \log(1 - d_t^{\theta,\ell}(\overrightarrow{X}_{T_f-t}^\ell))\right] \mathrm{d}t \,.$$

Based on the previous discussions, we consider a linear combination of the losses $\mathfrak{L}_{\mathrm{L}^2}^{\mathrm{den}}, \mathfrak{L}_{\mathrm{e}}, \mathfrak{L}_{\mathrm{CE}}$, respectively weighted by factors $\varpi_1, \varpi_2, \varpi_3$, which results in the loss $\mathfrak{L}_\varpi$:

$$\mathfrak{L}_\varpi = \varpi_1 \mathfrak{L}_{\mathrm{L}^2}^{\mathrm{den}} + \varpi_2 \mathfrak{L}_{\mathrm{e}} + \varpi_3 \mathfrak{L}_{\mathrm{CE}} \,. \tag{26}$$

The expected value of $d_t^\ell$ is given by

$$w_t = \mathbb{E}\left[d_t^\ell(\overrightarrow{X}_{T_f-t})\right] = (1 - \alpha_{T_f-t})/2 \,, \tag{27}$$

as detailed in Appendix C.4. Thus, we scale losses $\mathfrak{L}_{\mathrm{L}^2}, \mathfrak{L}_{\mathrm{CE}}$ by $1/w_t$, ensuring a more balanced average magnitude across timesteps; see (50) and Figure 4 in Appendix C.4. This leads to the updated loss $\mathfrak{L}_\varpi^w$ (see (52)). Detailed derivations are provided in Appendix C.4. The final training procedure is outlined in Algorithm 2.

### 1.5. Generative process

Alike classical continuous diffusion models, exact simulations of the reverse process are not possible and face the same challenges: i) we do not have access to i.i.d. samples from $\mu_{T_f}$, ii) the backward process characteristics depend on the score function of the forward process defined in (17), which is intractable, and iii) we have to discretize the continuous process.

**Initialize the backward from the uniform distribution.** We show that $(\overrightarrow{X}_t)_{t \in [0,T_f]}$ converges geometrically to $\gamma^d$, the uniform distribution over $\mathsf{X}$ (see Appendix B.2.3 in the supplementary document). This should be put in parallel with diffusion-based models, where the stochastic process at hand, e.g., Ornstein–Uhlenbeck, converges geometrically fast to some Gaussian distribution. The generative model can then be initialized to $\gamma^d$ rather than $\mu_{T_f}$.

**Score approximation.** We have access to a score approximation $(s_t^{\theta^\star})_{t \in [0,T_f]}$, so the generative model can then be sampled analogously to the backward process, replacing $(s_t)_{t \in [0,T_f]}$ with $(s_t^{\theta^\star})_{t \in [0,T_f]}$, leading to the non-homogeneous jump rate and kernel approximating (20):

$$\lambda_t^{\theta^\star}(x) = \lambda \sum_{\ell=1}^d (1 - s_t^{\theta^\star,\ell}(x)) \,,$$

$$k_t^{\theta^\star}(x, y) = \sum_{\ell=1}^d \mathbb{1}_{y = \varphi^{(\ell)}(x)} \cdot \lambda(1 - s_t^{\theta^\star,\ell}(x))/\lambda_t^{\theta^\star}(x) \,,$$

for $x, y \in \mathsf{X}$ and $t \in [0, T_f)$, where we denote by $s_t^{\theta^\star, \ell}$ the $\ell$-th component of $s_t^{\theta^\star}$. For completeness, Algorithm 1 in Appendix C.2 provides the pseudo-code for an ideal, continuous-time approximation of the backward process.

**Time discretization.** Exact integration of jump rates is infeasible in practice, so we discretize time and approximate the backward rate and kernel using piecewise constant functions. Let $\{t_k\}_{k=0}^K$ be a time grid with step sizes $\tau_k = t_k - t_{k-1}$. Given $\overleftarrow{X}_{t_k}^\star$, for $x \neq y \in \mathsf{X}$ and $t \in [t_k, t_{k+1})$, we set:

$$\overleftarrow{q}_t^{\theta^\star}(x, y) = \lambda \sum_{\ell=1}^d (1 - s_{t_k}^{\theta^\star, \ell}(\overleftarrow{X}_{t_k}^\star)) \mathbb{1}_{y = \varphi^{(\ell)}(x)}$$

$$= (1 - \hat{s}_t^{\theta^\star}(x, y)) \overrightarrow{q}(x, y) ,$$

where $\hat{s}_t^{\theta^\star}(x, y) = \sum_{\ell=1}^d s_{t_k}^{\theta^\star, \ell}(\overleftarrow{X}_{t_k}^\star) \mathbb{1}_{y = \varphi^{(\ell)}(x)}$.

This yields a tractable CTMC $(\overleftarrow{X}_t^\star)_{t \in [0, T_f]}$, which can be simulated starting from $\gamma^d$. Under mild conditions, its final law converges to the target distribution. The associated DMPM sampler is given in Algorithm 3, Appendix C.5, with time-schedule choices listed in Table 2.

**Flips sampler.** The standard sampler updates one bit at each timestep. To improve parallelism and sample diversity, we propose a flip-schedule where $M_{t_k}$ components are flipped simultaneously at step $t_k$, based on the probability distribution defined by $\hat{k}^\theta$. We consider linear and cosine flip-schedules (Table 3), implemented in Algorithm 4, Appendix C.5.

**Denoise-renoise sampler.** We also introduce a denoise-renoise sampler based on the discrete denoiser from Equation (23). Inspired by multistep consistency models (Song et al., 2023), this method alternates denoising from $t_0 = 0$ to $T_f$ and re-noising back to $t_1$, and so on. The full procedure is detailed in Algorithm 5 and Appendix C.5.

## 2. Convergence of DMPMs algorithm

This section provides quantitative error estimates between the generated final distribution $\mathrm{Law}(\overleftarrow{X}_{T_f}^\star)$ and our data distribution $\mu^\star$ via the Kullback–Leibler divergence KL. To this end, we consider the following assumptions on the parameterized score and the original data distribution:

**Assumption 2.1.** Let $h(a) := a \log(a) - (a - 1)$ for $a > 0$. There exists $\epsilon \in (0, 1)$ such that

$$\max_{0 \leqslant k \leqslant K} \mathbb{E}\left[ \sum_{\ell=1}^d (1 - s_{t_k}^{\theta^\star, \ell}) h\left( \frac{1 - s_{t_k}^\ell}{1 - s_{t_k}^{\theta^\star, \ell}} \right) (\overrightarrow{X}_{T_f - t_k}) \right] \leqslant \epsilon .$$

$$(28)$$

Note that Assumption 2.1 is induced by the entropic term

$\mathfrak{L}_e$ defined in (22) of the loss function we consider in practice. This condition naturally appears as we bound the KL divergence of the path probability measures corresponding to the approximate score $s^{\theta^\star}$ and the ideal one $s$ respectively. Indeed, we prove a Girsanov type theorem which provide an explicit expression of the density between these two measures in Theorem F.12 in the supplement F.2.1. While standard Girsanov theorem for diffusion implies an $L^2$-type approximation error condition for generative models (see, e.g., Conforti et al., 2025; Lee et al., 2023; Chen et al., 2022a), our result naturally involve the entropic-type condition (28) due to the discrete structure of our noising process.

**Assumption 2.2.** The data distribution does not admit any zero-value, i.e., $\mu^\star(x) \in (0, 1)$ for any $x \in \mathsf{X}$.

Assumption 2.2 implies that the data distribution has the finite Fisher-like information

$$\beta_{\gamma^d}(\mu^\star) := \mathbb{E}\left[ \sum_{\ell=1}^d h\left( e^{g(\overrightarrow{X}_0) - g(\varphi^{(\ell)}(\overrightarrow{X}_0))} \right) \right] < +\infty ,$$

$$(29)$$

with $g := -\log(\mathrm{d}\mu^\star / \mathrm{d}\gamma^d)$. Note that Assumption 2.2 is put in parallel with the finite relative Fisher information condition provided by Conforti et al. (2025). However, Assumption 2.2 is much simpler as the state space is finite, and the function $h$ is only infinite if $\mu^\star$ has not full support.

We are now ready to state the error's bound of the generated data using DMPMs given in Algorithm 3, noting that every measure on the hypercube possesses finite entropy.

**Theorem 2.3.** *Under Assumption 2.1 and Assumption 2.2, the following bound holds*

$$\mathrm{KL}(\mu^\star | \mathrm{Law}(\overleftarrow{X}_{T_f}^\star)) \leqslant e^{-4\lambda T_f} \mathrm{KL}(\mu^\star | \gamma^d)$$
$$+ \lambda \tau \beta_{\gamma^d}(\mu^\star) + \lambda \epsilon T_f ,$$

$$(30)$$

*with $\tau := \max\{\tau_k, k = 1, \ldots, K\}$.*

Theorem 2.3 is one of our distinguishing results, which guarantees the convergence of DMPMs algorithm, and makes it stronger than other algorithms built before for discrete target distribution.

The term $\epsilon T_f$ in (30) appears because the score function $s_t$ is replaced in the discretization by its approximation $s_t^{\theta^\star}$ satisfying Assumption 2.1. The term $e^{-4\lambda T_f} \mathrm{KL}(\mu^\star | \gamma^d)$ represents the initialization error, as our backward dynamic starts at $\gamma^d$ instead of $\mu_{T_f}$. Finally, the term $\tau \beta_{\gamma^d}(\mu^\star)$ means that the data distribution $\mu^\star$ cannot be peculiar, in the sense that $\mu^\star$ does not admit any zero-value. The detailed proof of Theorem 2.3 is given in the supplementary material F.5.1.

Following Conforti et al. (2025, Theorem 3), a tighter

bound on the sampling error, one that scales logarithmically rather than linearly with the discrete Fisher information, is achieved with an appropriate sequence of step sizes.

**Theorem 2.4.** *Let $c \in (0, 1/2)$ and $T_f \geqslant 1 + 2c$. Suppose Assumption 2.1 and Assumption 2.2 hold, and $L = d^{-1}\beta_{\gamma^d}(\mu^\star) \geqslant 2$. Choose an exponentially decreasing step-sizes, i.e., $\tau_{k+1} = c\min\{\max\{T_f - t_k, a\}, 1\}$ for $k < K$, with $a = 1/L$, then we have that*

$$\mathrm{KL}(\mu^\star | \mathrm{Law}(\overleftarrow{X}^\star_{T_f})) \lesssim \mathrm{e}^{-4\lambda T_f}\mathrm{KL}(\mu^\star | \gamma^d) + \lambda\epsilon T_f \\ + \lambda c d [1 + \log(L)]. \quad (31)$$

The proof of Theorem 2.4 benefits from the choice of the step-size's scheme and is postponed to Appendix F.5.2. We deduce the following complexity result for DMPMs to achieve an $\epsilon > 0$ discretization error.

**Corollary 2.5.** *Consider the sequence of step-size as in Theorem 2.4 and suppose Assumption 2.1 and Assumption 2.2 hold. Choosing*

$$c = \frac{\epsilon}{\lambda d[1 + \log(L)]} \quad and \quad T_f = \frac{1}{4\lambda}\log\frac{\mathrm{KL}(\mu^\star | \gamma^d)}{\epsilon}, \quad (32)$$

*we get*

$$K \lesssim d[1 + \log(L)]\left[\log(\mathrm{KL}(\mu^\star | \gamma^d)/\epsilon) + \lambda\log(L)\right]/\epsilon,$$

*and makes the approximation error $\tilde{O}(\epsilon\log(\mathrm{KL}(\mu^\star | \gamma^d)))$, where the notation $\tilde{O}$ means that logarithmic factors of $d, \epsilon$ have been dropped.*

The proof of Corollary 2.5 is provided in Appendix F.5.3.

In our next result, we get rid of Assumption 2.2 using an early stopping strategy.

**Theorem 2.6.** *Under Assumption 2.1, for any $\eta \in (0, T_f)$, let $c \in (0, 1/2]$ and $T_f - \eta \geqslant 1 + 2c$. Set $L = d^{-1}\beta_{\gamma^d}(\mu_\eta)$ and assume that $L \geqslant 2$. Choose the constant and exponentially decreasing sequence of step-size, i.e., satisfying $\tau_{k+1} = c\min\{\max\{T_f - \eta - t_k, 1/L\}, 1\}$ for $k < K$ and the associated discrete time scheme $\{t_k\}_{k=0}^K$ such that $t_0 = 0$ and $t_K = T_f - \eta$. Then, the following bound holds*

$$\mathrm{KL}(\mu_\eta | \mathrm{Law}(\overleftarrow{X}^\star_{T_f - \eta})) \lesssim d\eta^{-1}\mathrm{e}^{-4\lambda(T_f - \eta)} + \lambda\epsilon(T_f - \eta) \\ + \lambda c d[1 + \log(\eta^{-1})]. \quad (33)$$

Theorem 2.6 is a consequence of Theorem 2.4 when the backward dynamic stops early at $\mu_\eta$ instead of $\mu^\star$. We benefit from the structure of $\mu_\eta$ to obtain a bound growing linearly with dimension, which is advantageous for high-dimensional sampling. The full proof is deferred to Appendix F.6.1. It is worth noting that (16) ensures that $\mu_\eta$ is always positive for any $\eta \in (0, T_f)$, thus the Fisher-like

information $\beta_{\gamma^d}(\mu_\eta)$ is always finite. As a result, Assumption 2.2 is no longer required. To obtain then a complexity bound for DMPMs on its discretization error without Assumption 2.2, we bound in our next result, the total variation distance between $\mu^\star$ and $\mu_\eta$ for $\eta > 0$.

**Proposition 2.7.** *For any $\eta \in (0, \max\{T_f, \frac{1}{\lambda}\})$, the following holds*

$$\|\mu_\eta - \mu^\star\|_{\mathrm{TV}} \leqslant 2 - 2(1 - \lambda\eta)^d. \quad (34)$$

The proof of Proposition 2.7 is provided in the supplement F.6.2. Combining Theorem 2.6 and Proposition 2.7, we deduce that

**Corollary 2.8.** *Consider the sequence of step-size as in Theorem 2.6 and let Assumption 2.1 hold. Choosing*

$$\eta = \frac{1 - (1 - \epsilon)^{1/d}}{\lambda}, \quad c = \frac{\epsilon^2}{\lambda d[1 + \log(\eta^{-1})]}, \\ T_f = \eta + \frac{1}{4\lambda}\log\frac{d}{\eta\epsilon^2}, \quad (35)$$

*implies that*

$$K \lesssim d[1 + \log(\lambda d/\epsilon)][\log(d/\epsilon^2) + (\lambda + 1)\log(\lambda d/\epsilon)]/\epsilon^2,$$

*and the following bound holds*

$$\|\mu^\star - \mathrm{Law}(\overleftarrow{X}^\star_{T_f - \eta})\|_{\mathrm{TV}} \lesssim \epsilon + \sqrt{\lambda\epsilon(T_f - \eta)}.$$

The proof of Corollary 2.8 is given in Appendix F.6.3.

# 3. Existing works on diffusion-based generative models for discrete data

We briefly review existing approaches to discrete generative modeling based on diffusion processes. Additional discussion is provided in Appendix A.

A first class of methods maps discrete variables into continuous spaces, enabling the use of classical diffusion machinery (Dieleman et al., 2022; Chen et al., 2022b; Richemond et al., 2022), but struggle to scale in dimensions and lacks theoretical guarantees.

Other methods, such as Argmax Flows and Multinomial Diffusion (Hoogeboom et al., 2021), operate directly in discrete spaces and use categorical noise models or argmax transformations to handle discrete tokens, but can impose considerable computational overhead.

More recently, CTMC-based frameworks were introduced (Campbell et al., 2022), upon which flow matching techniques were adapted to the discrete domains (Gat et al., 2024; Campbell et al., 2024), using conditionally constructed rate matrices built ad-hoc, contrasting with our

principled time-reversal derivation.

Holderrieth et al. (2024) proposed a general framework for generator matching over arbitrary Markov processes, assuming access to a conditional interpolating distribution, leaving the choice of interpolating process and loss function to problem-specific adaptation.

Alternative directions include masked diffusion models (Austin et al., 2021) and stochastic integral formulations (Ren et al., 2024), aiming to balance tractability and theoretical soundness. Shi et al. (2024); Sahoo et al. (2024) propose efficient training via absorbing-state kernels, optimizing model parameterization and loss. Meanwhile, Lou et al. (2024) model the score function as a density ratio rather than a discrete denoiser. This yields an entropic regularization loss equivalent to (22), missing the $L^2$ projection and cross-entropy terms applied to the discrete denoiser employed in our formulation.

Concurrently with our work, Bach & Saremi (2025) propose a discrete analogue of Gaussian smoothing, entirely forgoing the continuous-time framework. Their denoising-based method offers a static-noise alternative to CTMCs, yielding Langevin-type sampling dynamics on the hypercube.

Theoretical results for discrete generative models are much scarcer than for their continuous counterparts. To the best of our knowledge, only Campbell et al. (2022); Ren et al. (2024) provide theoretical guarantees, and these rely on significantly stronger assumptions than those used in our work.

Regarding Theorem 1 in Campbell et al. (2022), we note that our approach does not require an $L^\infty$-bound on the score approximation error, but only an $L^2$-bound. Moreover, we do not impose any assumptions on the marginal density of the forward process (cf. Assumption 2), but only on the data distribution itself. We also avoid placing assumptions on the backward transition rates (cf. Assumption 3). In contrast to Campbell et al. (2022), we impose no conditions on the rates or densities of the forward and backward processes. Additionally, their discretization error scales linearly with the time horizon, whereas our bounds do not incur such a cost.

Concerning Ren et al. (2024), and in particular Theorems 4.7 and 4.9, we highlight that we make no assumptions on the score function. In their Assumption 4.4, a time-dependent $L^\infty$-bound of the form $|s_t(x)| \lesssim 1 \vee t^{-1}$ is imposed on the true score $s_t$, and a time-uniform $L^\infty$-bound is assumed for the learned score $s_t^\theta$. As in Campbell et al. (2022), these assumptions are arguably unnatural, as they are not placed directly on the data distribution but on more complex transformations of it. Furthermore, Assumption 4.5 in Ren et al. (2024) imposes a quantitative form of Lipschitz continuity on $s_t$, and as the authors themselves state, "As-

sumption 4.5 corresponds to the Lipschitz continuity of the score function." However, it is now well known that such an assumption is unnecessary in the continuous setting.

## 4. Experiments

The full experimental details are available in Appendix D. We evaluate our Discrete Markov Probabilistic Model (DMPM) on two datasets. The first is a low-dimensional synthetic *sawtooth* dataset, with dimension $4 \leqslant d \leqslant 16$. The second is binarized MNIST, with $d = 32 \times 32$. We explore various design choices, and compare DMPM against MD4 (masked diffusion) (Shi et al., 2024) and DFM (discrete flow matching) (Gat et al., 2024), two state-of-the-art discrete generative approaches.

### 4.1. Experiments on Small-Dimensional Bernoulli Data

We study a discrete data distribution $p$ such that each component of $X = (X_i)_{i=1}^d \sim p$ is independently distributed as Bernoulli$(p_i)$. The map $i \mapsto p_i$ forms a sawtooth pattern (see Figure 6). We evaluate performance using a custom Sliced Wasserstein Distance (SWD) between the learned and true distributions (see Appendix D.3). Indeed, the state space size $2^d$ can get too big for traditional histogram-based metrics like KL divergence or Hellinger distance.

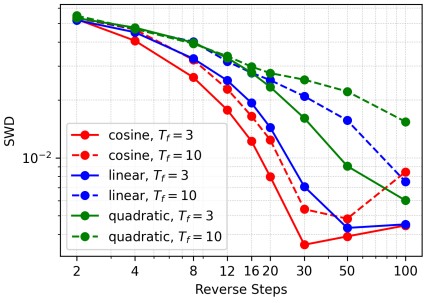

*Figure 1.* Comparison of time-schedules (*cosine*, *linear*, *quadratic*) and time horizon ($T_f = 3$ vs. $T_f = 10$).

**Time horizon and time-schedule.** We study the impact of the time horizon $T_f$ and various time-schedules—uniform, quadratic, and cosine (see Table 2)—on model performance. Figure 1 presents results for a model trained with the basic $\mathcal{L}_{L^2}$ loss on data with $d = 16$, evaluated across multiple reverse step counts. The cosine schedule with $T_f = 3$ yields the best performance in terms of sliced Wasserstein distance (SWD), outperforming other schedules and longer horizons, while also requiring fewer reverse steps. This indicates that $T_f = 3$ is sufficient to reach near-uniformity during forward diffusion, without excessive transitions to uniform states. We adopt this configuration for all subsequent experiments.

**Comparison with state-of-the-art methods.** We compare

| $d$ | 4 | 8 | 12 | 16 |
|---|---|---|---|---|
| **DFM** | 6.102 | 8.864 | 5.019 | 8.302 |
| **MD4** | 9.376 | 7.670 | 4.045 | 8.037 |
| **DMPM** | **3.174** | **3.308** | **2.342** | **2.515** |

*Figure 2.* SWD ↓, in 1e-3, for DMPM, MD4, and DFM across data dimension $d$. Selected the best result with #steps $2 \leqslant K \leqslant 200$ for each method.

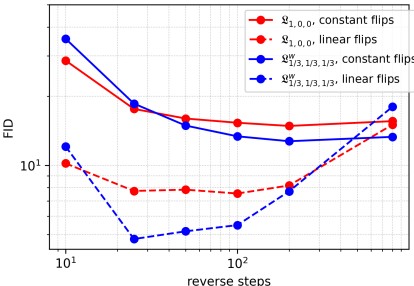

*Figure 3.* FID↓ on MNIST, linear vs. constant flip-schedules scaled for $d$ total bit flips, with various loss configurations.

DMPM (with cosine schedule, $T_f = 3$, and $\mathfrak{L}_{L^2}$ loss) against MD4 and Discrete Flow Matching (DFM). Figure 2 shows SWD scores across varying data dimensions $d$. DMPM consistently outperforms both baselines, achieving superior results with significantly fewer reverse steps (typically 30 vs 100), highlighting its sampling efficiency.

## 4.2. Experiments on Higher-Dimensional Binary MNIST

**DMPM sampler and flip-schedule.** We evaluate the DMPM sampler using constant and linear flip-schedules $\{M_{t_k}\}_{k=1}^{K}$. Empirically, performance is optimal when the total number of flipped bits matches the input dimension, i.e., $\sum_{k=1}^{K} M_{t_k} = d$. We scale each flip-schedule accordingly. Figure 3 illustrates that performance remains stable across different values of $K$ as long as the total number of flips is held constant, allowing for significant speedups by reducing the number of reverse steps and thus network calls.

Using a model trained with the weighted loss $\mathfrak{L}_{1/3,1/3,1/3}^{w}$ and a linear flip-schedule, we achieve a best FID of $4.77$ using only 25 network calls. The linear schedule consistently outperforms the constant variant; flipping fewer bits early helps guide the model toward more coherent samples, similarly as what is reported for MD4 (Shi et al., 2024).

**Loss function configuration.** Among the losses we tested, the balanced form $\mathfrak{L}_{1,,1,,1}^{w}$ consistently yields the best results. The weighting factor $w$ helps normalize the scale of the $\ell_2$, cross-entropy, and KL components at each timestep, improving overall synergy. Figure 5 illustrates these effects. Nonetheless, simpler variants such as $\mathfrak{L}_{L^2}$ and $\mathfrak{L}_{L^2}^{w}$ already achieve near-optimal performance in many settings.

| Method | | 10 | 25 | 50 | 100 | 200 | 500 |
|---|---|---|---|---|---|---|---|
| **DFM** | FID | 227.55 | 156.26 | 88.93 | 39.62 | 16.26 | 7.34 |
| | $F_1^{dc}$ | 0.00 | 0.00 | 0.01 | 0.14 | 0.41 | 0.68 |
| **MD4** | FID | 97.97 | 33.50 | 14.06 | 6.83 | 4.48 | *3.43* |
| | $F_1^{dc}$ | 0.04 | 0.29 | 0.57 | 0.76 | 0.83 | 0.86 |
| **DMPM**$_{flips}$ | FID | 16.30 | 9.98 | 11.07 | 9.07 | 7.80 | 10.84 |
| | $F_1^{dc}$ | 0.64 | 0.92 | 0.93 | 0.93 | 0.93 | 0.70 |
| **DMPM**$_{denoise}$ | FID | 78.20 | 20.94 | 8.62 | 3.98 | **2.89** | 4.36 |
| | $F_1^{dc}$ | 0.13 | 0.67 | 0.87 | 0.96 | *1.00* | **1.00** |

*Table 1.* FID↓ (first row of each method) and $F_1^{dc}$ ↑ (second row) on MNIST for various total reverse steps. We highlight the best result in **bold**, the 2$^{nd}$ best in *italics*, and underline the 3$^{rd}$ best.

**Denoise-renoise sampler and comparison with state-of-the-art.** We further exploit the discrete denoiser structure through a denoise-renoise sampler (Algorithm 5, Appendix C.4), which alternates single-step denoising and re-noising in a multistep loop. This approach leverages the model's learned transitions more effectively, leading to notable gains in sample quality.

We compare DMPM, trained with the balanced loss $\mathfrak{L}_{1,1,1}^{w}$, under two sampling strategies—denoise-renoise and linear flip-schedule—against state-of-the-art baselines: MD4 (masked diffusion) and DFM (discrete flow matching). Results are reported for varying numbers of reverse steps $K$, using both Fréchet Inception Distance (FID) and the $F_1^{dc}$ metric, a harmonic mean of coverage and density (Naeem et al., 2020). While FID captures overall realism, $F_1^{dc}$ reflects local distributional fidelity; full details are in Appendix D.3.

As shown in Table 1, both DMPM variants (rows 3 and 4) consistently outperform the baselines. At $K = 200$, DMPM (denoise-renoise) achieves the best FID (2.89) and perfect $F_1^{dc}$ (1.00), surpassing MD4 (4.48) and DFM (16.26). Even with $K = 50$, it maintains strong results (FID 8.62, $F_1^{dc}$ 0.87). Similarly, DMPM with flip-schedule achieves FID below 10 and $F_1^{dc}$ above 0.90 at $K = 25$, demonstrating excellent efficiency with minimal network calls.

Sample grids illustrating visual quality are shown in Figure 7 and Figure 8.

### 4.3. Conclusions

Our experiments show that DMPM consistently matches or surpasses state-of-the-art performance on both low- and high-dimensional discrete datasets. On binarized MNIST, it achieves better FID and $F_1^{dc}$ than competing methods, with fewer network calls. These gains stem from our principled reparameterization of the score function as a denoiser, and a stable, well-structured training objective. Together, they yield a scalable and theoretically sound framework for discrete generative modeling.

## Impact Statement

This paper presents work whose goal is to advance the field of Machine Learning. There are many potential societal consequences of our work, none which we feel must be specifically highlighted here.

## Acknowledgements

The work of L.T.N.P. was supported by the École Doctorale de Mathématiques Hadamard (EDMH). The work of A.O. was funded by the European Union (ERC-2022-SYG-OCEAN-101071601). The work of D.S was funded by the European Union (ERC, Dynasty, 101039676). Views and opinions expressed are however those of the authors only and do not necessarily reflect those of the European Union or the European Research Council Executive Agency. Neither the European Union nor the granting authority can be held responsible for them.

This work was granted access to the HPC resources of IDRIS under the allocation 2025-AD011015323R1 made by GENCI.

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

# A. Existing works on diffusion-based generative models for discrete data

This section provides details of the recent researches on discrete generative models.

**Embedding discrete structure in the continuous space.** To keep the benefits of continuous representations, Dieleman et al. (2022) and Chen et al. (2022b) mapped discrete structures into Euclidean space, while Richemond et al. (2022) placed them into the simplex, all while continuing to use forward continuous diffusion models. In particular, Dieleman et al. (2022) proposed a continuous diffusion model for categorical data, which has some advantages over autoregressive models, such as the ability to perform arbitrary infilling and a more flexible sampling process. However, this method comes with an expensive training cost and lacks of strong theoretical guarantees.

**Argmax flows and Multinomial Diffusion.** Hoogeboom et al. (2021) introduced two new generative models, Argmax Flows and Multinomial Diffusion, to handle categorical data like text and image segmentation. Argmax Flows connect discrete data with continuous models by using an argmax function combined with a probabilistic inverse, making categorical distributions easy-learning. Multinomial Diffusion process uses a categorical distribution to add noise to discrete data and then trains a model to reverse the process. However, both Argmax Flows and Multinomial Diffusion have some limitations: computational costs increase due to additional steps , and the theoretical guarantee is missing.

**Designing the flow processes over the discrete state space.** Campbell et al. (2022) introduced the first complete continuous-time framework for denoising diffusion models applied to discrete data. They used CTMCs to model the forward noising process and its time-reversal dynamics. While the core idea is similar to ours, their approach is more complex because their method consider generic CTMC and is not specialized to the noising process that we consider. As a result, their method essentially boils down learning density ratios which can be computationally demanding and fail to offer efficient approximation in high dimensions. They also added a correction step to bring the sample distribution closer to the desired one, which increased the practical training cost Gat et al. (2024). By focusing on the random-walk CTMC on X, we were able to provide a discrete counterpart to the score function that is learn in continuous diffusion models and also to establish strong convergence guarantees for our method. Campbell et al. (2024) further extended this line of work by adapting flow-matching techniques to discrete domains using conditionally defined rate matrices. However, their method does not derive the reverse process from a time-reversal principle, and thus relies on hand-crafted dynamics with limited theoretical justification.

**Generator Matching.** Another recent approach to handle discrete data is generative modeling with arbitrary Markov processes using generator matching, introduced by Holderrieth et al. (2024). In this approach, the authors design an appropriate Markov process that transforms a simple distribution into the desired data one using a generator, which can be efficiently trained with a neural network. This method is quite flexible and can be applied to different state spaces, especially in discrete settings. However, this method being very generic suffer from the same drawback as Campbell et al. (2022).

**Masked diffusion models.** One important step toward more advanced models is the "masked" diffusion process, a discrete diffusion approach first introduced by Austin et al. (2021). Recently, Shi et al. (2024) looked into this model further, simplifying its training objective by expressing it as a signal-to-noise ratio, which helps highlight some useful features. However, despite these improvements, the model still lacks theoretical guarantees. Sahoo et al. (2024) improved upon this direction by leveraging the structure of the absorbing kernel and refining the bridge-based reverse process, leading to more efficient optimization. The model's reliance on absorbing-state approximations and heuristic training objectives limits its theoretical grounding.

**Direct score parameterization.** Lou et al. (2024) propose to learn the score function directly as a density ratio rather than a denoising map. This formulation leads to a single entropic regularization loss equivalent to (22) in our paper. However, it misses our discrete denoiser decomposition we use and the associated $L^2$ projection and cross-entropy terms, which improve training stability.

**Discrete Diffusion Models via a Stochastic Integral Framework.** Ren et al. (2024) introduced a new way to analyze discrete diffusion models via Lévy-type stochastic integrals and expanded Poisson random measures. Specifically, they established the stochastic integral expressions of the noising and denoising processes for the categorical data. They provided a unified error analysis framework and showed the first error bound for their algorithms in KL divergence. However, their results rely on strong assumptions in contrast to our results. Besides, our bounds are simpler and better, in particular with respect to the time horizon.

**Denoising without time dynamics.** Concurrently with our work, Bach & Saremi (2025) propose a discrete denoising model

that avoids continuous-time formulations altogether. Their method treats Bernoulli corruption as an analogue of Gaussian smoothing, enabling a Langevin-style sampler on the hypercube. This approach is complementary to CTMC-based methods and their dynamic interpretability.

Our paper takes a step toward bridging these gaps. By clearly describing the forward Markov process, we can express the score function as a conditional expectation, which helps us avoid the costly signal-to-noise ratio training used in Shi et al. (2024). This way, we not only offer a simpler and more affordable training approach, but also provide solid theoretical guarantees for our models in practice.

## B. Interpretation of DMPMs

### B.1. The simple case $\mathsf{X} = \{0, 1\}$

We start by explicitly constructing our forward process $(\overrightarrow{X}_t)_{t\in[0,T_f]}$ starting from $\overrightarrow{X}_0 \sim \mu^\star$. Consider the fixed jump times $(T_i)_{i\in\{1,\dots,N\}}|N \overset{\text{iid}}{\sim} \text{Unif}([0,T_f])$ of a Poisson process over $[0,T_f]$ where $N \sim \text{Pn}(\lambda T_f)$ is the number of jumps, and $\lambda > 0$ is a prescribed jump rate. Without loss of generality, we assume that $0 = T_0 \leqslant T_1 < \dots < T_N$. We define recursively $(\overrightarrow{X}_t)_{t\in[0,T_f]}$ over $(T_i, T_{i+1})$. Suppose that $\overrightarrow{X}_{T_i}$ has been defined we set $\overrightarrow{X}_t = \overrightarrow{X}_{T_i}$ for any $t \in (T_i, T_{i+1})$ and $\overrightarrow{X}_{T_{i+1}} = 1 - \overrightarrow{X}_{T_i}$. It is well known that $(\overrightarrow{X}_t)_{t\in[0,T_f]}$ is a Markov jump process (Owen, 2021, Section 6) with generator $\overrightarrow{q}^1$ defined for any $x, y \in \mathsf{X}$ as

$$\overrightarrow{q}^1(x,y) := \begin{cases} \lambda, & \text{if } y \neq x, \\ -\lambda, & \text{otherwise}. \end{cases} \tag{36}$$

The transition probability matrix $\mathbb{P}(\overrightarrow{X}_t = y|\overrightarrow{X}_0 = x) = \overrightarrow{p}^1_t(x,y)$, for $x, y \in \mathsf{X}, 0 \leqslant t \leqslant T_f$, is known to be

$$\overrightarrow{p}^1_t(x,y) = \begin{cases} \frac{1}{2} + \frac{1}{2}\mathrm{e}^{-2\lambda t}, & \text{if } x = y, \\ \frac{1}{2} - \frac{1}{2}\mathrm{e}^{-2\lambda t}, & \text{otherwise}. \end{cases} \tag{37}$$

*Detailed calculation of the transition probability in* (6). Based on the Kolmogorov equation, the transition matrix $\overrightarrow{p}^1_t$ for $0 \leqslant t \leqslant T_f$ admits the following formula

$$\overrightarrow{p}^1_t = \mathrm{e}^{t\overrightarrow{q}^1},$$

where $\overrightarrow{q}^1$ is define in (5). Clearly, the generator $\overrightarrow{q}^1$ admits two eigenvalues $0$ and $-2\lambda$ associated with the eigenvectors $\begin{pmatrix} 1 & 1 \end{pmatrix}^\mathrm{T}$ and $\begin{pmatrix} 1 & -1 \end{pmatrix}^\mathrm{T}$ respectively. Thus we can diagonalize $\overrightarrow{q}^1$ as

$$\overrightarrow{q}^1 = \begin{pmatrix} 1 & 1 \\ 1 & -1 \end{pmatrix} \begin{pmatrix} 0 & 0 \\ 0 & -2\lambda \end{pmatrix} \begin{pmatrix} 1 & 1 \\ 1 & -1 \end{pmatrix}^{-1},$$

and the transition matrix $\overrightarrow{p}^1_t$ follows

$$\overrightarrow{p}^1_t = \mathrm{e}^{t\overrightarrow{q}^1} = \begin{pmatrix} 1 & 1 \\ 1 & -1 \end{pmatrix} \begin{pmatrix} 1 & 0 \\ 0 & \mathrm{e}^{-2\lambda t} \end{pmatrix} \begin{pmatrix} 1 & 1 \\ 1 & -1 \end{pmatrix}^{-1} = \frac{1}{2} \begin{pmatrix} 1 + \mathrm{e}^{-2\lambda t} & 1 - \mathrm{e}^{-2\lambda t} \\ 1 - \mathrm{e}^{-2\lambda t} & 1 + \mathrm{e}^{-2\lambda t} \end{pmatrix}.$$

$\square$

### B.2. General state space $\mathsf{X} = \{0, 1\}^d$

#### B.2.1. FORWARD PROCESS

We consider the jump times $(T_i)_{i\in\{1,\dots,N\}}|N \overset{\text{iid}}{\sim} \text{Unif}([0,T_f])$ of a Poisson process over $[0,T_f]$ where $N \sim \text{Pn}(\lambda T_f)$ is the number of jump. Without loss of generality, we suppose that $T_0 = 0 \leqslant T_1 < \dots < T_N$. We define recursively $(\overrightarrow{X}_t)_{t\in[0,T_f]}$ over $(T_i, T_{i+1})$ as follows. Suppose $\overrightarrow{X}_{T_i}$ has been defined. We set $\overrightarrow{X}_t = \overrightarrow{X}_{T_i}$ for $t \in (T_i, T_{i+1})$, and finally, set $\overrightarrow{X}^{\ell_i}_{T_{i+1}} = 1 - \overrightarrow{X}^{\ell_i}_{T_i}$, where $\ell_i \sim \text{Unif}(\{1,\dots,d\})$, with $\ell_i$ independent from the past, and $\overrightarrow{X}^j_{T_{i+1}} = \overrightarrow{X}^j_{T_i}$ for $j \neq \ell_i$. The associated generator matrix is given in (14). We now seek to obtain the associated transition matrix.

***Proof of*** (15). We start with a note that the generator matrix $\overrightarrow{q}$ can be expressed as a sum of matrices $\overrightarrow{q}^{\ell}$ as follows

$$\overrightarrow{q} = \sum_{\ell=1}^{d} \overrightarrow{q}^{\ell}, \quad \text{with } \overrightarrow{q}^{\ell}(x,y) = \begin{cases} \lambda, & \text{if } x^i = y^i \text{ for } i \neq \ell \text{ and } x^{\ell} \neq y^{\ell}, \\ -\lambda, & \text{if } x = y, \\ 0, & \text{otherwise}. \end{cases}$$

Notice that $\overrightarrow{q}^{\ell}$ also admits the following formula with respect concerning the tensor product

$$\overrightarrow{q}^{\ell} = \underbrace{\mathbb{I} \otimes \mathbb{I} \otimes ... \otimes \mathbb{I} \otimes \overrightarrow{A} \otimes \mathbb{I} \otimes ... \otimes \mathbb{I}}_{d \text{ times}}, \tag{38}$$

with $\mathbb{I}$ the $2 \times 2$ identity matrix and $\overrightarrow{A} = \begin{pmatrix} -\lambda & \lambda \\ \lambda & -\lambda \end{pmatrix}$, which is the $\ell^{th}$ matrix in the previous product. Indeed, by the definition of tensor product, for any $x = (x^i)_{i=1}^{d}, y = (y^i)_{i=1}^{d} \in \mathsf{X}$, we observe that

$$(\mathbb{I} \otimes \mathbb{I} \otimes ... \otimes \mathbb{I} \otimes \underbrace{\overrightarrow{A}}_{\ell^{th}} \otimes \mathbb{I} \otimes ... \otimes \mathbb{I})(x,y) = \mathbb{I}(x^1, y^1)\mathbb{I}(x^2, y^2)...\overrightarrow{A}(x^{\ell}, y^{\ell})...\mathbb{I}(x^d, y^d)$$

$$= \begin{cases} \lambda, & \text{if } x^i = y^i \text{ for } i \neq \ell \text{ and } x^{\ell} \neq y^{\ell}, \\ -\lambda, & \text{if } x = y, \\ 0, & \text{otherwise}. \end{cases}$$

which is exactly the expression of $\overrightarrow{q}^{\ell}(x,y)$. We now use the Kolmogorov equation combined with the expression of $\overrightarrow{q}_t^{\ell}$ in (38), and apply the formula $e^{\mathbb{I} \otimes A + B \otimes \mathbb{I}} = e^A \otimes e^B$ for any matrix $A, B$ (Gavrilyuk et al., 2011, Appendix) to get

$$\overrightarrow{p}_t = e^{t\overrightarrow{q}} = e^{\sum_{\ell=1}^{d} t\overrightarrow{q}^{\ell}} = \underbrace{e^{t\overrightarrow{A}} \otimes ... \otimes e^{t\overrightarrow{A}}}_{d \text{ times}}.$$

We are thus left with the computation of $e^{t\overrightarrow{A}}$. It is clear that the eigenvalues of $\overrightarrow{A}$ are $0$ and $-2\lambda$, with the corresponding eigenvectors $\begin{pmatrix} 1 & 1 \end{pmatrix}^T$ and $\begin{pmatrix} 1 & -1 \end{pmatrix}^T$ respectively. Consequently, we can compute $e^{t\overrightarrow{A}}$ as: for any $a, b \in \{0, 1\}$,

$$\overrightarrow{p}_t^1(a,b) := e^{t\overrightarrow{A}}(a,b) = \begin{cases} \frac{1}{2} + \frac{1}{2}e^{-2\lambda t}, & \text{if } a = b, \\ \frac{1}{2} - \frac{1}{2}e^{-2\lambda t}, & \text{if } a \neq b, \end{cases}$$

and the formula of transition probability $\overrightarrow{p}_t$ for $0 \leqslant t \leqslant T_f$ follows: for any $x = (x^i)_{i=1}^{d}$ and $y = (y^i)_{i=1}^{d}$ in $\mathsf{X}$,

$$\overrightarrow{p}_t(x,y) = \prod_{i=1}^{d} \overrightarrow{p}_t^1(x^i, y^i), \quad \text{with } \overrightarrow{p}_t^1(x^i, y^i) = \begin{cases} \frac{1}{2} + \frac{1}{2}e^{-2\lambda t}, & \text{if } x^i = y^i, \\ \frac{1}{2} - \frac{1}{2}e^{-2\lambda t}, & \text{otherwise}. \end{cases}$$

$\square$

### B.2.2. CONDITIONAL EXPECTATION EXPRESSION OF THE SCORE FUNCTION

***Proof of Proposition 1.1.*** Fix $x \in \mathsf{X}$ and $\ell = 1, \ldots, d$. First note that by definition of $\mu_{T_f - t}$ as the marginal distribution of the noising process, we have

$$\mu_{T_f - t}(x) - \mu_{T_f - t}(\varphi^{(\ell)}(x)) = \sum_{z \in \mathsf{X}} \mu_0(z)(\overrightarrow{p}_{T_f - t}(z, x) - \overrightarrow{p}_{T_f - t}(z, \varphi^{(\ell)}(x))). \tag{39}$$

The formula of transition probabilities $\overrightarrow{p}_{T_f-t}(z, \varphi^{(\ell)}(x))$ combined with the definition of $\varphi^{(\ell)}(x)$ lead to

$$
\begin{aligned}
\overrightarrow{p}_{T_f-t}(z, \varphi^{(\ell)}(x)) &= \prod_{i=1}^{d} \overrightarrow{p}^{1}_{T_f-t}(z^i, \varphi^i_\ell(x)) \\
&= \overrightarrow{p}^{1}_{T_f-t}(z^\ell, \varphi^\ell_\ell(x)) \prod_{\substack{i=1 \\ i\neq\ell}}^{d} \overrightarrow{p}^{1}_{T_f-t}(z^i, x^i) \\
&= \frac{\overrightarrow{p}^{1}_{T_f-t}(z^\ell, \varphi^\ell_\ell(x))}{\overrightarrow{p}^{1}_{T_f-t}(z^\ell, x^\ell)} \overrightarrow{p}_{T_f-t}(z, x) .
\end{aligned}
$$

Substituting this into (39) implies

$$
\begin{aligned}
\mu_{T_f-t}(x) - \mu_{T_f-t}(\varphi^{(\ell)}(x)) &= \sum_{z\in\mathsf{X}} \mu_0(z) \overrightarrow{p}^{1}_{T_f-t}(z, x)(1 - \frac{\overrightarrow{p}^{1}_{T_f-t}(z^\ell, \varphi^{(\ell),\ell}(x))}{\overrightarrow{p}^{1}_{T_f-t}(z^\ell, x^\ell)}) \\
&= \sum_{z\in\mathsf{X}} \left[ \frac{2\mathrm{e}^{-2\lambda(T_f-t)}}{1+\mathrm{e}^{-2\lambda(T_f-t)}} - \frac{4\mathrm{e}^{-2\lambda(T_f-t)}(x^\ell - z^\ell)^2}{1-\mathrm{e}^{-4\lambda(T_f-t)}} \right] \mathbb{P}\left[ \overrightarrow{X}_0 = z, \overrightarrow{X}_{T_f-t} = x \right] ,
\end{aligned}
$$

where the last equality comes from the formula of $\overrightarrow{p}^{1}_{T_f-t}$ and the fact that if $z^\ell = \varphi^{(\ell),\ell}(x)$ then $z^\ell \neq x^\ell$. Therefore, the score function in components are

$$
\begin{aligned}
s^\ell_t(x) &= \frac{\mu_{T_f-t}(x) - \mu_{T_f-t}(\varphi^{(\ell)}(x))}{\mu_{T_f-t}(x)} \\
&= \sum_{z\in\mathsf{X}} \left[ \frac{2\mathrm{e}^{-2\lambda(T_f-t)}}{1+\mathrm{e}^{-2\lambda(T_f-t)}} - \frac{4\mathrm{e}^{-2\lambda(T_f-t)}(x^\ell - z^\ell)^2}{1-\mathrm{e}^{-4\lambda(T_f-t)}} \right] \mathbb{P}\left[ \overrightarrow{X}_0 = z | \overrightarrow{X}_{T_f-t} = x \right] \\
&= \mathbb{E}\left[ \frac{2\alpha_{T_f-t}}{1+\alpha_{T_f-t}} - \frac{4\alpha_{T_f-t}(\overrightarrow{X}^\ell_{T_f-t} - \overrightarrow{X}^\ell_0)^2}{1-\alpha^2_{T_f-t}} \Bigg| \overrightarrow{X}_{T_f-t} = x \right] , \quad \text{for } t \in [0, T_f) ,
\end{aligned}
$$

where $\alpha_t = \mathrm{e}^{-2\lambda t}$, and we finish the proof of Proposition 1.1. $\qquad\square$

### B.2.3. INVARIANT MEASURE OF THE FORWARD PROCESS

As we have a comprehensive understanding of the forward process, we observe that its invariant measure is the uniform distribution over $\mathsf{X}$, denoted by $\gamma^d$. Indeed, for any $x \in \mathsf{X}$ and $t \in [0, T_f]$,

$$
(\gamma^d \overrightarrow{p}_t)(x) = \sum_{z\in\mathsf{X}} \gamma^d(z) \overrightarrow{p}_t(z, x) = \frac{1}{2^d} \sum_{z\in\mathsf{X}} \overrightarrow{p}_t(z, x) = \frac{1}{2^d} = \gamma^d(x) .
$$

Furthermore, by formula of $\overrightarrow{p}$ given in (15), we have $\overrightarrow{p}_t(x, y) \xrightarrow{t\to\infty} \frac{1}{2^d}$ for any $x, y \in \mathsf{X}$. Consequently, the following holds for any $x \in \mathsf{X}$,

$$
\mu_t(x) = \sum_{z\in\mathsf{X}} \mu_0(z) \overrightarrow{p}_t(z, x) \xrightarrow{t\to\infty} \frac{1}{2^d} \sum_{z\in\mathsf{X}} \mu_0(z) = \frac{1}{2^d} = \gamma^d(x) ,
$$

meaning that the forward dynamic $(\overrightarrow{X}_t)_{t\in[0,T_f]}$ converges geometrically fast to $\gamma^d$.

## C. Implementation of DMPMs

### C.1. Alternative ideal backward simulation

Besides the simulation of the backward process provided in Section 1.3, we can also use the following procedure to produce the time-reversal dynamic.

The second procedure to sample $(\overleftarrow{X}_t)_{t\in[0,T_f]}$ is to consider a sample $\overleftarrow{X}_0$ from $\mu_{T_f}$ and a sequence of i.i.d. random variables distributed according to the exponential distribution with parameter 1, $\{E_i^\ell : i \in \mathbb{N}, \ell \in \{1,\ldots,d\}\}$, we can define the jump times $(T_i)_{i\in\mathbb{N}}$ of the backward process and its transition by induction setting $T_0 = 0$. Given $(T_i, \overleftarrow{X}_{T_i})$, we define the next jump time as $T_{i+1}^j = T_i + \Delta T_{i+1}^j$, where $\Delta T_{i+1}^j = \inf\{t \geqslant 0 : \int_0^t \lambda(1 - s^j(\overleftarrow{X}_{T_i}))\mathrm{d}r \geqslant E_i^j\}$. Then, set $T_{i+1} = T_{i+1}^{\ell_i}$, where $\ell_i = \arg\min_{j\in\{1,\ldots,d\}} T_{i+1}^j$, and $\overleftarrow{X}_t = \overleftarrow{X}_{T_i}$ for $t \in (T_i, T_{i+1} \wedge T_f)$, and finally if $T_{i+1} < T_f$, $\overleftarrow{X}_{T_{i+1}}^{\ell_i} = 1 - \overleftarrow{X}_{T_i}^{\ell_i}$ for $\ell_i \in \{1,\ldots,d\}$.

## C.2. Perfect backward approximation

We provide here the pseudo-code of backward approximation sampling in continuous time scheme:

---

**Algorithm 1** DMPMs Algorithm (Continuous time scheme)

**Input:** a time horizon $T_f \gg 1$ large enough, a prescribed jump rate $\lambda$, an approximate score function $s^{\theta^\star}$

    **Backward process:**
    Set $T_0 = 0$ and initialize $\overleftarrow{X}_0 \sim \gamma^d$
    $i \leftarrow 0$
    **while** $T_i \leqslant T_f$ **do**
      Draw $E_i \sim \mathrm{Exp}(1)$
      Solve $\Delta T_{i+1} = \inf\{t \geqslant 0 : \int_0^t \lambda_{T_i+r}^{\theta^\star}(\overleftarrow{X}_{T_i})\mathrm{d}r \geqslant E_i\}$, with $\lambda_t^{\theta^\star}(x) = \lambda\sum_{\ell=1}^d(1 - s_t^{\theta^\star,\ell}(x))$
      Set $T_{i+1} = T_i + \Delta T_{i+1}$
      **if** $T_i < t < \min(T_{i+1}, T_f)$ **then**
        Set $\overleftarrow{X}_t = \overleftarrow{X}_{T_i}$
      **end if**
      **if** $T_{i+1} < T_f$ **then**
        Draw $\ell_i \in \{1,\ldots,d\} \sim \mathrm{Cate}(\{\lambda(1 - s_{T_{i+1}}^{\theta^\star,\ell}(\overleftarrow{X}_{T_i}))/\lambda_{T_{i+1}}^{\theta^\star}(\overleftarrow{X}_{T_i})\}_{\ell=1}^d)$
        Set $\overleftarrow{X}_{T_{i+1}} = \varphi^{(\ell_i)}(\overleftarrow{X}_{T_i})$
      **end if**
      $i \leftarrow i + 1$
    **end while**
**Output:** $\overleftarrow{X}_{T_f}$

---

## C.3. Discrete denoiser and score reparameterization

**Discrete-denoiser structure.**

Recall from Proposition 1.1 that each score component admit the following conditional expectation:

$$s_t^\ell(x) = \mathbb{E}\left[f_t^\ell(\overrightarrow{X}_0, \overrightarrow{X}_{T_f-t})|\overrightarrow{X}_{T_f-t} = x\right] , \tag{40}$$

where

$$f_t^\ell(\overrightarrow{X}_0^\ell, \overrightarrow{X}_{T_f-t}) = \frac{2\alpha_{T_f-t}}{1 + \alpha_{T_f-t}} - \frac{4\alpha_{T_f-t}(\overrightarrow{X}_{T_f-t}^\ell - \overrightarrow{X}_0^\ell)^2}{1 - \alpha_{T_f-t}^2} \tag{41}$$

for $t \in [0, T_f)$, $x \in \mathsf{X}$ and $\ell = 1,\ldots,d$.

Remark that

$$\mathbb{E}\left[f_t^\ell(\overrightarrow{X}_0^\ell, \overrightarrow{X}_{T_f-t})|\overrightarrow{X}_{T_f-t} = x\right] = \frac{2\alpha_{T_f-t}}{1 + \alpha_{T_f-t}} - \frac{4\alpha_{T_f-t}\mathbb{E}\left[(\overrightarrow{X}_{T_f-t}^\ell - \overrightarrow{X}_0^\ell)^2|\overrightarrow{X}_{T_f-t} = x\right]}{1 - \alpha_{T_f-t}^2} . \tag{42}$$

Thus we introduce the function $d_t^\ell$ defined as

$$d_t^\ell : x \mapsto \mathbb{E}\left[(\overrightarrow{X}_{T_f-t}^\ell - \overrightarrow{X}_0^\ell)^2 | \overrightarrow{X}_{T_f-t} = x\right] , \tag{43}$$

which can be further rewritten as

$$\begin{aligned}
d_t^\ell(x) &= \mathbb{E}\left[\left(\overrightarrow{X}_{T_f-t}^\ell - \overrightarrow{X}_0^\ell\right)^2 \bigg| \overrightarrow{X}_{T_f-t} = x\right] \\
&= \mathbb{E}\left[\mathbb{1}_{\overrightarrow{X}_{T_f-t}^\ell \neq \overrightarrow{X}_0^\ell} \bigg| \overrightarrow{X}_{T_f-t} = x\right] \\
&= \mathbb{P}\left(\overrightarrow{X}_0^\ell \neq x^\ell \bigg| \overrightarrow{X}_{T_f-t} = x\right) .
\end{aligned}$$

In some sense, this is the discrete version of the continuous denoiser $\mathbb{E}[\overrightarrow{X}_0 | \overrightarrow{X}_t]$ approximated by classical diffusion models (Song et al., 2021), as obtained from the score by Tweedie's formula. Thus we call $d_t^\ell(x)$ the discrete denoiser.

**Score reparameterization.** Based on the previous derivations, each score component $s_t^\ell(x)$ can be written as a function of $d_t^\ell$:

$$s_t^\ell(x) = \frac{2\alpha_{T_f-t}}{1 + \alpha_{T_f-t}} - \frac{4\alpha_{T_f-t} d_t^\ell(x)}{1 - \alpha_{T_f-t}^2} , \tag{44}$$

So we can reparameterize our score models $s_t^\theta$ as

$$s_t^{\theta,\ell}(x) = \frac{2\alpha_{T_f-t}}{1 + \alpha_{T_f-t}} - \frac{4\alpha_{T_f-t} d_t^{\theta,\ell}(x)}{1 - \alpha_{T_f-t}^2} , \tag{45}$$

where $d_t^{\theta,\ell}(x)$ aims to approximate $d_t^\ell(x)$.

### C.4. Objective functions derived from the discrete denoiser structure

Inspired by the previous derivations, we modify our existing $\mathfrak{L}_{L^2}$ loss function to replace by a denoising loss equivalent. We introduce a cross-entropy loss, and finally propose a scaling of the loss functions, based on the average output magnitude of the discrete denoiser, thus helping with the learning, and improving synergies between loss elements.

**Score-matching objective $\mathfrak{L}_{L^2}$.** We rewrite the objective function $\mathfrak{L}_{L^2}$ to fit the *discrete denoiser*, considered as a conditional expectation:

$$\mathfrak{L}_{L^2}^{\text{den}} : \theta \mapsto \int_0^{T_f} \mathfrak{L}_{t,L^2}(\theta) \mathrm{d}t , \quad \mathfrak{L}_{t,L^2}(\theta) = \mathbb{E}\left[\|d_{T_f-t}^\theta(\overrightarrow{X}_t) - (\overrightarrow{X}_0 - \overrightarrow{X}_t)\mathrm{d}(\overrightarrow{X}_0 - \overrightarrow{X}_t)\|^2\right] , \tag{46}$$

where $\mathrm{d}$ is the element-wise product.

**Cross-entropy objective $\mathfrak{L}_{\mathbf{CE}}$.** Instead of the $\mathfrak{L}_{L^2}$ loss suggested by the conditional expectation structure, we can consider a cross-entropy loss to fit our model to the correct distribution: classical derivations from the conditional log-likelihood $\sum_{\ell=1}^d \mathbb{E}\left[\log p_t^{\theta,\ell}(\overrightarrow{X}_{T_f-t}^\ell | \overrightarrow{X}_0^\ell)\right]$, where

$$p_t^{\theta,\ell}(x_{T_f-t}|x_0) = \begin{cases} d_t^{\theta,\ell}(x_{T_f-t}) & \text{if } x_{T_f-t} \neq x_0 \\ 1 - d_t^{\theta,\ell}(x_{T_f-t}) & \text{else} \end{cases} , \tag{47}$$

lead to the following cross entropy loss:

$$\mathfrak{L}_{\text{CE}}(\theta) = -\int_0^{T_f} \mathfrak{L}_{t,\text{CE}}(\theta) \mathrm{d}t , \tag{48}$$

where

$$\mathfrak{L}_{t,\mathrm{CE}}(\theta) = \mathbb{E}\left[\sum_{l=1}^{d} Y_t^{\ell} \log d_t^{\theta,\ell}(\overrightarrow{X}_{T_f-t}) + (1 - Y_t^{\ell}) \log\left(1 - d_t^{\theta,\ell}(\overrightarrow{X}_{T_f-t})\right)\right] , \quad Y_t^{\ell} = \begin{cases} 1 & \text{if } \overrightarrow{X}_0^{\ell} \neq \overrightarrow{X}_{T_f-t}^{\ell} , \\ 0 & \text{else .} \end{cases}$$

**Further improvements.** To address training efficiency, we inspect the average magnitude of the loss across the dataset, at each timestep. Indeed, the average value of $d_t^{\ell}$ is

$$\begin{aligned} w_{T_f-t} = \mathbb{E}\left[d_t^{\ell}(\overrightarrow{X}_{T_f-t})\right] &= \mathbb{E}\left[\mathbb{E}\left[d_t^{\ell}(\overrightarrow{X}_{T_f-t})\Big|\overrightarrow{X}_0\right]\right] \\ &= \mathbb{E}\left[\mathbb{P}\left(\overrightarrow{X}_0^{\ell} \neq \overrightarrow{X}_{T_f-t}^{\ell}\Big|\overrightarrow{X}_0\right)\right] \\ &= \frac{1}{2}\left(1 - \alpha_{T_f-t}\right) , \end{aligned} \quad (49)$$

as given by the formulas for the transition kernels of the forward process. We can see that the value of $w_t$ is close to zero for small values of $t$, which stalls the learning process. Empirically, we find that dividing the integrand of either loss terms $\mathfrak{L}_{\mathrm{L}^2}$ or $\mathfrak{L}_{\mathrm{CE}}$ by $w_t$ yields improvements. As a result, we modify the losses to counterbalance their diminishing magnitude across timesteps: $\mathfrak{L}_{t,\mathrm{L}^2}^w = \frac{\mathfrak{L}_{t,\mathrm{L}^2}^{\mathrm{denoiser}}}{w_t}$, $\mathfrak{L}_{t,\mathrm{CE}}^w = \frac{\mathfrak{L}_{t,\mathrm{CE}}}{w_t}$, and define the associated losses

$$\mathfrak{L}_{\mathrm{L}^2}^w(\theta) = \int_0^{T_f} \mathfrak{L}_{t,\mathrm{L}^2}^w(\theta)\mathrm{d}t , \quad \mathfrak{L}_{\mathrm{CE}}^w(\theta) = -\int_0^{T_f} \mathfrak{L}_{t,\mathrm{CE}}^w(\theta)\mathrm{d}t . \quad (50)$$

**Comparing $\mathfrak{L}_{\mathrm{L}^2}, \mathfrak{L}_{\mathrm{CE}}, \mathfrak{L}_{\mathrm{L}^2}^w, \mathfrak{L}_{\mathrm{CE}}^w$.**

In Figure 4, we plot the average loss per timestep, for a trained model on MNIST (following the specifications given in Appendix D.2)). It shows that, on average, the $\mathfrak{L}_{\mathrm{L}}$ loss effectively becomes a scaled variant of the cross-entropy objective, which is reflected in similar performance results. This corroborates our derivation that L2 acts as an effective lower bound to the log-likelihood. This also supports its relevancy with respect to the underlying structure of this generative model. It must be noted that both losses still benefit from positive synergies when used together.

Importantly, dividing by $w_t = (1 - \alpha_t)/2$ particularly helps at smaller timesteps, and keeps the loss values at the same magnitude across timesteps, enhancing training dynamics. This is illustrated in Figure 5, where scaling the losses with the $w$ scale factor consistently yields improvements.

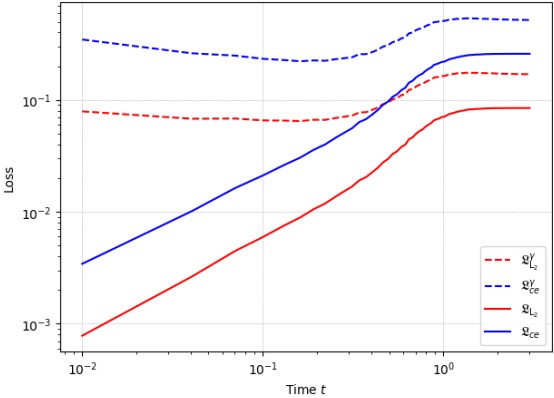

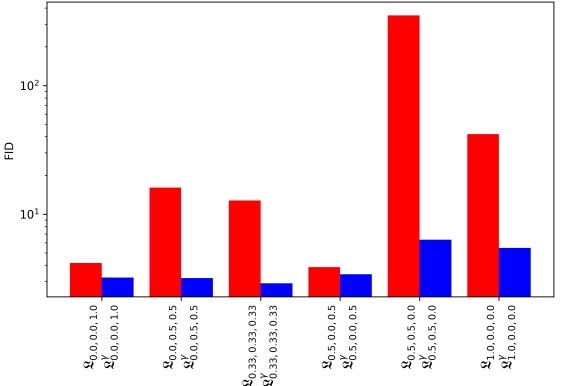

*Figure 4.* Comparison of $\mathfrak{L}_{\mathrm{L}^2}, \mathfrak{L}_{\mathrm{CE}}, \mathfrak{L}_{\mathrm{L}^2}^w, \mathfrak{L}_{\mathrm{CE}}^w$ average losses over timesteps. The two losses become scaled version of one another only when averaged over data, but otherwise benefit from positive synergies when mixed together.

*Figure 5.* FID↓, on MNIST, for models trained with $\mathfrak{L}_{\varpi}$ and $\mathfrak{L}_{\varpi}^w$ losses, evaluated using 200 reverse steps with the denoise-renoise sampler. Scaling with $w$ yields consistent improvements, with the best loss configuration $\mathfrak{L}_{1/3,1/3,1/3}^w$ involving all the methodological improvements we discussed.

**Final objective functions.** We choose a linear combination of the previous loss objectives, weighted by positive coefficients $\varpi_1, \varpi_2, \varpi_3$:

$$\mathfrak{L}_\varpi = \varpi_1 \mathfrak{L}_{L^2}^{den} + \varpi_2 \mathfrak{L}_e + \varpi_3 \mathfrak{L}_{CE} \,, \tag{51}$$

and, if we choose their version weighted by $1/w_t$:

$$\mathfrak{L}_\varpi^w = \varpi_1 \mathfrak{L}_{L^2}^w + \varpi_2 \mathfrak{L}_e + \varpi_3 \mathfrak{L}_{CE}^w \,. \tag{52}$$

---

**Algorithm 2** Training Algorithm for DMPM (Reparameterized Score)

---

**Require:** Dataset $\mathcal{D}$ of samples $X \in \{0,1\}^d$;
    Time horizon $T_f > 0$ and rate $\lambda > 0$;
    Parameterized discrete denoiser model $\{d_t^{\theta,\ell}(x) : \theta \in \Theta\}_{t,\ell,x}$;
    Derived score function $s_t^\theta := \frac{2\alpha_{T_f-t}}{1+\alpha_{T_f-t}} - \frac{4\alpha_{T_f-t}d_t^\theta}{1-\alpha_{T_f-t}^2}$ (score reparameterization (24));
    Define $\alpha_t$ as in (13), $f_t$ as in (19);
    Loss coefficients $\varpi_1, \varpi_2, \varpi_3 \geqslant 0$;

1:  **while** optimization has not converged **do**
2:     Sample a batch $\{X_i\}_{i=1}^B$ from $\mathcal{D}$.
3:     Draw $t_1, \ldots, t_B \overset{iid}{\sim} \mathrm{Unif}([0, T_f])$
4:     **Forward sampling: fast simulation via** $p_{t|0} = (p_{t|0}^1)^{\otimes d}$
5:     **for** $i = 1$ to $B$ **do**
6:       $\overrightarrow{X}_{i,0} \leftarrow X_i$
7:       $p_{T_f-t_i} \leftarrow (1 - \alpha_{T_f-t_i})/2$
8:       Compute $\overrightarrow{X}_{i,T_f-t_i}$ by flipping each bit of $\overrightarrow{X}_{i,0}$ independently with probability $p_{T_f-t_i}$
9:       **if** Scaling losses with average $d_t$ magnitude **then**
10:        $w_i \leftarrow (1 - \alpha_{T_f-t_i})/2$
11:       **else**
12:        $w_i \leftarrow 1$
13:       **end if**
14:     **end for**

15:     $\mathfrak{L}_{L^2}(\theta) \leftarrow \frac{1}{B}\sum_{i=1}^B \frac{1}{w_i}\|d_t^\theta(\overrightarrow{X}_{i,T_f-t_i}) - (\overrightarrow{X}_{i,T_f-t_i} - \overrightarrow{X}_{i,0})\mathrm{d}(\overrightarrow{X}_{i,t} - \overrightarrow{X}_{i,0})\|^2$
16:     $\mathfrak{L}_{CE}(\theta) \leftarrow \frac{1}{Bd}\sum_{i=1}^B \frac{1}{w_i}\sum_{\ell=1}^d \left(\mathbb{1}_{\overrightarrow{X}_0^\ell \neq \overrightarrow{X}_{T_f-t_i}^\ell} \log d_{t_i}^{\theta,\ell}(\overrightarrow{X}_{T_f-t_i}^\ell) + (1 - \mathbb{1}_{\overrightarrow{X}_0^\ell \neq \overrightarrow{X}_{T_f-t_i}^\ell})\log(1 - d_{t_i}^{\theta,\ell}(\overrightarrow{X}_{T_f-t_i}^\ell))\right)$
17:     $\mathfrak{L}_e(\theta) \leftarrow \frac{1}{B}\sum_{i=1}^B \sum_{\ell=1}^d \left(-s_{T_f-t_i}^{\theta,\ell}(\overrightarrow{X}_{t_i}) + (f_{T_f-t_i}^\ell(\overrightarrow{X}_{t_i}) - 1)\log(1 - s_{T_f-t_i}^{\theta,\ell}(\overrightarrow{X}_{t_i}))\right)$
18:     $\mathfrak{L}_\varpi(\theta) \leftarrow \varpi_1 \mathfrak{L}_{L^2}^{den} + \varpi_2 \mathfrak{L}_e + \varpi_3 \mathfrak{L}_{CE}$
19:     Perform a gradient step on $\mathfrak{L}_\varpi(\theta)$ w.r.t. $\theta$.
20:  **end while**
21:  **Return** the final parameter $\theta^\star$.

---

### C.5. Generative process and sampling procedures

Once we obtain our neural network $d_t^\theta$ approximating $d_t$, we use it to produce fresh samples that closely mimic the observed data. To do so, we first introduce a DMPM sampler based on the true reverse process. We then propose a slight modification, leveraging the distribution on indices available at each step, by flipping multiple bits instead of just one, using a flip-schedule. Finally, we derive a denoise-renoise sampler, solely based on the discrete denoiser structure of the problem, as inspired by similar lines of work in conitnuous diffusion.

**DMPM sampler.** A first sampling procedure is given in Algorithm 3. It is designed to be as close as possible to the true backward process, while enabling efficient parallelization when implemented. It consists in a piecewise-approximation of the functions of interest, parameterized by the choice of a time discretization grid $0 = t_0 < t_1 < \cdots < t_K = T_f$, which we call a **time-schedule**.

In Table 2, we give the different time-schedules we experiment with. We draw inspiration from numerous lines of work on continuous and discrete diffusion (Shi et al., 2024; Karras et al., 2022), in which these are common choices.

**DMPM sampler with flip-schedule** In Algorithm 4, we further take advantage of the specific structure of our backward process, by leveraging the distribution over indices given by the learned score model at each timestep $t$. Instead of flipping a single bit per timestep $t_k$, we flip a total of $M_{t_k}$ bits sampled without replacements from the given distribution. We call the sequence $\{M_t\}_{0 \leqslant t \leqslant T_f}$ the flip-schedule. When a time-schedule $\{t_k\}_{k=1}^K$ has been chosen, we also call the corresponding discrete sequence $\{M_{t_k}\}_{k=1}^K$ a flip-schedule.

In Table 3, we give the two flip-schedules we explore in this paper. The choice for the linear schedule is inspired from the philosophy of the masking schedule introduced in the context of masked diffusion by Shi et al. (2024).

| Time-schedule | Value of $t_k$ |
| --- | --- |
| Linear | $T_f \frac{k}{K}$ |
| Quadratic | $T_f \left(\frac{k}{K}\right)^2$ |
| Cosine | $T_f \cos\left(\frac{(1-k/K)\pi}{2}\right)$ |

*Table 2.* Different time schedules $(t_k)_{k=1}^K$ used in our experiments. $T_f$ denotes the final time, and $K$ is the number of reverse steps.

| Flip-schedule | Value of $M_t$ |
| --- | --- |
| Constant | $M$ |
| Linear | $M\frac{t}{T_f}$ |

*Table 3.* Different flip schedules $(M_t)_{0 \leqslant t \leqslant T_f}$ used in our experiments. In both schedules, $M$ is a constant to be fixed and controls the total number of bits flipped during generation.

**Denoise-renoise sampler.** In Algorithm 5, we introduce the following denoise/renoise cycle, interpreting the model output $d_t^\theta$ as the probability that each bit should be flipped at timestep $t$ to reach timestep 0. After doing a full denoise pass (from time $T_f \to 0$), we noise the sample with the transition kernel of the forward process (from time $0 \to T_f - \Delta$). Then we can do another denoise pass from $(T_f - \Delta) \to 0$, etc.

---

**Algorithm 3** Backward sampling of DMPM with piecewise-constant score

**Require:** Time horizon $T_f > 0$ and rate $\lambda > 0$;
  $K > 0$ number of reverse steps and time-schedule $0 = t_0 < t_1 < \cdots < t_K = T_f$;
  Flip-schedule, *i.e.*, sequence of positive integers $\{M_{t_k}\}_{k=1}^K$;
  Discrete denoiser model $d^\theta$;
  Derived score function $s_t^\theta := \frac{2\alpha_{T_f-t}}{1+\alpha_{T_f-t}} - \frac{4\alpha_{T_f-t}d_t^\theta}{1-\alpha_{T_f-t}^2}$ (score reparameterization (24));
  Define $\alpha_t$ as in (13);
1: $\overleftarrow{X}_0^\star \sim \text{Unif}(0,1)^{\otimes d}$
2: $E \sim \mathcal{E}(1)$
3: $\Lambda \leftarrow 0$
4: **for** $k = 0$ to $K - 1$ **do**
5: $\quad \overline{\lambda}_{t_k} \leftarrow \lambda \sum_{\ell=1}^d \left(1 - s_{t_k}^{\theta,\ell}\right)$
6: $\quad \Delta t_k \leftarrow t_{k+1} - t_k$
7: $\quad \Lambda \leftarrow \Lambda + \overline{\lambda}_{t_k} \Delta t_k$
8: $\quad$ **if** $\Lambda > E$ **then**
9: $\quad\quad \ell^\star \sim \text{Cate}\left(\left\{\frac{\lambda\left(1 - s_{t_k}^{\theta,\ell}\right)}{\overline{\lambda}_{t_k}}\right\}_{\ell=1}^d\right)$
10: $\quad\quad \overleftarrow{X}_{t_k}^{\star,\ell^\star} \leftarrow 1 - \overleftarrow{X}_{t_k}^{\star,\ell^\star}$
11: $\quad\quad \Lambda \leftarrow 0$
12: $\quad\quad E \sim \mathcal{E}(1)$
13: $\quad$ **end if**
14: $\quad \overleftarrow{X}_{t_{k+1}}^\star \leftarrow \overleftarrow{X}_{t_k}^\star$
15: **end for**
  **Output:** $\overleftarrow{X}_{T_f}^\star$

---

---

**Algorithm 4** Backward sampling of DMPM with piecewise-constant score and flip-schedule

---

**Require:** Time horizon $T_f > 0$ and rate $\lambda > 0$;

    $K > 0$ number of reverse steps and time-schedule $0 = t_0 < t_1 < \cdots < t_K = T_f$;

    Flip-schedule, *i.e.*, sequence of positive integers $\{M_{t_k}\}_{k=0}^{K}$;

    Discrete denoiser model $d^\theta$;

    Derived score function $s_t^\theta := \frac{2\alpha_{T_f-t}}{1+\alpha_{T_f-t}} - \frac{4\alpha_{T_f-t}d_t^\theta}{1-\alpha_{T_f-t}^2}$ (score reparameterization (24));

    Define $\alpha_t$ as in (13);

1: $\overleftarrow{X}_0^\theta \sim \mathrm{Unif}(0,1)^{\otimes d}$
2: $E \sim \mathcal{E}(1)$
3: $\Lambda \leftarrow 0$
4: **for** $k = 0$ to $K - 1$ **do**
5:     $\overline{\lambda}_{t_k} \leftarrow \lambda \sum_{\ell=1}^{d}\left(1 - s_{t_k}^{\theta,\ell}\right)$
6:     $\Delta t_k \leftarrow t_{k+1} - t_k$
7:     $\Lambda \leftarrow \Lambda + \overline{\lambda}_{t_k}\,\Delta t_k$
8:     **if** $\Lambda > E$ **then**
9:         $[\ell_1^\star, \ldots, \ell_M^\star] \sim \mathrm{Hypergeometric}\left(\left\{\frac{\lambda\left(1 - s_{t_k}^{\theta,\ell}\right)}{\overline{\lambda}_{t_k}}\right\}_{\ell=1}^{d}, M_{t_k}\right)$
10:         **for** $i = 1$ to $M_{t_k}$ **do**
11:             $\overleftarrow{X}_{t_k}^{\theta,\ell_i^\star} \leftarrow 1 - \overleftarrow{X}_{t_k}^{\theta,\ell_i^\star}$
12:         **end for**
13:         $\Lambda \leftarrow 0$
14:         $E \sim \mathcal{E}(1)$
15:     **end if**
16:     $\overleftarrow{X}_{t_{k+1}}^\theta \leftarrow \overleftarrow{X}_{t_k}^\theta$
17: **end for**
    **Output:** $\overleftarrow{X}_{T_f}^\theta$

---

---

**Algorithm 5** Denoise–Noise Cycling with a Discrete Denoiser Model

---

**Require:** Time horizon $T_f > 0$ and rate $\lambda > 0$;

    $K > 0$ number of reverse steps and time-schedule $0 = t_0 < t_1 < \cdots < t_K = T_f$;

    Discrete denoiser model $d^\theta$;

1: $\overleftarrow{X}_0^\theta \sim \mathrm{Unif}(0,1)^{\otimes d}$ {initial sample in $\{0,1\}^d$}
2: **for** $k = 0$ to $K - 1$ **do**
3:     **Denoise phase:**
4:     $d_{t_k} \leftarrow d_{t_k}^\theta\!\left(\overleftarrow{X}_{t_k}^\theta\right)$
5:     Compute $\overleftarrow{X}_{T_f}^\theta$ by flipping each component $l$ of $\overleftarrow{X}_{t_k}^\theta$ with probability $d_{t_k}^l$
6:     **Noise phase:**
7:     Sample $\overleftarrow{X}_{t_{k+1}}^\theta \sim p_{T_f-t_{k+1}|0}(\cdot\,|\overleftarrow{X}_{T_f}^\theta)$, as in Algorithm 2
8: **end for**
    **Output:** $\overleftarrow{X}_{T_f}^\theta$

---

# D. Experiments

All experiments are conducted using PyTorch. All the training and experiments are conducted on four NVIDIA RTX8000 GPU.

We use the score parameterization introduced in (45):

$$s_t^{\theta,\ell}(x) = \frac{2\alpha_{T_f-t}}{1+\alpha_{T_f-t}} - \frac{4\alpha_{T_f-t}d_t^{\theta,\ell}(x)}{1-\alpha_{T_f-t}^2} \; , \tag{53}$$

where the neural network $d_t^{\theta,\ell}(x)$ aims to approximate $d_t^\ell(x) = \mathbb{P}(\overrightarrow{X}_0^\ell \neq x^\ell | \overrightarrow{X}_{T_f-t} = x)$. Since the output of the neural network is $d_t^\theta(x) \in (0,1)^d$, we add a sigmoid activation function at the last layer.

We consider various loss configurations $\mathfrak{L}_\varpi, \mathfrak{L}_\varpi^w$ as introduced in (26), (52), with 6 choices of coefficients $(\varpi_1, \varpi_2, \varpi_3)$ normalized in the 2-simplex $\Delta_2 \subset \mathbb{R}^3$. We test all $2^3 - 1 = 7$ possible non-empty combinations, minus the single $\mathfrak{L}_e$ loss combination ($\varpi_2 = 1$), as the latter only acts as entropic regularization and does not perform well by itself. This lets us study the synergies between the different loss terms.

## D.1. Small dimension data

We first conduct experiments on a discrete data distribution $p$ supported on $\{0,1\}^d$. Each component of $X = (X_i)_{i=1}^d \sim p$ is independently distributed as a Bernoulli distribution with parameter $p_i$:

$$p(x) = \prod_{i=1}^d p_i(x_i) \; , \tag{54}$$

where the map $i \mapsto p_i$ forms a sawtooth-like pattern, oscillating linearly between $0.05$ and $0.95$, as can be seen in Figure 6.

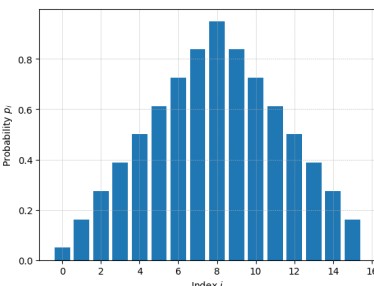

*Figure 6.* Sawtooth pattern used to define $i \mapsto p_i$, plotted with $d = 16$. Values oscillate linearly from $0.05$ to $0.95$, and back.

For training, we use $20\,000$ datapoints resampled at each epoch, and a batch size of $1024$. We train each model for $300$ epochs, using AdamW with a learning rate of 1e-3. We employ a network composed of multiple MLP blocks: 4 residual blocks, each consisting of two feed-forward layers of width $256$; layer normalization and SiLU activations in each block; a feed-forward embedding for the timesteps, mapping $\mathbb{R}$ to a hidden dimension of $256$, whose output is then injected into each residual block by an additional MLP of dimension $256 \times 256$.

For evaluation, we estimate each distribution with $20{,}000$ samples, and draw $1000$ vectors uniformly on the simplex $\Delta_d$ to compute our SWD metric (see Appendix D.3).

## D.2. Image data

We work on the binarized MNIST dataset, which we scale from $28 \times 28$ to $32 \times 32$ in order to fit in the U-Net architecture. We set the pixel value to $0$ if its intensity is below $0.5$, and to $1$ otherwise.

We compare DMPM to MD4 (masked diffusion, as in Shi et al. (2024)) and DFM (discrete flow matching, as in Gat et al. (2024)). We reimplement MD4 with the cosine schedule and the algorithms given in Appendix F of Shi et al. (2024). We implement DFM based on the Pytorch implementation in https://github.com/gle-bellier/discrete-fm,

and we use corrector sampling for better results.

For DMPM, we are using the cosine time-schedule and time horizon $T_f = 3$. For both MD4 and DFM, we set the mask value to the integer 2.

To establish a fair comparison, we use the same network model for every method. We use a U-Net following the implementation of (Nichol & Dhariwal, 2021) available in `https://github.com/openai/improved-diffusion`. We dimension the network as follows.

The first layer is an embedding layer of output dimension 32 and input dimension $d_{\text{input}}$, where $d_{\text{input}} = 2$ for DMPM (input values are either 0 and 1) and $d_{\text{input}} = 3$ for MD4 and DFM (input values are either $0, 1$ or the mask value 2).

We set the hidden layers to $[128, 256, 256, 256]$, fix the number of residual blocks to 2 at each level, and add self-attention block at resolution $16 \times 16$, using 4 heads. We use an exponential moving average with a rate of 0.99. We use the silu activation function at every layer. Timestep $t$ is fed to the model through the Transformer sinusoidal position embedding.

For DMPM and MD4, we set the number of output channels to 1 and add a sigmoid activation at the last layer. For DFM, we set the output channels to 3 and apply softmax channel-wise.

The optimizer is AdamW with learning rate 5e-4. We use the StepLR scheduler which scales the learning rate by $\gamma = .99$ every 400 steps. We train on MNIST for $120\,000$ steps with batch size 256. A single training run on MNIST takes approximately 6 hours per GPU, and requires about 6-12GB of VRAM for our settings.

To assess the quality of our generative models, we compute our metrics between $4\,000$ real images and $4\,000$ generated images. Generating $4\,000$ images with $1\,000$ reverse steps takes approximately 2 hours on one GPU.

### D.3. Metrics

For low-dimensional data, we use a custom sliced Wasserstein metric. For image data, in addition to the classical FID metric, we use a $F_1^{\text{DC}}$ summary score, based on the density and coverage metrics.

**$F_1^{\text{DC}}$ as summary metric of density-coverage** The density and coverage metrics are introduced in the setting of generative models by (Naeem et al., 2020). They assess the overlap of sample distributions using local geometric structures. Density measures how much the generated distribution is contained in the original data distribution (measuring quality), and coverage measures how much of the original data distribution is covered by the generated distribution (diversity).

These metrics are improvements of the precision and recall metrics for generative models (Kynkäänniemi et al., 2019). They offer different measures to characterize the performance of generative models. For instance they can decorrelate the negative effect of mode collapse from the negative effect of noisy/blurry generations, each of them decreasing respectively coverage and density, and have been of importance in recent studies, e.g., in heavy-tailed generative modeling (Shariatian et al., 2024; Yoon et al., 2023).

We consider a single summary $F_1^{\text{DC}}$ score, which we define as the harmonic mean of these two values:

$$F_1^{\text{DC}} = 2 \cdot \frac{\text{density} \cdot \text{coverage}}{\text{density} + \text{coverage}} . \tag{55}$$

**Sliced Wasserstein metric SWD.**

Since the state space of our dataset over $\{0, 1\}^d$ is of size $2^d$, we cannot work with histogram-based metrics, which would require exponentially many samples when $d$ increases.

We address this issue with our sliced Wasserstein metric SWD. This metric is defined between distributions $\mu, \nu$ on $\{0, 1\}^d$ as:

$$\text{SWD}(\mu, \nu) = \int_{\Delta_d} \text{W}\left(u_\# \mu, u_\# \nu\right) \text{d}u , \tag{56}$$

where, for $u \in \Delta_d$, the pushforward $u_\#$ is derived from the function

$$x \in \{0, 1\}^d \mapsto \langle u, x \rangle \in [0, 1] . \tag{57}$$

Simple Monte-Carlo averages are used to evaluate the integral with respect to the uniform distribution over the simplex $\Delta_d$,

and we compute the Wasserstein distance between the pushforward measures with the `pyemd` package (Laszuk, 2017).

## E. Additional results

In this section, we give grid images of generated samples for DMPM models trained on binarized MNIST, with the loss $\mathfrak{L}^w_{1/3,1/3,1/3}$.

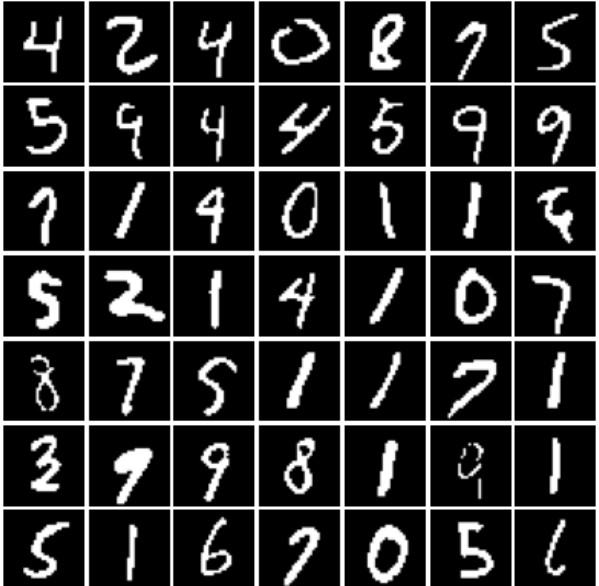 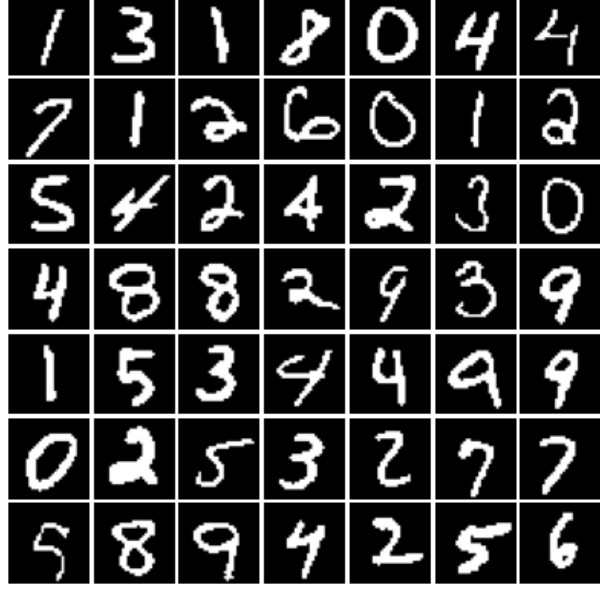

*Figure 7.* Default DMPM sampler, 25 reverse steps, cosine time-schedule, linear flip-schedule dimensioned for 1000 total bit flips.

*Figure 8.* Denoise-renoise sampler, 200 reverse steps, cosine time-schedule.

# F. Convergence of DMPMs

The proof of DMPMs' convergence requires understanding the backward dynamic under the canonical process point of view equivalent to the transition matrix point of view we provided in Section 1. We provide first some essential preliminaries on Poisson measures and the corresponding Itô's formula.

## F.1. Some basic facts on stochastic calculus for CTMCs

### F.1.1. POINT PROCESSES AND POISSON POINT PROCESSES

Let $(\mathsf{Y}, \mathcal{Y})$ be a measurable space. Let $\mathscr{M}_\mathsf{Y}$ be the set of non-negative (possibly infinite) integer-valued measures on $(\mathsf{Y}, \mathcal{Y})$. Let $\mathcal{M}_\mathsf{Y}$ be the smallest $\sigma$-field on $\mathscr{M}_\mathsf{Y}$ with respect to which the mappings $\boldsymbol{\mu} \in \mathscr{M}_\mathsf{Y} \mapsto \boldsymbol{\mu}(\mathsf{B}) \in \mathbb{N} \cup \{+\infty\}$, $\mathsf{B} \in \mathcal{Y}$, are measurable.

**Definition F.1** (Poisson random measure). An $(\mathscr{M}_\mathsf{Y}, \mathcal{M}_\mathsf{Y})$-valued random variable $\boldsymbol{\mu}$ (*i.e.*, a mapping $\boldsymbol{\mu} : \Omega \to \mathscr{M}_\mathsf{Y}$ defined on a probability space $(\Omega, \mathcal{F}, \mathbb{P})$ which is $\mathcal{F}/\mathcal{M}_\mathsf{Y}$-measurable) is called a *Poisson random measure* if

- for each $\mathsf{B} \in \mathcal{Y}$, $\boldsymbol{\mu}(\mathsf{B})$ is Poisson distributed; *i.e.*, $\mathbb{P}(\boldsymbol{\mu}(\mathsf{B}) = n) = \lambda_{\boldsymbol{\mu}}(\mathsf{B})^n \exp[-\lambda_{\boldsymbol{\mu}}(\mathsf{B})]/n!$ for any $n \in \mathbb{N}$ where $\lambda_{\boldsymbol{\mu}}(\mathsf{B}) = \mathbb{E}[\boldsymbol{\mu}(\mathsf{B})]$, $\mathsf{B} \in \mathcal{Y}$. Note that if $\lambda_{\boldsymbol{\mu}}(B) = \infty$, we understand that $\boldsymbol{\mu}(B) = \infty$ a.s.

- If $\mathsf{B}_1, \mathsf{B}_2. \ldots, \mathsf{B}_n \in \mathcal{Y}$ are disjoint, then $\boldsymbol{\mu}(\mathsf{B}_1), \boldsymbol{\mu}(\mathsf{B}_2), \ldots, \boldsymbol{\mu}(\mathsf{B}_n)$ are mutually independent.

Remark that it is easy to show that $\lambda_{\boldsymbol{\mu}}$ is a non-negative measure on $(\mathsf{Y}, \mathcal{Y})$ that uniquely determines the distribution of $\boldsymbol{\mu}$ by the monotone class theorem; see (Ikeda & Watanabe, 2014, Chapter I, Section 9). It is called the *mean measure* or the *intensity measure* of the Poisson random measure $\boldsymbol{\mu}$.

Let $(\mathsf{X}, \mathcal{X})$ be a measurable space. We define a *point function* $\mathfrak{p}$ on $\mathsf{X}$ by a mapping $\mathfrak{p} : D_{\mathfrak{p}} \to \mathsf{X}$, where its domain $D_{\mathfrak{p}}$ is a countable subset of $(0, \infty)$. $\mathfrak{p}$ defines a non-negative (possibly infinite) integer-valued measure $N_{\mathfrak{p}}$ on $(0, \infty) \times \mathsf{X}$ endowed with the product $\sigma$-field $\mathcal{B}((0, \infty)) \times \mathcal{X}$ by

$$N_{\mathfrak{p}}((0, t] \times \mathsf{U}) = \mathrm{Card}\left\{s \in D_{\mathfrak{p}} \ : \ s \leqslant t, \ \mathfrak{p}(s) \in \mathsf{U}\right\}, \quad t > 0, \quad \mathsf{U} \in \mathcal{X} .$$

A point process is obtained by randomizing the notion of point functions. Let $\Pi_\mathsf{X}$ be the set of point functions with values in $\mathsf{X}$ and $\mathcal{B}(\Pi_\mathsf{X})$ be the smallest $\sigma$-field on $\Pi_\mathsf{X}$ with respect to which the mappings $\mathfrak{p} \mapsto N_{\mathfrak{p}}((0, t] \times \mathsf{U}), t > 0, \mathsf{U} \in \mathcal{X}$, are measurable.

**Definition F.2** (Point process). A *point process* $\mathbf{p}$ on $\mathsf{X}$ is a $(\Pi_\mathsf{X}, \mathcal{B}(\Pi_\mathsf{X}))$-valued random variable, that is, a mapping $\mathbf{p} : \Omega \to \Pi_\mathsf{X}$ defined on a probability space $(\Omega, \mathcal{F}, \mathbb{P})$ which is $\mathcal{F}/\mathcal{B}(\Pi_\mathsf{X})$-measurable.

A point process $\mathbf{p}$ is called *stationary* if for every $t > 0$, $\mathbf{p}$ and $\theta_t \mathbf{p}$ have the same probability law, where $\theta_t \mathbf{p}$ is defined by $D_{\theta_t \mathbf{p}} = \{s \in (0, \infty) \ : \ s + t \in D_{\mathbf{p}}\}$ and $(\theta_t \mathbf{p})(s) = \mathbf{p}(s + t)$.

**Definition F.3** (Poisson point process). A point process $\mathbf{p}$ is called *Poisson* if $N_{\mathbf{p}}$ is a Poisson random measure on $(0, \infty) \times \mathsf{X}$. A Poisson point process is stationary if and only if its intensity measure $\hat{N}_{\mathbf{p}}(\mathrm{d}t\mathrm{d}x) = \mathbb{E}[N_{\mathbf{p}}(\mathrm{d}t\mathrm{d}x)]$ is of the form

$$\hat{N}_{\mathbf{p}}(\mathrm{d}t\mathrm{d}x) = \mathrm{d}t\mathrm{n}(\mathrm{d}x)$$

for some measure $\mathrm{n}$ on $(\mathsf{X}, \mathcal{X})$.

### F.1.2. STOCHASTIC INTEGRAL WITH RESPECT TO POINT PROCESS

Here, we review the construction and definition of stochastic integral with respect to Point Processes for completeness; see (Ikeda & Watanabe, 2014, Chapter II, Section 3).

Let $(\Omega, (\mathcal{F}_t)_{t \geqslant 0}, \mathbb{P})$ be a filtered probability space, where $\mathbb{P}$ is defined on the smallest $\sigma$-algebra containing all $\mathcal{F}_t, t \geqslant 0$ and $(\mathcal{F}_t)_{t \geqslant 0}$ satisfies the usual conditions (*i.e.*, right-continuity and completeness). A point process $\mathbf{p} = (\mathbf{p}(t))$ on $\mathsf{X}$ defined on $\Omega$ is called $(\mathcal{F}_t)$-*adapted* if for every $t > 0$ and $\mathsf{U} \in \mathcal{X}$, $N_{\mathbf{p}}(t, \mathsf{U}) = \sum_{s \in D_{\mathbf{p}}, s \leqslant t} \mathbb{1}_\mathsf{U}(\mathbf{p}(s))$ is $\mathcal{F}_t$-measurable. $\mathbf{p}$ is called $\sigma$-*finite* if there exist $\mathsf{U}_n \in \mathcal{X}, n = 1, 2, \ldots$ such that $\cup_n \mathsf{U}_n = \mathsf{X}$ and $\mathbb{E}[N_{\mathbf{p}}(t, \mathsf{U}_n)] < \infty$ for all $t > 0$ and $n = 1, 2, \ldots$. For a given $(\mathcal{F}_t)$-adapted, $\sigma$-finite point process $\mathbf{p}$, let $\Gamma_{\mathbf{p}} = \{U \in \mathcal{X}; \ \mathbb{E}[N_{\mathbf{p}}(t, \mathsf{U})] < \infty \text{ for all } t > 0\}$. If $\mathsf{U} \in \Gamma_{\mathbf{p}}$, then

$t \in [0, \infty) \mapsto N_{\mathbf{p}}(t, \mathsf{U})$ is an adapted, integrable increasing process and hence there exists a natural integrable increasing process $\hat{N}_{\mathbf{p}}(t, \mathsf{U})$ such that $\tilde{N}_{\mathbf{p}} : t \in (0, \infty) \mapsto \tilde{N}_{\mathbf{p}}(t, \mathsf{U}) = N_{\mathbf{p}}(t, \mathsf{U}) - \hat{N}_{\mathbf{p}}(t, \mathsf{U})$ is a martingale.

**Definition F.4.** An $(\mathcal{F}_t)$-adapted point process $\mathbf{p}$ on $(\Omega, \mathcal{F}, \mathbb{P})$ is said to be *of the class* (QL) (quasi left-continuous) (w.r.t. $(\mathcal{F}_t)$) if it is $\sigma$-finite and there exists $\hat{N}_{\mathbf{p}}$ such that

1. for $\mathsf{U} \in \Gamma_{\mathbf{p}}, t \in (0, \infty) \mapsto \hat{N}_{\mathbf{p}}(t, \mathsf{U})$ is a continuous $(\mathcal{F}_t)$-adapted increasing process,

2. for each $t \in (0, \infty)$ and a.a. $\omega \in \Omega, \mathsf{U} \mapsto \hat{N}_{\mathbf{p}}(t, \mathsf{U})$ is a $\sigma$-finite measure on $(\mathsf{X}, \mathcal{X})$,

3. for $\mathsf{U} \in \Gamma_{\mathbf{p}}, t \in (0, \infty) \mapsto \tilde{N}_{\mathbf{p}}(t, \mathsf{U}) = N_{\mathbf{p}}(t, \mathsf{U}) - \hat{N}_{\mathbf{p}}(t, \mathsf{U})$ is an $(\mathcal{F}_t)$-martingale.

The random measure $\hat{N}_{\mathbf{p}}$ is called the *compensator* of the point process $\mathbf{p}$ (or $N_{\mathbf{p}}$).

**Definition F.5.** A point process $\mathbf{p}$ is called an $(\mathcal{F}_t)$-*Poisson point process* if it is an $(\mathcal{F}_t)$-adapted, $\sigma$-finite Poisson point process such that $\{N_{\mathbf{p}}(t + h, \mathsf{U}) - N_{\mathbf{p}}(t, \mathsf{U})\}_{h > 0, \mathsf{U} \in \mathcal{X}}$ is independent of $\mathcal{F}_t$ for any $t \in [0, \infty)$.

An $(\mathcal{F}_t)$-Poisson point process is of class (QL) if and only if $t \mapsto \mathbb{E}[N_{\mathbf{p}}(t, \mathsf{U})]$ is continuous for $\mathsf{U} \in \Gamma_{\mathbf{p}}$; in this case, the compensator $\hat{N}_{\mathbf{p}}$ is given by $\hat{N}_{\mathbf{p}}(t, \mathsf{U}) = \mathbb{E}[N_{\mathbf{p}}(t, \mathsf{U})]$.

**Theorem F.6.** *Let $\mathbf{p}$ be a point process of class* (QL) *w.r.t.* $(\mathcal{F}_t)$ *on some state space* $(\mathsf{X}, \mathcal{X})$ *such that its compensator* $\hat{N}_{\mathbf{p}}(\mathrm{d}t\mathrm{d}x)$ *is a non-random $\sigma$-finite measure on* $[0, \infty) \times \mathsf{X}$. *Then $\mathbf{p}$ is an $(\mathcal{F}_t)$-Poisson point process. If, in particular,* $\hat{N}_{\mathbf{p}}(\mathrm{d}t\mathrm{d}x) = \mathrm{d}t\mathrm{n}(\mathrm{d}x)$ *where* $\mathrm{n}(\mathrm{d}x)$ *is a non-random $\sigma$-finite measure on* $\mathsf{X}$, *$\mathbf{p}$ is a stationary $(\mathcal{F}_t)$-Poisson point process with* $\mathrm{n}$ *as its characteristic measure.*

We are now going to discuss stochastic integrals w.r.t. a given point process of the class (QL). For this it is convenient to generalize the notion of predictable processes.

**Definition F.7** (Predictable functions). A real function $f(t, x, \omega)$ defined on $[0, \infty) \times \mathsf{X} \times \Omega$ is called $(\mathcal{F}_t)$-predictable if the mapping $(t, x, \omega) \mapsto f(t, x, \omega)$ is $\mathcal{S}/\mathcal{B}(\mathbb{R})$-measurable where $\mathcal{S}$ is the smallest $\sigma$-field on $[0, \infty) \times \mathsf{X} \times \Omega$ w.r.t. which all $g$ having the following properties are measurable:

1. for each $t > 0, (x, \omega) \mapsto g(t, x, \omega)$ is $\mathcal{X} \times \mathcal{F}_t$-measurable;

2. for each $(x, \omega), t \in [0, \infty) \mapsto g(t, x, \omega)$ is left continuous.

We introduce the following classes

$$
\begin{aligned}
F_{\mathbf{p}} &= \left\{ f(t, x, \omega); f \text{ is } (\mathcal{F}_t)\text{-predictable and for each } t > 0, \int_0^{t+} \int_{\mathsf{X}} |f(s, x, \omega)| N_{\mathbf{p}}(\mathrm{d}s\mathrm{d}x) < \infty \text{ a.s.} \right\}, \\
F_{\mathbf{p}}^1 &= \left\{ f(t, x, \omega); f \text{ is } (\mathcal{F}_t)\text{-predictable and for each } t > 0, \mathbb{E}\left[ \int_0^t \int_{\mathsf{X}} |f(s, x, \cdot)| \hat{N}_{\mathbf{p}}(\mathrm{d}s\mathrm{d}x) \right] < \infty \right\}.
\end{aligned}
\tag{58}
$$

**Theorem F.8** (Stochastic integral). *For $f \in F_{\mathbf{p}}$, the stochastic integral $\int_0^{t+} \int_{\mathsf{X}} f(s, x, \cdot) N_{\mathbf{p}}(\mathrm{d}s\mathrm{d}x)$ is well-defined a.s. and equals the absolutely convergent sum,*

$$
\int_0^{t+} \int_{\mathsf{X}} f(s, x, \cdot) N_{\mathbf{p}}(\mathrm{d}s\mathrm{d}x) = \sum_{s \leqslant t, s \in D_{\mathbf{p}}} f(s, \mathbf{p}(s), \cdot).
$$

*We then denote the stochastic integral w.r.t.* $\tilde{N}_{\mathbf{p}}$ *as*

$$
\int_0^{t+} \int_{\mathsf{X}} f(s, x, \cdot) \tilde{N}_{\mathbf{p}}(\mathrm{d}s\mathrm{d}x) = \int_0^{t+} \int_{\mathsf{X}} f(s, x, \cdot) N_{\mathbf{p}}(\mathrm{d}s\mathrm{d}x) - \int_0^t \int_{\mathsf{X}} f(s, x, \cdot) \hat{N}_{\mathbf{p}}(\mathrm{d}s\mathrm{d}x),
$$

*and the process $t \mapsto \int_0^{t+} \int_{\mathsf{X}} |f(s, x, \cdot)| \tilde{N}_{\mathbf{p}}(\mathrm{d}s\mathrm{d}x)$ is an $(\mathcal{F}_t)$-local martingale.*

*Now let $f \in F_{\mathbf{P}}^1$, the stochastic integral $\int_0^t \int_{\mathsf{X}} f(s, x, \cdot) \hat{N}_{\mathbf{P}}(\mathrm{d}s\mathrm{d}x)$ satisfies*

$$\mathbb{E}\left[\int_0^{t+} \int_{\mathsf{X}} |f(s, x, \cdot)| N_{\mathbf{P}}(\mathrm{d}s\mathrm{d}x)\right] = \mathbb{E}\left[\int_0^t \int_{\mathsf{X}} |f(s, x, \cdot)| \hat{N}_{\mathbf{P}}(\mathrm{d}s\mathrm{d}x)\right] .$$

*This implies, in particular, that $F_{\mathbf{P}}^1 \subset F_{\mathbf{P}}$. Then the stochastic integral $t \mapsto \int_0^{t+} \int_{\mathsf{X}} f(s, x, \cdot) \tilde{N}_{\mathbf{P}}(\mathrm{d}s\mathrm{d}x)$ with $f \in F_{\mathbf{P}}^1$ is an $(\mathcal{F}_t)$-true martingale.*

### F.1.3. Itô's formula for point process

Itô's formula is one of the most important tools in the study of semi-martingales. It provides us with the differential-integral calculus for sample functions of stochastic processes.

Let $(\Omega, (\mathcal{F}_t)_{t \geqslant 0}, \mathbb{P})$ be a filtered probability space given as above and let $(\mathsf{X}, \mathcal{X})$ be a measurable space. Define a $d$-dimensional semi-martingale $\mathbf{X}_t = (\mathbf{X}_t^1, \mathbf{X}_t^2, \ldots, \mathbf{X}_t^d)$ adapted to $(\mathcal{F}_t)_{t \geqslant 0}$, taking values in $\mathsf{X}$ by

$$\mathbf{X}_t^i = \mathbf{X}_0^i + M_t^i + A_t^i + \int_0^{t+} \int_{\mathsf{X}} f^i(s, x, \cdot) N_{\mathbf{P}}(\mathrm{d}s\mathrm{d}x) + \int_0^{t+} \int_{\mathsf{X}} g^i(s, x, \cdot) \tilde{N}_{\mathbf{P}}(\mathrm{d}s\mathrm{d}x) ,$$

for $i = 1, 2, \ldots, d$, where

- each $(M_t^i)_{t \geqslant 0} \in \mathcal{M}_2^{\mathrm{c,loc}}$ with

  $$\mathcal{M}_2^{\mathrm{c,loc}} = \{(M_t)_{t \geqslant 0}; (M_t)_{t \geqslant 0} \text{ is a locally square integrable } (\mathcal{F}_t)\text{-martingale}, M_0 = 0 \text{ a.s. }; t \mapsto X_t \text{ is continuous a.s.}\} ;$$

- each $(A_t^i)_{t \geqslant 0}$ is a continuous $(\mathcal{F}_t)$-adapted process satisfying $A_0^i = 0$ and for almost every $\omega \in \Omega$, the sample path $t \mapsto A_t^i(\omega)$ has bounded variation on each finite interval;

- $\mathbf{p}$ is a point process of the class (QL) w.r.t. $(\mathcal{F}_t)$ on some state such that $f^i(t, x, \omega) g^j(t, x, \omega) = 0$ for any $(i, j, t, x, \omega) \in \{1, \ldots, d\}^2 \times [0, \infty) \times \mathsf{X} \times \Omega$; furthermore, we assume that a constant $C > 0$ exists such that

  $$|g^i(t, x, \omega)| \leqslant C \quad \text{for all } (i, t, x, \omega) \in \{1, \ldots, d\} \times [0, \infty) \times \mathsf{X} \times \Omega ;$$

- each $\mathbf{X}_0^i$ is an $\mathcal{F}_0$-measurable random variable.

Denote also $f = (f^1, \ldots, f^d)$ and $g = (g^1, \ldots, g^d)$.

**Theorem F.9** (Itô's formula, (Ikeda & Watanabe, 2014, Chapter II, Section 5)). *Let $F$ be a function of class $\mathrm{C}^2$ on $\mathbb{R}^d$ and let $(\mathbf{X}_t)_{t \geqslant 0}$ be a $d$-dimensional semi-martingale given above. Then the stochastic process $(F(\mathbf{X}_t))_{t \geqslant 0}$ is also a semi-martingale w.r.t. $(\mathcal{F}_t)_{t \geqslant 0}$ and the following formula holds:*

$$\begin{aligned}
F(\mathbf{X}_t) - F(\mathbf{X}_0) = & \sum_{i=1}^d \int_0^t \frac{\partial F}{\partial x_i}(\mathbf{X}_s) \mathrm{d}M_s^i + \sum_{i=1}^d \int_0^t \frac{\partial F}{\partial x_i}(\mathbf{X}_s) \mathrm{d}A_s^i \\
& + \frac{1}{2} \sum_{i,j=1}^d \int_0^t \frac{\partial F^2}{\partial x_i \partial x_j}(\mathbf{X}_s) \mathrm{d}\langle M^i, M^j \rangle_s \\
& + \int_0^{t+} \int_{\mathsf{X}} \{F(\mathbf{X}_{s-} + f(s, x, \cdot)) - F(\mathbf{X}_{s-})\} N_{\mathbf{P}}(\mathrm{d}s\mathrm{d}x) \\
& + \int_0^{t+} \int_{\mathsf{X}} \{F(\mathbf{X}_{s-} + g(s, x, \cdot)) - F(\mathbf{X}_{s-})\} \tilde{N}_{\mathbf{P}}(\mathrm{d}s\mathrm{d}x) \\
& + \int_0^t \int_{\mathsf{X}} \left\{F(\mathbf{X}_s + g(s, x, \cdot)) - F(\mathbf{X}_s) - \sum_{i=1}^d g^i(s, x, \cdot) \frac{\partial F}{\partial x_i}(\mathbf{X}_s)\right\} \hat{N}_{\mathbf{P}}(\mathrm{d}s\mathrm{d}x) .
\end{aligned}$$

**Corollary F.10** (Itô's isometry). *Let $N_\mathbf{p}$, $\tilde{N}_\mathbf{p}$ and $\hat{N}_\mathbf{p}$ given as above. Then the following holds for real-valued functions $f \in F_\mathbf{p}^1$,*

$$\mathbb{E}\left[\left(\int_0^{t+}\int_\mathsf{X} f(s,x,\omega)\tilde{N}_\mathbf{p}(\mathrm{d}s\mathrm{d}x)\right)^2\right] = \mathbb{E}\left[\int_0^t\int_\mathsf{X} f^2(s,x,\omega)\hat{N}_\mathbf{p}(\mathrm{d}s\mathrm{d}x)\right] .$$

***Proof of Corollary F.10.*** Denote the process $(\mathbf{X}_t)_{t\geqslant 0}$ by

$$\mathbf{X}_t = \int_0^{t+}\int_\mathsf{X} f(s,x,\omega)\tilde{N}_\mathbf{p}(\mathrm{d}s\mathrm{d}x) ,$$

and apply Itô's formula (see Theorem F.9) for function $F(x) = x^2$, we get that

$$\begin{aligned}
\mathbf{X}_t^2 - \mathbf{X}_0^2 = &\int_0^{t+}\int_\mathsf{X} \left\{(\mathbf{X}_{s-} + f(s,x,\omega))^2 - \mathbf{X}_{s-}^2\right\} \tilde{N}_\mathbf{p}(\mathrm{d}s\mathrm{d}x) \\
&+ \int_0^t\int_\mathsf{X} \left\{(\mathbf{X}_s + f(s,x,\omega))^2 - \mathbf{X}_s^2 - 2\mathbf{X}_s f(s,x,\omega)\right\} \hat{N}_\mathbf{p}(\mathrm{d}s\mathrm{d}x) , \quad \text{for } t \geqslant 0 .
\end{aligned}$$

Taking expectations on both sides and using the martingale property of the stochastic integral with respect to $\tilde{N}_\mathbf{p}$ implies that

$$\mathbb{E}\left[\mathbf{X}_t^2 - \mathbf{X}_0^2\right] = \mathbb{E}\left[\int_0^t\int_\mathsf{X} f^2(s,x,\omega)\hat{N}_\mathbf{p}(\mathrm{d}s\mathrm{d}x)\right] .$$

Plugging the formula of $\mathbf{X}_t$ into the above and noting that $\mathbf{X}_0 = 0$ yields the desired equation,

$$\mathbb{E}\left[\left(\int_0^{t+}\int_\mathsf{X} f(s,x,\omega)\tilde{N}_\mathbf{p}(\mathrm{d}s\mathrm{d}x)\right)^2\right] = \mathbb{E}\left[\int_0^t\int_\mathsf{X} f^2(s,x,\omega)\hat{N}_\mathbf{p}(\mathrm{d}s\mathrm{d}x)\right] ,$$

and concludes the proof. $\qquad\square$

### F.1.4. APPLICATION TO CTMCS

Let $(\mathsf{X},\mathcal{X})$ be a measurable space with $\mathsf{X} \subset \mathbb{R}^d$ and let $(X_t)_{t\in[0,T_f]}$ be a CTMC on $\mathsf{X}$ associated with the jump rate function $r : (0,T_f] \times \mathsf{X} \to \mathbb{R}_+$ and the kernel $k_t(X_{t-},x)$ determining the probability of jumping into $x \in \mathsf{X}$ at time $t$, given a jump occurs from a current state $X_{t-}$. The generator $\mathfrak{q}_t$ is defined by $\mathfrak{q}_t(X_{t-},x) = r_t(X_{t-})k_t(X_{t-},x)$ represents the rate of jumping from the current state $X_{t-}$ to the new state $x \in \mathsf{X}$ at time $t \in (0,T_f]$. The CTMC $(X_t)_{t\in[0,T_f]}$ defines a Poisson point process $\mathbf{p}_X = (\mathbf{p}_X(t))_{t\in[0,T_f]}$ on $(\mathsf{X},\mathcal{X})$, where

$$\mathbf{p}_X : D_{\mathbf{p}_X} \subset (0,T_f] \to \mathsf{X} , \quad \mathbf{p}_X(t) = X_t , \; t \in D_{\mathbf{p}_X} ,$$

with $D_{\mathbf{p}_X}$ is the set of jump times of $(X_t)_{t\in[0,T_f]}$. We observe that $\mathbf{p}_X$ describes the new state after jumping at time $t \in (0,T_f]$ and it constructs a corresponding Poisson random measure $N_{\mathbf{p}_X}(\mathrm{d}t\mathrm{d}x)$ on $(0,T_f] \times \mathsf{X}$ by

$$\begin{aligned}
N_{\mathbf{p}_X}((0,t] \times \mathsf{U}) &= \mathrm{Card}\left\{s \in D_{\mathbf{p}_X} : s \leqslant t, \mathbf{p}_X(s) \in \mathsf{U}\right\} \\
&= \sum_{s\in D_{\mathbf{p}_X}} \delta_{(s,X_s)}((0,t] \times \mathsf{U}) \qquad \text{for } t \in (0,T_f], \quad \mathsf{U} \in \mathcal{X} ,
\end{aligned}$$

that counts the total jumps into $\mathsf{U} \in \mathcal{X}$ occurring during $(0,t]$. Then the random compensator $\bar{\mathrm{n}}^\mathfrak{q}$ of $N_{\mathbf{p}_X}$ is given by

$$\bar{\mathrm{n}}^\mathfrak{q}(\mathrm{d}t\mathrm{d}x) = \sum_{y\in\mathsf{X}} \mathfrak{q}(X_{t-},y)\mathbb{1}_{X_{t-}\neq y}\delta_y(\mathrm{d}x)\mathrm{d}t ,$$

since the corresponding compensated measure

$$\tilde{N}_{\mathbf{p}_X}^{\mathsf{q}}(\mathrm{d}t\mathrm{d}x) = N_{\mathbf{p}_X}(\mathrm{d}t\mathrm{d}x) - \bar{\mathsf{n}}^{\mathsf{q}}(\mathrm{d}t\mathrm{d}x)$$

is an $(\mathcal{F}_t)$-martingale, where $(\mathcal{F}_t)_{t\in[0,T_f]}$ denotes the right-continuous and complete natural filtration generated by the process $(X_t)_{t\in[0,T_f]}$. Indeed, we can show the martingale property of $\tilde{N}_{\mathbf{p}_X}^{\mathsf{q}}$ as follows. For any function $f \in F_{\mathbf{p}_X}^1$, where the class $F_{\mathbf{p}_X}^1$ is given in (58), define the following stochastic integrals:

$$\int_0^{t+} \int_{\mathsf{X}} f(s, x, \cdot) N_{\mathbf{p}_X}(\mathrm{d}s\mathrm{d}x) = \sum_{\substack{0<s\leqslant t \\ s\in D_{\mathbf{p}_X}}} f(s, \mathbf{p}_X(s), \cdot) \,,$$

$$\int_0^{t+} \int_{\mathsf{X}} f(s, x, \cdot) \tilde{N}_{\mathbf{p}_X}^{\mathsf{q}}(\mathrm{d}s\mathrm{d}x) = \int_0^{t+} \int_{\mathsf{X}} f(s, x, \cdot) N_{\mathbf{p}_X}(\mathrm{d}s\mathrm{d}x) - \int_0^t \int_{\mathsf{X}} f(s, x, \cdot)\bar{\mathsf{n}}^{\mathsf{q}}(\mathrm{d}s\mathrm{d}x) \,.$$

Then for $0 \leqslant s < t \leqslant T_f$ , we have

$$\mathbb{E}\left[\int_s^{t+} \int_{\mathsf{X}} f(z, x, \cdot) N_{\mathbf{p}_X}(\mathrm{d}z\mathrm{d}x)\middle|\mathcal{F}_s\right] = \mathbb{E}\left[\sum_{\substack{s<z\leqslant t \\ z\in D_{\mathbf{p}_X}}} f(z, X_z, \cdot)\middle|\mathcal{F}_s\right]$$

$$= \mathbb{E}\left[\int_{\mathsf{X}} \int_s^t f(z, x, \cdot)r(X_{z-})\frac{\mathsf{q}(X_{z-}, x)}{r(X_{z-})}\mathbb{1}_{X_{z-}\neq x}\mathrm{d}z\mathrm{d}x\middle|\mathcal{F}_s\right]$$

$$= \mathbb{E}\left[\int_s^t \int_{\mathsf{X}} f(z, x, \cdot)\mathsf{q}(X_{z-}, x)\mathbb{1}_{X_{z-}\neq x}\mathrm{d}x\mathrm{d}z\middle|\mathcal{F}_s\right]$$

$$= \mathbb{E}\left[\int_s^t \int_{\mathsf{X}} f(z, x, \cdot)\bar{\mathsf{n}}^{\mathsf{q}}(\mathrm{d}z\mathrm{d}x)\middle|\mathcal{F}_s\right] \,,$$

meaning that $\bar{\mathsf{n}}^{\mathsf{q}}$ is indeed the compensator of the Poisson random measure $N_{\mathbf{p}_X}$. With those notations in hand, we can decompose the CTMC $(X_t)_{t\in[0,T_f]}$ as

$$X_t = X_0 + \sum_{\substack{0<s\leqslant t \\ s\in D_{\mathbf{p}_X}}} (X_s - X_{s-}) = X_0 + \int_0^{t+} \int_{\mathsf{X}}(x - X_{s-})N_{\mathbf{p}_X}(\mathrm{d}s\mathrm{d}x) \,, \quad \text{for } t \in [0, T_f] \,,$$

under the assumption $f(s, x, \omega) := x - \omega_{s-} \in F_{\mathbf{p}_X}$, where $F_{\mathbf{p}_X}$ defined in (58). Applying Itô's formula to this process, for any function $F$ in $\mathrm{C}_b^2(\mathsf{X})$, we get that

$$F(X_t) - F(X_0) = \int_0^{t+} \int_{\mathsf{X}} \{F(X_{s-} + x - X_{s-}) - F(X_{s-})\} N_{\mathbf{p}_X}(\mathrm{d}s\mathrm{d}x)$$

$$= \int_0^{t+} \int_{\mathsf{X}} \{F(x) - F(X_{s-})\} N_{\mathbf{p}_X}(\mathrm{d}s\mathrm{d}x)$$

Expressing the Poisson random measure as $N_{\mathbf{p}_X} = \tilde{N}_{\mathbf{p}_X}^{\mathsf{q}} + \bar{\mathsf{n}}^{\mathsf{q}}$ and plugging it into the formula above yield

$$F(X_t) - F(X_0) - \int_0^t \int_{\mathsf{X}} \{F(x) - F(X_{s-})\} \bar{\mathsf{n}}^{\mathsf{q}}(\mathrm{d}s\mathrm{d}x) = \int_0^{t+} \int_{\mathsf{X}} \{F(x) - F(X_{s-})\} \tilde{N}_{\mathbf{p}_X}^{\mathsf{q}}(\mathrm{d}s\mathrm{d}x) \,.$$

In other words, the process

$$\left(F(X_t) - F(X_0) - \int_0^t \int_{\mathsf{X}} \{F(x) - F(X_{s-})\} \mathbb{1}_{X_{s-}\neq x}\mathsf{q}(X_{s-}, \mathrm{d}x)\mathrm{d}s\right)_{t\in[0,T_f]}$$

is an $(\mathcal{F}_t)$-local martingale as the compensated measure $\tilde{N}^{\mathfrak{q}}_{\mathbf{P}X}$ was shown to be an $(\mathcal{F}_t)$-martingale in the previous computation. It follows that for the CTMC $(X_t)_{t\in[0,T_f]}$ with generator $\mathfrak{q}$, Itô's formula asserts that the process

$$\left( F(X_t) - F(X_0) - \int_0^t \mathfrak{q}F(X_{s-})\mathrm{d}s \right)_{t\in[0,T_f]}$$

is an $(\mathcal{F}_t)$-local martingale for any function $F \in \mathrm{C}^2_b(\mathsf{X})$. This result aligns with Dynkin's formula.

### F.2. Canonical process point of view

We now want to give a description of the time reversal process as a controlled process like in the continuous setting in Conforti et al. (2025). To this purpose, we consider the following canonical setting. Let $\mathbb{D}_{T_f} = \mathbb{D}([0,T_f];\mathsf{X})$ be the canonical space of all càdlàg (right-continuous with left limits) paths from $[0,T_f]$ to $\mathsf{X} = \{0,1\}^d$, and let $(\mathbf{X}_t)_{t\in[0,T_f]}$ be the canonical process defined by

$$\mathbf{X}_t(\omega) = \omega_t, \quad \text{for } t \in [0,T_f] \, , \, (\omega_t)_{t\in[0,T_f]} \in \mathbb{D}_{T_f} \, .$$

We endow this space with the canonical filtration, that is, the right-continuous and complete augmentation of the filtration generated by $(\mathbf{X}_t)_{t\in[0,T_f]}$, denoted by $(\mathcal{F}_t)_{t\in[0,T_f]}$.

For any $\mathbb{P} \in \mathcal{P}(\mathbb{D}_{T_f})$ we denote by $\mathbb{E}_\mathbb{P}$ the corresponding expectation. For any $t \in [0,T_f]$, we denote by $\mathbb{P}_t$, the distribution of $\mathbf{X}_t$ under $\mathbb{P}$.

As usual convention, we simply denote the the random variable $\omega \mapsto U(\omega, t, x)$ as $U(t,x)$ for a predictable process $U : \mathbb{D}_{T_f} \times [0,T_f] \times \mathsf{X} \to \mathbb{R}$.

For any random generator $\mathbf{q} : \mathbb{D}_{T_f} \times [0,T_f] \times \mathsf{X}^2 \to \mathbb{R}$ and predictable process $U : \mathbb{D}_{T_f} \times [0,T_f] \times \mathsf{X} \to \mathbb{R}$, the new random generator $U\mathbf{q} : \mathbb{D}_{T_f} \times [0,T_f] \times \mathsf{X}^2 \to \mathbb{R}$ is defined as $(U\mathbf{q})(\omega, t, x, y) := U(\omega, t, y)\mathbf{q}(\omega, t, x, y)$ for $x \neq y$.

For convenience, we denote by $\mathbb{F}(\mathsf{X})$ the set of functions from $\mathsf{X}$ to $\mathbb{R}$.

We now follow the approach for time reversal used in Léonard (2012) to characterize the distribution of $(\mathbf{X}_t)_{t\in[0,T_f]}$ as the solution of a martingale problem.

**Definition F.11** (Martingale problem). Let $\mathbf{q} : \mathbb{D}_{T_f} \times [0,T_f] \times \mathsf{X}^2$ be a non-homogeneous predictable random generator. We say that $\mathbb{P} \in \mathcal{P}(\mathbb{D}_{T_f})$ solves *the Martingale problem* $\mathrm{MP}(\mathbf{q})$ with initial condition $\mu_0$, and write $\mathbb{P} \in \mathrm{MP}(\mathbf{q})$, if under $\mathbb{P}$, $\mathbf{X}_0$ has distribution $\mu_0$ and the process

$$\left( f(\mathbf{X}_t) - f(\mathbf{X}_0) - \int_0^t \mathbf{q}(s)f(\mathbf{X}_{s-})\mathrm{d}s \right)_{t\in[0,T_f]}$$

is an $(\mathcal{F}_t)_{t\in[0,T_f]}$-local martingale for any function $f \in \mathbb{F}(\mathsf{X})$, where we denote $\mathbf{q}(t)f(x) = \sum_{y\in\mathsf{X}} \mathbf{q}(\omega, t, x, y)f(y)$.

Note that by Itô's formula and Appendix F.1.4, if under $\mathbb{P}$, $(\mathbf{X}_t)_{t\in[0,T_f]}$ is a CTMC with generator $\mathbf{q}$, then $\mathbb{P}$ solves $\mathrm{MP}(\mathbf{q})$.

Recall that for the forward process under consideration, the generator $\overrightarrow{q}$ is defined as

$$\overrightarrow{q}(x,y) := \begin{cases} \lambda \, , & \text{if } y = \varphi^{(\ell)}(x) \text{ for some } \ell \in \{1, \ldots, d\} \, , \\ -\lambda d \, , & \text{if } y = x \, , \\ 0 \, , & \text{otherwise} \, , \end{cases} \tag{59}$$

where $\varphi^{(\ell)} : \mathsf{X} \to \mathsf{X}$ is the function which flips the $\ell$-th component for $\ell \in \{1, \ldots, d\}$. Note that by (15), $\gamma^d = \mathrm{Unif}(\mathsf{X})$ is an invariant distribution for the CTMC with generator $\overrightarrow{q}$, *i.e.*, for any measurable function $f$, $\sum_{x\in\mathsf{X}} \overrightarrow{p}_t f(x)\gamma^d(x) = \sum_{x\in\mathsf{X}} f(x)\gamma^d(x)$. In fact the transition density $\overrightarrow{p}_t$ is reversible with respect to $\gamma^d$ for any $t \in [0,T_f]$ using (15), *i.e.*, for any $x,y \in \mathsf{X}$ and $t \in [0,T_f]$,

$$\gamma^d(x)\overrightarrow{p}_t(x,y) = \gamma^d(y)\overrightarrow{p}_t(y,x) \, .$$

As a result, we get that for any $0 \leqslant t_1 < \ldots < t_n \leqslant T_f$, under $\overrightarrow{R}$, where $\overrightarrow{R}$ denoted the distribution of the CTMC with

generator $\overrightarrow{q}$ started at stationarity $\gamma^d$, $(\mathbf{X}_{t_1}, \ldots, \mathbf{X}_{t_n})$ has the same distribution as $(\mathbf{X}_{T_f - t_1}, \ldots, \mathbf{X}_{T_f - t_n})$ and therefore the reference path measure $\overrightarrow{R}$ is reversible, *i.e.*, $\overrightarrow{R} = \overleftarrow{R}$, where $\overleftarrow{R}$ is the distribution of $(\mathbf{X}_{T_f - t})_{t \in [0, T_f]}$ under $\overrightarrow{R}$.

Following Appendix F.1.4, for any $\mathbb{P} \in \mathcal{P}(\mathbb{D}_{T_f})$ such that $(\mathbf{X}_t)_{t \in [0, T_f]}$ is a CTMC with generator $\mathsf{q} : [0, T_f] \times \mathsf{X}^2 \to \mathbb{R}$ under $\mathbb{P}$, we denote by $\mathbf{p_X}$ the point process and by $N_\mathbf{X}$ the Poisson random measure associated with $(\mathbf{X}_t)_{t \in [0, T_f]}$.

We also define for any $(\omega_t)_{t \in [0, T_f]} \in \mathbb{D}_{T_f}$:

$$\bar{\mathrm{n}}^{\mathsf{q}}((\omega_t)_{t \in [0, T_f]}, \mathrm{d}t\mathrm{d}x) = \mathrm{n}^{\mathsf{q}}(t, \mathrm{d}x)\mathrm{d}t , \quad \mathrm{n}^{\mathsf{q}}(t, \mathrm{d}x) = \sum_{y \in \mathsf{X}} \mathbb{1}_{\omega_{t-} \neq y} \mathsf{q}_t(\omega_{t-}, y) \delta_y(\mathrm{d}x) . \tag{60}$$

By convention, we denote $\bar{\mathrm{n}}^{\mathsf{q}}((\mathbf{X}_t)_{t \in [0, T_f]}, \mathrm{d}t\mathrm{d}x)$ by $\bar{\mathrm{n}}^{\mathsf{q}}(\mathrm{d}t\mathrm{d}x)$ which corresponds to the compensator of $(\mathbf{X}_t)_{t \in [0, T_f]}$ under $\mathbb{P}$, if under this distribution $(\mathbf{X}_t)_{t \in [0, T_f]}$ is a CTMC with generator $\mathsf{q} : \mathsf{X}^2 \to \mathbb{R}$. Consequently, the compensated sum of jumps $\tilde{N}_\mathbf{X}^{\mathsf{q}} = N_\mathbf{X} - \bar{\mathrm{n}}^{\mathsf{q}}$ forms a martingale under $\mathbb{P}$.

### F.2.1. GIRSANOV'S THEOREM

As proven in Léonard (2012, Theorem 2.6-2.9), the relative entropy of two path measures associated to two jump processes can be expressed using the Young function

$$\varrho(a) := \mathrm{e}^a - a - 1 , \qquad \text{for } a \in \mathbb{R} .$$

Note that the convex conjugate $\varrho$ is equal to

$$\varrho^*(b) = (b + 1) \log(b + 1) - b , \text{ for } b > -1 ,$$

with the convention $\varrho^*(-1) = 1$ and $\varrho^*(b) = \infty$, for $b < -1$. We recall that the functions $\varrho$ and $\varrho^*$ are respectively equivalent to $a^2/2$ and $b^2/2$ near zero. Finally, we recall the definition of the function

$$h(a) = \varrho^*(a - 1) , \qquad a \in \mathbb{R} .$$

**Theorem F.12** (Girsanov's theorem). *(Léonard, 2012, Theorem 2.6-2.9) Let $\mathbb{P} \in \mathcal{P}(\mathbb{D}_{T_f})$ verifying $\mathrm{KL}(\mathbb{P}|\overrightarrow{R}) < \infty$. Then, there exists a unique predictable non-negative process $U : \mathbb{D}_{T_f} \times [0, T_f] \times \mathsf{X} \to [0, \infty)$ satisfying the integrability condition*

$$\mathbb{E}_\mathbb{P}\left[ \int_{[0, T_f] \times \mathsf{X}} \varrho^*(|U(t, x) - 1|) \bar{\mathrm{n}}^{\overrightarrow{q}}(\mathrm{d}t\mathrm{d}x) \right] < \infty , \tag{61}$$

*and $\mathbb{P} \in \mathrm{MP}(U\overrightarrow{q})$ where $(U\overrightarrow{q})(\omega, t, x, y) = U(\omega, t, y)\overrightarrow{q}(x, y)$ for $x \neq y$. Moreover, we have that*

$$\frac{\mathrm{d}\mathbb{P}}{\mathrm{d}\overrightarrow{R}}((\mathbf{X}_t)_{t \in [0, T_f]}) = \frac{\mathrm{d}\mathbb{P}_0}{\mathrm{d}\overrightarrow{R}_0}(\mathbf{X}_0) \exp\left( \int_{[0, T_f] \times \mathsf{X}} \log U(t, x) \tilde{N}_\mathbf{X}^{\overrightarrow{q}}(\mathrm{d}t\mathrm{d}x) - \int_{[0, T_f] \times \mathsf{X}} \varrho(\log U(t, x)) \bar{\mathrm{n}}^{\overrightarrow{q}}(\mathrm{d}t\mathrm{d}x) \right) ,$$

*and the $\mathrm{KL}$ divergence reads as*

$$\mathrm{KL}(\mathbb{P}|\overrightarrow{R}) = \mathrm{KL}(\mathbb{P}_0|\overrightarrow{R}_0) + \mathbb{E}_\mathbb{P}\left[ \int_{[0, T_f] \times \mathsf{X}} h(U(t, x)) \bar{\mathrm{n}}^{\overrightarrow{q}}(\mathrm{d}t\mathrm{d}x) \right] .$$

The proof of Theorem F.12 is given for completeness and is based on several technical lemmas, which we introduce in the following framework. Let $\mathbb{P}^{\mathsf{q}} \in \mathcal{P}(\mathbb{D}_{T_f})$ such that $(\mathbf{X}_t)_{t \in [0, T_f]}$ is a CTMC with generator $\mathsf{q} : [0, T_f] \times \mathsf{X}^2 \to \mathbb{R}$, *i.e.*, $\sum_{y \in \mathsf{X}} \mathsf{q}(t, x, y) = 0$ for any $(t, x) \in [0, T_f] \times \mathsf{X}$, and denote $\mathbf{p_X}$, $N_\mathbf{X}$, $\bar{\mathrm{n}}^{\mathsf{q}}$, $\mathrm{n}^{\mathsf{q}}$ and $\tilde{N}_\mathbf{X}^{\mathsf{q}}$ as in previous Section.

Let $\chi$ be a $\mathbb{R}$-valued predictable process on $\mathbb{D}_{T_f} \times [0, T_f] \times \mathsf{X}$ such that $\int_{[0, T_f] \times \mathsf{X}} \varrho(\chi_t(\omega, x)) \bar{\mathrm{n}}^{\mathsf{q}}(\mathrm{d}t\mathrm{d}x) < \infty$, $\mathbb{P}^{\mathsf{q}}$-a.s. Define

$$\mathsf{Z}_t^\chi := \exp\left( \int_{[0, t] \times \mathsf{X}} \chi_s(\omega, x) \tilde{N}_\mathbf{X}^{\mathsf{q}}(\mathrm{d}s\mathrm{d}x) - \int_{[0, t] \times \mathsf{X}} \varrho(\chi_s(\omega, x)) \bar{\mathrm{n}}^{\mathsf{q}}(\mathrm{d}s\mathrm{d}x) \right) , \quad \text{for } t \in [0, T_f] . \tag{62}$$

**Lemma F.13.** *Let $\mathbb{P}^{\mathfrak{q}} \in \mathcal{P}(\mathbb{D}_{T_f})$ such that $(\mathbf{X}_t)_{t \in [0, T_f]}$ is a CTMC with generator $\mathfrak{q} : [0, T_f] \times \mathsf{X}^2 \to \mathbb{R}$. Assume that $\chi$ is a $\mathbb{R}$-valued predictable process on $\mathbb{D}_{T_f} \times [0, T_f] \times \mathsf{X}$ satisfying the integrability condition*

$$\mathbb{E}_{\mathbb{P}^{\mathfrak{q}}} \int_{[0, T_f] \times \mathsf{X}} \varrho(\chi_s(\omega, x)) \bar{\mathrm{n}}^{\mathfrak{q}}(\mathrm{d}s\mathrm{d}x) < \infty . \tag{63}$$

*Then $\int_{[0,t] \times \mathsf{X}} \chi_s(\omega, x) \tilde{N}_{\mathbf{X}}^{\mathfrak{q}}(\mathrm{d}s\mathrm{d}x)$ is a local $\mathbb{P}^{\mathfrak{q}}$-martingale. Moreover, the process $(\mathrm{Z}_t^\chi)_{t \in [0, T_f]}$ defined in (62) is a local $\mathbb{P}^{\mathfrak{q}}$-martingale and a positive $\mathbb{P}^{\mathfrak{q}}$-supermartingale, which satisfies*

$$\mathrm{d}\mathrm{Z}_t^\chi = \mathrm{Z}_{t-}^\chi \int_\mathsf{X} (\mathrm{e}^{\chi_t(\omega, x)} - 1) \tilde{N}_{\mathbf{X}}^{\mathfrak{q}}(\mathrm{d}t\mathrm{d}x) .$$

*Proof of Lemma F.13.* By Theorem F.8, the process

$$M_t^\chi := \int_{[0, t] \times \mathsf{X}} \chi_s(\omega, x) \tilde{N}_{\mathbf{X}}^{\mathfrak{q}}(\mathrm{d}s\mathrm{d}x)$$

is a local $\mathbb{P}^{\mathfrak{q}}$-martingale. Denote $\mathrm{Y}_t^\chi := M_t^\chi - \int_{[0,t]} \beta_s \mathrm{d}s$ with $\beta_s := \int_\mathsf{X} \varrho(\chi_s(\omega, x)) \mathrm{n}^{\mathfrak{q}}(s, \mathrm{d}x)$. Applying Itô's formula provided in Theorem F.9 for the jump process $(\mathrm{Y}_t^\chi)_{t \in [0, T_f]}$ and for a function $f$ of class $\mathrm{C}^2(\mathbb{R})$ implies

$$\mathrm{d}f(\mathrm{Y}_t^\chi) = \left[ \int_\mathsf{X} \left[ f(\mathrm{Y}_{t-}^\chi + \chi_t(\omega, x)) - f(\mathrm{Y}_{t-}^\chi) - f'(\mathrm{Y}_{t-}^\chi) \cdot \chi_t(\omega, x) \right] \mathrm{n}^{\mathfrak{q}}(t, \mathrm{d}x) \right] \mathrm{d}t$$
$$+ f'(\mathrm{Y}_{t-}^\chi) \cdot \beta_t \mathrm{d}t + \mathrm{d}M_t^f , \quad \mathbb{P}^{\mathfrak{q}}\text{-a.s.} ,$$

where $M_t$ is given by

$$M_t^f = \int_{[0, t] \times \mathsf{X}} \left[ f(Y_{s-}^\chi + \chi_s(\omega, x)) - f(Y_{s-}^\chi) \right] \tilde{N}_{\mathbf{X}}^{\mathfrak{q}}(\mathrm{d}s\mathrm{d}x)$$

is a local $\mathbb{P}^{\mathfrak{q}}$-martingale, since the integrand is $\mathbb{R}$-valued predictable process and $\tilde{N}_{\mathbf{X}}^{\mathfrak{q}}$ forms a martingale under $\mathbb{P}^{\mathfrak{q}}$. Using this formula for $f(y) = \mathrm{e}^y$, we obtain

$$\mathrm{d}\mathrm{e}^{\mathrm{Y}_t^\chi} = \left[ \int_\mathsf{X} (\mathrm{e}^{\mathrm{Y}_{t-}^\chi + \chi_t(\omega, x)} - \mathrm{e}^{\mathrm{Y}_{t-}^\chi} - \mathrm{e}^{\mathrm{Y}_{t-}^\chi} \cdot \chi_t(\omega, x)) \mathrm{n}^{\mathfrak{q}}(t, \mathrm{d}x) \right] \mathrm{d}t - \mathrm{e}^{\mathrm{Y}_{t-}^\chi} \beta_t \mathrm{d}t + \mathrm{d}M_t^{\exp}$$
$$= \mathrm{e}^{\mathrm{Y}_{t-}^\chi} \beta_t \mathrm{d}t - \mathrm{e}^{\mathrm{Y}_{t-}^\chi} \beta_t \mathrm{d}t + \mathrm{d}M_t^{\exp} = \mathrm{d}M_t^{\exp} , \quad \mathbb{P}^{\mathfrak{q}}\text{-a.s.} .$$

This implies $\mathrm{Z}_t^\chi = \mathrm{e}^{\mathrm{Y}_t^\chi}$ is a local $\mathbb{P}^{\mathfrak{q}}$-martingale and, since $\mathrm{Z}_t^\chi$ is positive, we can conclude that $\mathrm{Z}_t^\chi$ is a $\mathbb{P}^{\mathfrak{q}}$-supermartingale thanks to Fatou's lemma. In addition, we have

$$\mathrm{d}M_t^{\exp} = \int_\mathsf{X} \left( \mathrm{e}^{\mathrm{Y}_{t-}^\chi + \chi_t(\omega, x)} - \mathrm{e}^{\mathrm{Y}_{t-}^\chi} \right) \tilde{N}_{\mathbf{X}}^{\mathfrak{q}}(\mathrm{d}t\mathrm{d}x) = \mathrm{e}^{\mathrm{Y}_{t-}^\chi} \int_\mathsf{X} (\mathrm{e}^{\chi_t(\omega, x)} - 1) \tilde{N}_{\mathbf{X}}^{\mathfrak{q}}(\mathrm{d}t\mathrm{d}x) ,$$

*i.e.*, $\mathrm{d}\mathrm{Z}_t^\chi = \mathrm{Z}_{t-}^\chi \int_\mathsf{X} (\mathrm{e}^{\chi_t(\omega, x)} - 1) \tilde{N}_{\mathbf{X}}^{\mathfrak{q}}(\mathrm{d}t\mathrm{d}x)$ and we conclude the proof of Lemma F.13. $\qquad\square$

We now define the stopping time for $k, j \geqslant 1$,

$$\sigma_j^k := \inf \left\{ t \in [0, T_f] : \int_{[0, t] \times \mathsf{X}} \varrho(\chi_s(\omega, x)) \bar{\mathrm{n}}^{\mathfrak{q}}(\mathrm{d}s\mathrm{d}x) \geqslant k \text{ or } \chi_t(\omega, \omega_t) \notin [-j, k] \right\} . \tag{64}$$

For $\mathbb{P} \in \mathcal{P}(\mathbb{D}_{T_f})$, let us denote $\mathbb{P}^{\sigma_j^k} := \mathbf{X}_\#^{\sigma_j^k} \mathbb{P}$ the law under $\mathbb{P}$ of the process $\mathbf{X}^{\sigma_j^k}$ which is stopped at the stopping time $\sigma_j^k$.

**Lemma F.14.** *Let $\mathbb{P}^{\mathfrak{q}} \in \mathcal{P}(\mathbb{D}_{T_f})$ such that $(\mathbf{X}_t)_{t \in [0, T_f]}$ is a CTMC with generator $\mathfrak{q} : [0, T_f] \times \mathsf{X}^2 \to \mathbb{R}$ and let $\chi$ be a $\mathbb{R}$-valued predictable process on $\mathbb{D}_{T_f} \times [0, T_f] \times \mathsf{X}$ satisfying the integrability condition (63). Let $(\mathrm{Z}_t^\chi)_{t \in [0, T_f]}$ be*

*defined in* (62) *and* $\sigma_j^k$ *be defined in* (64). *For all* $j, k \geqslant 1$, *the process* $(Z_t^{\sigma_j^k})_{t \in [0, T_f]}$ *defined as* $Z_t^{\sigma_j^k} := Z_t^{\chi_j^k}$, *is a genuine* $\mathbb{P}^q$-*martingale with* $\chi_j^k = \mathbb{1}_{[0, \sigma_j^k]}\chi$, *and the measure* $\mathbb{Q}_j^k$ *defined for any measurable function* $F : \mathbb{D}_{T_f} \to \mathbb{R}_+$ *by*

$$\mathbb{E}_{\mathbb{Q}_j^k}[F((\mathbf{X}_t)_{t \in [0, T_f]})] = \mathbb{E}_{\mathbb{P}^q}[F((\mathbf{X}_{t \wedge \sigma_j^k})_{t \in [0, T_f]})Z_{T_f}^{\sigma_j^k}], \quad i.e., \quad \mathbb{Q}_j^k = Z_{T_f}^{\sigma_j^k}(\mathbb{P}^q)^{\sigma_j^k}$$

*is a probability measure on* $\mathbb{D}_{T_f}$ *which satisfies*

$$\mathbb{Q}_j^k \in \mathrm{MP}(\mathbb{1}_{[0, \sigma_j^k]}\mathrm{e}^{\chi}q) .$$

**Proof of Lemma F.14.** Fix $j, k \geqslant 1$. We have

$$Z_t^{\sigma_j^k} = \exp\left(\int_{[0,t] \times \mathsf{X}} \chi_j^k \mathrm{d}\tilde{N}_{\mathbf{X}}^q - \int_{[0,t] \times \mathsf{X}} \varrho(\chi_j^k) \mathrm{d}\bar{\mathrm{n}}^q\right) ,$$

where $\chi_j^k = \mathbb{1}_{[0, \sigma_j^k]}\chi$ is predictable since $\chi$ is predictable and $\mathbb{1}_{[0, \sigma_j^k]}$ is left continuous. For simplicity, we drop the subscripts and superscripts and write $\tilde{\chi} = \chi_j^k$ and $\tilde{Z}_t = Z_t^{\sigma_j^k}$ for the rest of the proof. From the definition of $\sigma_j^k$, we obtain that $\mathbb{P}^q$ a.s.,

$$\int_{[0,t] \times \mathsf{X}} \varrho(\tilde{\chi}_s) \mathrm{d}\bar{\mathrm{n}}^q \leqslant k , \quad \text{and } \tilde{\chi}_t(\omega, \mathbf{X}_t) \in [-j, k] , \text{ for any } t \in [0, T_f] . \tag{65}$$

First, we prove that $(\tilde{Z}_t)$ is a $\mathbb{P}^q$-martingale. From Lemma F.13, $(\tilde{Z}_t)$ is a local martingale. Therefore, it is enough to show that for $t \in [0, T_f]$,

$$\mathbb{E}_{\mathbb{P}^q}[\tilde{Z}_t^p] < \infty , \quad \text{for some } p > 1 .$$

For $p > 1$, we have

$$\tilde{Z}_t^p = \exp\left(p\int_{[0,t] \times \mathsf{X}} \tilde{\chi}_s \mathrm{d}\tilde{N}_{\mathbf{X}}^q - p\int_{[0,t] \times \mathsf{X}} \varrho(\tilde{\chi}_s) \mathrm{d}\bar{\mathrm{n}}^q\right) \leqslant \exp\left(p\int_{[0,t] \times \mathsf{X}} \tilde{\chi}_s \mathrm{d}\tilde{N}_{\mathbf{X}}^q\right) ,$$

and

$$\exp\left(p\int_{[0,t] \times \mathsf{X}} \tilde{\chi}_s \mathrm{d}\tilde{N}_{\mathbf{X}}^q - \int_{[0,t] \times \mathsf{X}} \varrho(p\tilde{\chi}_s) \mathrm{d}\bar{\mathrm{n}}^q\right) \geqslant \exp\left(p\int_{[0,t] \times \mathsf{X}} \tilde{\chi}_s \mathrm{d}\tilde{N}_{\mathbf{X}}^q\right) \bigg/ C(k, p, t) ,$$

for some finite deterministic constant $0 < C(k, p, t) < \infty$. Indeed, $\mathbb{P}^q$-a.s., for any $s \in [0, T_f]$, it holds that $\varrho(p\tilde{\chi}_s) \leqslant \mathrm{e}^{k(p-1)}(\varrho(\tilde{\chi}_s) + k + 1)$ since $\tilde{\chi}_s \leqslant k$ and $p > 1$. It yields that $\mathbb{P}^q$-a.s., it holds

$$\exp\left(\int_{[0,t] \times \mathsf{X}} \varrho(p\tilde{\chi}_s) \mathrm{d}\bar{\mathrm{n}}^q\right) \leqslant \exp\left(\mathrm{e}^{k(p-1)}\int_{[0,t] \times \mathsf{X}} (\varrho(\tilde{\chi}_s) + k + 1)\mathrm{d}\bar{\mathrm{n}}^q\right)$$

$$\overset{(65)}{\leqslant} \exp\left(k\mathrm{e}^{k(p-1)} + (k+1)\mathrm{e}^{k(p-1)}\int_{[0,t] \times \mathsf{X}} 1\mathrm{d}\bar{\mathrm{n}}^q\right) \leqslant C(k, p, t) < \infty ,$$

where the last inequality follows from the formula of $\bar{\mathrm{n}}^q$ given in (60) and the fact that $\mathsf{X}$ is finite. This implies $\mathbb{P}^q$-a.s.,

$$\tilde{Z}_t^p \leqslant \exp\left(p\int_{[0,t] \times \mathsf{X}} \tilde{\chi}_s \mathrm{d}\tilde{N}_{\mathbf{X}}^q\right) \leqslant C(k, p, t)\exp\left(p\int_{[0,t] \times \mathsf{X}} \tilde{\chi}_s \mathrm{d}\tilde{N}_{\mathbf{X}}^q - \int_{[0,t] \times \mathsf{X}} \varrho(p\tilde{\chi}_s) \mathrm{d}\bar{\mathrm{n}}^q\right) . \tag{66}$$

On the other hand, applying Lemma F.13 for $p\tilde{\chi}$ yields that $\exp\left(p\int_{[0,t] \times \mathsf{X}} \tilde{\chi}_s \mathrm{d}\tilde{N}_{\mathbf{X}}^q - \int_{[0,t] \times \mathsf{X}} \varrho(p\tilde{\chi}_s) \mathrm{d}\bar{\mathrm{n}}^q\right)$ is a $\mathbb{P}^q$-

supermartingale, and we get

$$\mathbb{E}_{\mathbb{P}^{\mathfrak{q}}}\left[\exp\left(p\int_{[0,t]\times\mathsf{X}}\tilde{\chi}_s\mathrm{d}\tilde{N}_{\mathbf{X}}^{\mathfrak{q}} - \int_{[0,t]\times\mathsf{X}}\varrho(p\tilde{\chi}_s)\mathrm{d}\bar{\mathrm{n}}^{\mathfrak{q}}\right)\right] \leqslant \mathbb{E}_{\mathbb{P}^{\mathfrak{q}}}\left[\exp\left(p\int_{[0,0]\times\mathsf{X}}\tilde{\chi}_s\mathrm{d}\tilde{N}_{\mathbf{X}}^{\mathfrak{q}} - \int_{[0,0]\times\mathsf{X}}\varrho(p\tilde{\chi}_s)\mathrm{d}\bar{\mathrm{n}}^{\mathfrak{q}}\right)\right] = 1 \,.$$

Plugging this estimate into (66) gives

$$\mathbb{E}_{\mathbb{P}^{\mathfrak{q}}}[\tilde{Z}_t^p] \leqslant C(k,p,t) < \infty \,, \quad \text{for any } t \in [0,T_f] \,,$$

which allow us to conclude that $(\tilde{Z}_{T_f})$ is a $\mathbb{P}^{\mathfrak{q}}$-martingale (see, *e.g.*, Zitkovic, 2015). Thereby $\mathbb{E}_{\mathbb{P}^{\mathfrak{q}}}[\tilde{Z}_t] = \mathbb{E}_{\mathbb{P}^{\mathfrak{q}}}[\tilde{Z}_0] = 1$ for any $t \in [0,T_f]$ and it follows $\mathbb{Q}_j^k$ is a probability measure on $\mathbb{D}_{T_f}$.

Now, we show the second claim of Lemma F.14:

$$\mathbb{Q}_j^k \in \mathrm{MP}(\mathbb{1}_{[0,\sigma_j^k]}\mathrm{e}^{\chi}\mathfrak{q}) \,.$$

Let $\tau$ be a finitely valued stopping time which will be specified later, and for any function $f \in \mathbb{F}(\mathsf{X})$, we denote

$$\mathrm{F}_t = \sum_{0\leqslant s\leqslant t}\{f(\mathbf{X}_s) - f(\mathbf{X}_{s-})\} = \int_0^{t+}\int_{\mathsf{X}}\{f(x) - f(\mathbf{X}_{s-})\}N_{\mathbf{X}}(\mathrm{d}s\mathrm{d}x) \,, \quad \text{for } t \in [0,T_f] \,.$$

Recall that by Lemma F.13, the martingale $(\tilde{Z}_t)$ satisfies the followings for $\mathbb{P}^{\mathfrak{q}}$-a.s.,

$$\mathrm{d}\tilde{Z}_t = \mathbb{1}_{[0,\sigma_j^k]}(t)\tilde{Z}_{t-}\int_{\mathsf{X}}(\mathrm{e}^{\tilde{\chi}_t(\omega,x)} - 1)\tilde{N}_{\mathbf{X}}^{\mathfrak{q}}(\mathrm{d}t\mathrm{d}x) \,.$$

We have

$$\mathbb{E}_{\mathbb{Q}_j^k}\left[\sum_{0\leqslant s\leqslant t\wedge\tau}\{f(\mathbf{X}_s) - f(\mathbf{X}_{s-})\}\bigg|\mathcal{F}_0\right] = \mathbb{E}_{\mathbb{Q}_j^k}\left[\mathrm{F}_{t\wedge\tau}|\mathcal{F}_0\right] = \mathbb{E}_{\mathbb{P}^{\mathfrak{q}}}\left[\tilde{Z}_{t\wedge\tau\wedge\sigma_j^k}\mathrm{F}_{t\wedge\tau\wedge\sigma_j^k} - \tilde{Z}_0\mathrm{F}_0\bigg|\mathcal{F}_0\right] \,. \qquad (67)$$

Let us denote the two-dimensional process $(\mathrm{I}_t)_{t\in[0,T_f]} = (\mathrm{I}_t^1, \mathrm{I}_t^2)_{t\in[0,T_f]}$, where

$$\mathrm{I}_t^1 := \mathrm{F}_t = \int_{[0,t]\times\mathsf{X}}\underbrace{[f(x) - f(\mathbf{X}_{s-})]}_{=:v^1(s,x,\omega)}\tilde{N}_{\mathbf{X}}^{\mathfrak{q}}(\mathrm{d}s\mathrm{d}x) + \underbrace{\int_{[0,t]\times\mathsf{X}}[f(x) - f(\mathbf{X}_{s-})]\bar{\mathrm{n}}^{\mathfrak{q}}(\mathrm{d}s\mathrm{d}x)}_{=:A_t^1} \,,$$

and

$$\mathrm{I}_t^2 = \tilde{Z}_t = \int_{[0,t]\times\mathsf{X}}\underbrace{\tilde{Z}_{s-}(\mathrm{e}^{\tilde{\chi}_s(\omega,x)} - 1)}_{:=v^2(s,x,\omega)}\tilde{N}_{\mathbf{X}}^{\mathfrak{q}}(\mathrm{d}s\mathrm{d}x) \,.$$

Let $v = (v^1, v^2)$ and apply Itô's formula (see Theorem F.9) to the process $(\mathrm{I}_t)_{t\in[0,T_f]}$ using the function given by the product of the coordinates, treating $(A_t^1)_{t\in[0,T_f]}$ as a continuous, finite variation process adapted to the filtration $(\mathcal{F}_t)$,

$$\mathbb{E}_{\mathbb{P}^q}\left[\mathrm{F}_{t\wedge\tau\wedge\sigma_j^k}\tilde{\mathrm{Z}}_{t\wedge\tau\wedge\sigma_j^k} - \mathrm{F}_0\tilde{\mathrm{Z}}_0\Big|\mathcal{F}_0\right] = \mathbb{E}_{\mathbb{P}^q}\left[\mathrm{I}_{t\wedge\tau\wedge\sigma_j^k}^1\mathrm{I}_{t\wedge\tau\wedge\sigma_j^k}^2 - \mathrm{I}_0^1\mathrm{I}_0^2\Big|\mathcal{F}_0\right]$$

$$= \mathbb{E}_{\mathbb{P}^q}\left[\int_{[0,t\wedge\tau\wedge\sigma_j^k]\times\mathsf{X}}\left\{(\mathrm{I}_{s-}^1 + v^1(s,x,\omega))(\mathrm{I}_{s-}^2 + v^2(s,x,\omega)) - \mathrm{I}_{s-}^1\mathrm{I}_{s-}^2\right\}\tilde{N}_{\mathbf{X}}^q(\mathrm{d}s\mathrm{d}x)\Big|\mathcal{F}_0\right]$$

$$+ \mathbb{E}_{\mathbb{P}^q}\left[\int_{[0,t\wedge\tau\wedge\sigma_j^k]\times\mathsf{X}}\left\{(\mathrm{I}_s^1 + v^1(s,x,\omega))(\mathrm{I}_s^2 + v^2(s,x,\omega)) - \mathrm{I}_s^1\mathrm{I}_s^2 - \mathrm{I}_s^2v^1(s,x,\omega) - \mathrm{I}_s^1v^2(s,x,\omega)\right\}\bar{\mathrm{n}}^q(\mathrm{d}s\mathrm{d}x)\Big|\mathcal{F}_0\right]$$

$$+ \mathbb{E}_{\mathbb{P}^q}\left[\int_{[0,t\wedge\tau\wedge\sigma_j^k]\times\mathsf{X}}\mathrm{I}_s^2v^1(s,x,\omega)\bar{\mathrm{n}}^q(\mathrm{d}s\mathrm{d}x)\Big|\mathcal{F}_0\right]$$

$$= \mathbb{E}_{\mathbb{P}^q}\left[\int_{[0,t\wedge\tau\wedge\sigma_j^k]\times\mathsf{X}}\left\{\mathrm{I}_s^2v^1(s,x,\omega) + v^1(s,x,\omega)v^2(s,x,\omega)\right\}\bar{\mathrm{n}}^q(\mathrm{d}s\mathrm{d}x)\Big|\mathcal{F}_0\right] \quad \text{(as } \tilde{N}_{\mathbf{X}}^q \text{ is a } \mathbb{P}^q\text{-martingale)}$$

$$= \mathbb{E}_{\mathbb{P}^q}\left[\int_{[0,t\wedge\tau\wedge\sigma_j^k]\times\mathsf{X}}\left\{\tilde{\mathrm{Z}}_sv^1(s,x,\omega) + v^1(s,x,\omega)\tilde{\mathrm{Z}}_{s-}\left(\mathrm{e}^{\tilde{\chi}_s(\omega,x)} - 1\right)\right\}\bar{\mathrm{n}}^q(\mathrm{d}s\mathrm{d}x)\Big|\mathcal{F}_0\right],$$

where we reduce the stochastic integral w.r.t. $\tilde{N}_{\mathbf{X}}^q$ as it is a local $\mathbb{P}^q$-martingale, since the integrand is $\mathbb{R}$-valued predictable process and $\tilde{N}_{\mathbf{X}}^q$ forms a martingale under $\mathbb{P}^q$. Since $\tilde{Z}_s = \tilde{Z}_{s-}$ for Lebesgue almost all $s \in [0, t \wedge \tau \wedge \sigma_j^k]$ (Mozumder, 2009, Proposition 2.1) and $\bar{\mathrm{n}}^q$ is atomless in time, the calculation follows

$$\mathbb{E}_{\mathbb{P}^q}\left[\mathrm{F}_{t\wedge\tau\wedge\sigma_j^k}\tilde{\mathrm{Z}}_{t\wedge\tau\wedge\sigma_j^k} - \mathrm{F}_0\tilde{\mathrm{Z}}_0\Big|\mathcal{F}_0\right] = \mathbb{E}_{\mathbb{P}^q}\left[\int_{[0,t\wedge\tau\wedge\sigma_j^k]\times\mathsf{X}}\tilde{\mathrm{Z}}_sv^1(s,x,\omega)\left(1 + \mathrm{e}^{\tilde{\chi}_s(\omega,x)} - 1\right)\bar{\mathrm{n}}^q(\mathrm{d}s\mathrm{d}x)\Big|\mathcal{F}_0\right]$$

$$= \mathbb{E}_{\mathbb{P}^q}\left[\int_{[0,t\wedge\tau\wedge\sigma_j^k]\times\mathsf{X}}\tilde{\mathrm{Z}}_sv^1(s,x,\omega)\mathrm{e}^{\tilde{\chi}_s(\omega,x)}\bar{\mathrm{n}}^q(\mathrm{d}s\mathrm{d}x)\Big|\mathcal{F}_0\right]. \tag{68}$$

Denote $\mathrm{G}_t := \int_{[0,t]\times\mathsf{X}} v^1(s,x,\omega)\mathrm{e}^{\tilde{\chi}_s(\omega,x)}\bar{\mathrm{n}}^q(\mathrm{d}s\mathrm{d}x)$. Applying Itô's formula for the process $(\mathrm{G}_t, \tilde{Z}_t)$ analogously as argued before, we obtain that

$$\mathbb{E}_{\mathbb{P}^q}\left[\tilde{\mathrm{Z}}_{t\wedge\tau\wedge\sigma_j^k}\mathrm{G}_{t\wedge\tau\wedge\sigma_j^k} - \tilde{\mathrm{Z}}_0\mathrm{G}_0\Big|\mathcal{F}_0\right]$$

$$= \mathbb{E}_{\mathbb{P}^q}\left[\int_{[0,t\wedge\tau\wedge\sigma_j^k]\times\mathsf{X}}\left\{\mathrm{G}_{s-}(\tilde{\mathrm{Z}}_{s-} + v^2(s,x,\omega)) - \mathrm{G}_{s-}\tilde{\mathrm{Z}}_{s-}\right\}\tilde{N}_{\mathbf{X}}^q(\mathrm{d}s\mathrm{d}x)\Big|\mathcal{F}_0\right]$$

$$+ \mathbb{E}_{\mathbb{P}^q}\left[\int_{[0,t\wedge\tau\wedge\sigma_j^k]\times\mathsf{X}}\left\{(\mathrm{G}_s(\tilde{\mathrm{Z}}_s + v^2(s,x,\omega)) - \mathrm{G}_s\tilde{\mathrm{Z}}_s - \mathrm{G}_sv^2(s,x,\omega)\right\}\bar{\mathrm{n}}^q(\mathrm{d}s\mathrm{d}x)\Big|\mathcal{F}_0\right]$$

$$+ \mathbb{E}_{\mathbb{P}^q}\left[\int_{[0,t\wedge\tau\wedge\sigma_j^k]\times\mathsf{X}}\tilde{\mathrm{Z}}_sv^1(s,x,\omega)\mathrm{e}^{\tilde{\chi}_s(\omega,x)}\bar{\mathrm{n}}^q(\mathrm{d}s\mathrm{d}x)\Big|\mathcal{F}_0\right]$$

$$= \mathbb{E}_{\mathbb{P}^q}\left[\int_{[0,t\wedge\tau\wedge\sigma_j^k]\times\mathsf{X}}\tilde{\mathrm{Z}}_sv^1(s,x,\omega)\mathrm{e}^{\tilde{\chi}_s(\omega,x)}\bar{\mathrm{n}}^q(\mathrm{d}s\mathrm{d}x)\Big|\mathcal{F}_0\right], \tag{69}$$

as the stochastic integral w.r.t. $\tilde{N}_{\mathbf{X}}^q$ is a local $\mathbb{P}^q$-martingale, since the integrand is $\mathbb{R}$-valued predictable process and $\tilde{N}_{\mathbf{X}}^q$ forms a martingale under $\mathbb{P}^q$. Combining (67), (68) and (69) implies

$$\mathbb{E}_{\mathbb{Q}_j^k}\left[\sum_{0\leqslant s\leqslant t\wedge\tau}\{f(\mathbf{X}_s)-f(\mathbf{X}_{s-})\}\bigg|\mathcal{F}_0\right]=\mathbb{E}_{\mathbb{P}^{\mathfrak{q}}}\left[\tilde{Z}_{t\wedge\tau\wedge\sigma_j^k}\mathrm{G}_{t\wedge\tau\wedge\sigma_j^k}\bigg|\mathcal{F}_0\right]\quad(\text{since }\mathrm{G}_0=0)$$

$$=\mathbb{E}_{\mathbb{Q}_j^k}\left[\mathrm{G}_{t\wedge\tau}|\mathcal{F}_0\right]$$

$$=\mathbb{E}_{\mathbb{Q}_j^k}\left[\int_{[0,t\wedge\tau]\times\mathsf{X}}\{f(x)-f(\mathbf{X}_{s-})\}\mathrm{e}^{\tilde{\chi}_s(\omega,x)}\bar{\mathrm{n}}^{\mathfrak{q}}(\mathrm{d}s\mathrm{d}x)\bigg|\mathcal{F}_0\right]$$

$$=\mathbb{E}_{\mathbb{Q}_j^k}\left[\int_{[0,t\wedge\tau]\times\mathsf{X}}\{f(x)-f(\mathbf{X}_{s-})\}\mathbb{1}_{\mathbf{X}_{s-}\neq x}\mathrm{e}^{\tilde{\chi}_s(\omega,x)}\mathfrak{q}_s(\mathbf{X}_{s-},\mathrm{d}x)\mathrm{d}s\bigg|\mathcal{F}_0\right]\ .$$

Recall that the random generator $\mathrm{e}^{\tilde{\chi}}\mathfrak{q}:\mathbb{D}_{T_f}\times[0,T_f]\times\mathsf{X}^2\to\mathbb{R}$ is defined by $(\mathrm{e}^{\tilde{\chi}}\mathfrak{q})(\omega,t,x,y):=\mathrm{e}^{\tilde{\chi}(\omega,t,y)}\mathfrak{q}(w,t,x,y)=\mathrm{e}^{\tilde{\chi}_t(\omega,y)}\mathfrak{q}(t,x,y)$ for $y\neq x$, since the generator $\mathfrak{q}$ under consideration is deterministic. Denote by $\bar{\mathrm{n}}^{\mathrm{e}^{\tilde{\chi}}\mathfrak{q}}$ the corresponding jump kernel for $\omega=(\omega_t)_{t\in[0,T_f]}\in\mathbb{D}_{T_f}$ and $(t,x)\in[0,T_f]\times\mathsf{X}$,

$$\bar{\mathrm{n}}^{\mathrm{e}^{\tilde{\chi}}\mathfrak{q}}(\omega,\mathrm{d}t\mathrm{d}x):=\mathbb{1}_{\omega_{t-}\neq x}(\mathrm{e}^{\tilde{\chi}}\mathfrak{q})(\omega,t,\omega_{t-},\mathrm{d}x)\mathrm{d}t$$

$$=\mathbb{1}_{\omega_{t-}\neq x}\mathrm{e}^{\tilde{\chi}_t(\omega,x)}\mathfrak{q}_t(\omega_{t-},\mathrm{d}x)\mathrm{d}t\ ,$$

then the previous equation rewrites

$$\mathbb{E}_{\mathbb{Q}_j^k}\left[f(\mathbf{X}_{t\wedge\tau})-f(\mathbf{X}_0)|\mathcal{F}_0\right]=\mathbb{E}_{\mathbb{Q}_j^k}\left[\int_{[0,t\wedge\tau]\times\mathsf{X}}\{f(x)-f(\mathbf{X}_{s-})\}\bar{\mathrm{n}}^{\mathrm{e}^{\tilde{\chi}}\mathfrak{q}}(\mathrm{d}s\mathrm{d}x)\bigg|\mathcal{F}_0\right]$$

$$=\mathbb{E}_{\mathbb{Q}_j^k}\left[\int_{[0,t\wedge\tau]}(\mathrm{e}^{\tilde{\chi}}\mathfrak{q})(s)f(\mathbf{X}_{s-})\mathrm{d}s\bigg|\mathcal{F}_0\right]\ .$$

Choosing $\tau$ such that the above terms are meaningful, we conclude that $\mathbb{Q}_j^k\in\mathrm{MP}(\mathrm{e}^{\chi_j^k}\mathfrak{q})$ and finish the proof. $\qquad\square$

***Proof of Theorem F.12.*** This proof is an adaptation of Theorem 2.6 in Léonard (2012) based on technical lemmas provided above applying on the reference measure $\overrightarrow{R}\in\mathrm{MP}(\overrightarrow{q})$ defined at the beginning of Section F. Consider $\|.\|_\varrho$ defined as

$$\|\phi\|_\varrho:=\inf\left\{a>0\ :\ \mathbb{E}_\mathbb{P}\left[\int_{[0,T_f]\times\mathsf{X}}\varrho(\phi/a)\mathrm{d}\bar{\mathrm{n}}^{\overrightarrow{q}}\right]\leqslant 1\right\}\ ,$$

for any $\phi\in S_\varrho$, where $S_\varrho$ is the Orlicz space defined by

$$S_\varrho:=\left\{\phi:\mathbb{D}_{T_f}\times[0,T_f]\times\mathsf{X}\to\mathbb{R}\text{ measurable s.t. }\mathbb{E}_\mathbb{P}\left[\int_{[0,T_f]\times\mathsf{X}}\varrho(b|\phi|)\mathrm{d}\bar{\mathrm{n}}^{\overrightarrow{q}}\right]<\infty,\text{ for any }b\geqslant 0\right\}\ .$$

It is well-known that it is a norm called the Luxemburg norm of $S_\varrho$.

Furthermore, for any $\chi\in S_\varrho$, by Theorem F.8, the process

$$M_t^{\chi,\overrightarrow{q}}:=\int_{[0,t]\times\mathsf{X}}\chi_s(\omega,x)\tilde{N}_{\mathbf{X}}^{\overrightarrow{q}}(\mathrm{d}s\mathrm{d}x)$$

is a local $\overrightarrow{R}$-martingale. Then, we define the two stochastic processes $(Y_t^{\chi,\overrightarrow{q}})_{t\in[0,T_f]}$ and $(Z_t^{\chi,\overrightarrow{q}})_{t\in[0,T_f]}$ as $Y_t^{\chi,\overrightarrow{q}}:=M_t^{\chi,\overrightarrow{q}}-\int_{[0,t]}\beta_s^{\chi,\overrightarrow{q}}\mathrm{d}s$ with $\beta_s^{\chi,\overrightarrow{q}}:=\int_\mathsf{X}\varrho(\chi_s(\omega,x))\mathrm{n}^{\overrightarrow{q}}(s,\mathrm{d}x)$, and $Z_t^{\chi,\overrightarrow{q}}=\mathrm{e}^{Y_t^{\chi,\overrightarrow{q}}}$. By Lemma F.13, the process $Z_{T_f}^{\chi,\overrightarrow{q}}$ is

a $\overrightarrow{R}$-supermartingale, thus $0 < \mathbb{E}_{\overrightarrow{R}}\left[Z_{T_f}^{\chi;\overrightarrow{q}}\right] \leqslant 1$.

For $\mathbb{P} \in \mathcal{P}(\mathbb{D}_{T_f})$ such that $\mathrm{KL}(\mathbb{P}|\overrightarrow{R}) < \infty$, the Donsker-Varadhan variational formulation of the KL implies that

$$\mathrm{KL}(\mathbb{P}|\overrightarrow{R}) = \sup\left\{\int u\mathrm{d}\mathbb{P} - \log\int \mathrm{e}^u \mathrm{d}\overrightarrow{R} : \quad u \text{ measurable and such that } \int \mathrm{e}^u \mathrm{d}\overrightarrow{R} < \infty\right\} . \tag{70}$$

For any $\chi \in S_\varrho$, choosing $u = \mathrm{Y}_{T_f}^{\chi;\overrightarrow{q}}$ and noting that $\log \mathbb{E}_{\overrightarrow{R}}\left[\mathrm{Z}_{T_f}^{\chi;\overrightarrow{q}}\right] \leqslant \log 1 = 0$, we derive

$$\mathbb{E}_{\mathbb{P}}\left[\int_{[0,T_f]\times\mathsf{X}} \chi_t(\omega,x)\tilde{N}_{\mathbf{X}}^{\overrightarrow{q}}(\mathrm{d}t\mathrm{d}x) - \int_{[0,T_f]\times\mathsf{X}} \varrho(\chi_t(\omega,x))\bar{\mathrm{n}}^{\overrightarrow{q}}(\mathrm{d}t\mathrm{d}x)\right] \leqslant \mathrm{KL}(\mathbb{P}|\overrightarrow{R}) .$$

Therefore, for any $\chi \in S_\varrho$,

$$\mathbb{E}_{\mathbb{P}}\left[\int_{[0,T_f]\times\mathsf{X}} \chi\mathrm{d}\tilde{N}_{\mathbf{X}}^{\overrightarrow{q}}\right] \leqslant \mathrm{KL}(\mathbb{P}|\overrightarrow{R}) + \int_{[0,T_f]\times\mathsf{X}} \varrho(\chi)\mathrm{d}\bar{\mathrm{n}}^{\overrightarrow{q}} . \tag{71}$$

For any function $\phi \in S_\varrho$, taking $\chi := \phi/\|\phi\|_\varrho$ in (71) implies that for any $\phi \in S_\varrho$,

$$\mathbb{E}_{\mathbb{P}}\left[\int_{[0,T_f]\times\mathsf{X}} \phi\mathrm{d}\tilde{N}_{\mathbf{X}}^{\overrightarrow{q}}\right] \leqslant [\mathrm{KL}(\mathbb{P}|\overrightarrow{R}) + 1]\|\phi\|_\varrho . \tag{72}$$

Consider now the sub-space $\mathcal{H} \subset S_\varrho$,

$$\mathcal{H} := \left\{\phi : \mathbb{D}_{T_f} \times [0,T_f] \times \mathsf{X} \to \mathbb{R} \text{ predictable and bounded s.t. } \mathbb{E}_{\mathbb{P}}\left[\int_{[0,T_f]\times\mathsf{X}} \varrho(b|\phi|)\mathrm{d}\bar{\mathrm{n}}^{\overrightarrow{q}}\right] < \infty, \text{ for any } b \geqslant 0\right\} .$$

Since any $\phi \in \mathcal{H}$ satisfies (63), Lemma F.13 entails (72) for all $\phi \in \mathcal{H}$, as $\mathrm{KL}(\mathbb{P}|\overrightarrow{R}) < \infty$. This implies the linear mapping $\phi \mapsto \mathbb{E}_{\mathbb{P}}\left[\int_{[0,T_f]\times\mathsf{X}} \phi\mathrm{d}\tilde{N}_{\mathbf{X}}^{\overrightarrow{q}}\right]$ is continuous on $\mathcal{H}$ equipped with the norm $\|\cdot\|_\varrho$. It is worth noting that $(S_\varrho, \|\cdot\|_\varrho)$ forms a Banach space (see Léonard, 2007, Proposition 1.18), and $\mathcal{H}$ is a linear subspace of $S_\varrho$ (see Léonard, 2007, Proposition 1.11), hence the Hahn-Banach extension theorem (Delatte (2022, Theorem 1)) ensures that the previously defined linear functional on $\mathcal{H}$ can be extended to the entire space $S_\varrho$ without loss of continuity since the Luxemburg norm $\|\cdot\|_\varrho$ is a convex function on $S_\varrho$, see Rao & Ren (1991, Section 2.2). Note that the convex conjugate of the Young function $\varrho(|a|)$ is $\varrho^*(|b|)$. Therefore, as showed in Rao & Ren (1991, Theorem 3.1.9), the dual space of $(S_\varrho, \|\cdot\|_\varrho)$ is isomorphic to the space

$$L_{\varrho^*} := \left\{K : \mathbb{D}_{T_f} \times [0,T_f] \times \mathsf{X} \to \mathbb{R} \text{ measurable s.t. } \mathbb{E}_{\mathbb{P}}\left[\int_{[0,T_f]\times\mathsf{X}} \varrho^*(|K|)\mathrm{d}\bar{\mathrm{n}}^{\overrightarrow{q}}\right] < \infty\right\} ,$$

that means there exists some $K \in L_{\varrho^*}$ such that

$$\mathbb{E}_{\mathbb{P}}\left[\int_{[0,T_f]\times\mathsf{X}} \phi\mathrm{d}\tilde{N}_{\mathbf{X}}^{\overrightarrow{q}}\right] = \mathbb{E}_{\mathbb{P}}\left[\int_{[0,T_f]\times\mathsf{X}} K\phi\mathrm{d}\bar{\mathrm{n}}^{\overrightarrow{q}}\right], \quad \text{for any } \phi \in \mathcal{H} . \tag{73}$$

We now prove the uniqueness and predictability of $K$. Introduce the predictable projection of $K \in L_{\varrho^*}$ as $K^{pr} := \mathbb{E}_{\mathbb{P}}(K|\mathbf{X}_{[0,t)})$, for $t \in [0,T_f]$. Since $\mathcal{B}$ is dense in $S_\varrho$, $\mathcal{H}$ is dense in the subspace of all the predictable processes in $S_\varrho$. Then, any two functions $K_1, K_2 \in L_{\varrho^*}$ satisfying (73) must share the same projection, *i.e.*, $K_1^{pr} = K_2^{pr}$. It follows that there exists a unique predictable process $K$ in the space

$$\mathcal{K}(P) := \left\{K : \mathbb{D}_{T_f} \times [0,T_f] \times \mathsf{X} \to \mathbb{R} \text{ predictable s.t. } \mathbb{E}_{\mathbb{P}}\left[\int_{[0,T_f]\times\mathsf{X}} \varrho^*(|K|)\mathrm{d}\bar{\mathrm{n}}^{\overrightarrow{q}}\right] < \infty\right\} ,$$

which satisfies (73). Moreover, for any function $\phi \in \mathcal{H}$,

$$\int_{[0,T_f] \times \mathsf{X}} \phi \mathrm{d}(\tilde{N}_{\mathbf{X}}^{\overrightarrow{q}} - K \bar{\mathrm{n}}^{\overrightarrow{q}}) = \int_{[0,T_f] \times \mathsf{X}} \phi \mathrm{d}(N_{\mathbf{X}} - \bar{\mathrm{n}}^{\overrightarrow{q}} - K \bar{\mathrm{n}}^{\overrightarrow{q}})$$

$$= \int_{[0,T_f] \times \mathsf{X}} \phi \mathrm{d}(N_{\mathbf{X}} - (K+1) \bar{\mathrm{n}}^{\overrightarrow{q}})$$

$$= \int_{[0,T_f] \times \mathsf{X}} \phi \mathrm{d}(N_{\mathbf{X}} - U \bar{\mathrm{n}}^{\overrightarrow{q}}) \,,$$

with $U = K + 1$, and the equation (73) is thus equivalent to

$$\mathbb{E}_{\mathbb{P}} \left[ \int_{[0,T_f] \times \mathsf{X}} \phi \mathrm{d}(N_{\mathbf{X}} - U \bar{\mathrm{n}}^{\overrightarrow{q}}) \right] = 0, \quad \text{for any } \phi \in \mathcal{H} \,. \tag{74}$$

Thus, $U \bar{\mathrm{n}}^{\overrightarrow{q}}$ is a positive measure and $U$ is non-negative. Furthermore, we can argue analogously to obtain equation (74) on the interval $[s,t]$ for $0 \leqslant s \leqslant t \leqslant T_f$ then choose $\phi_s(\omega, x) = f(x) - f(\omega_{s-})$ to deduce

$$\mathbb{E}_{\mathbb{P}} \left[ f(\mathbf{X}_t) - f(\mathbf{X}_s) | \mathcal{F}_s \right] = \mathbb{E}_{\mathbb{P}} \left[ \int_{[s,t] \times \mathsf{X}} (f(x) - f(\mathbf{X}_{z-}))(U \bar{\mathrm{n}}^{\overrightarrow{q}})(\mathrm{d}z \mathrm{d}x) \middle| \mathcal{F}_s \right], \quad \text{for any } f \in \mathbb{F}(\mathsf{X}) \,. \tag{75}$$

Define the random generator $U \overrightarrow{q}$ on $\mathbb{D}_{T_f} \times [0, T_f] \times \mathsf{X}^2$ by $(U \overrightarrow{q})(\omega, t, x, y) := U(\omega, t, y) \overrightarrow{q}(x, y)$ for $y \neq x$ and use the convention

$$(U \overrightarrow{q})(\omega, t, x, x) := U(\omega, t, x) \overrightarrow{q}(x, x) = - \sum_{y \neq x} U(\omega, t, y) \overrightarrow{q}(x, y) \,.$$

Then $U \overrightarrow{q}$ forms a generator and (75) rewrites

$$\mathbb{E}_{\mathbb{P}} \left[ f(\mathbf{X}_t) - f(\mathbf{X}_s) | \mathcal{F}_s \right] = \mathbb{E}_{\mathbb{P}} \left[ \int_{[s,t]} (U \overrightarrow{q})(z) f(\mathbf{X}_{z-}) \mathrm{d}z \middle| \mathcal{F}_s \right], \quad \text{for any } f \in \mathbb{F}(\mathsf{X}) \,.$$

As a result, we conclude that $\mathbb{P} \in \mathrm{MP}(U \overrightarrow{q})$. We now show the formulation of the Radon-Nikodym density $\mathrm{d}\mathbb{P}/\mathrm{d}\overrightarrow{R}$. When $\mathbb{P} \sim \overrightarrow{R}$, define the stopping time $\tau_j^k$ as

$$\tau_j^k := \inf \left\{ t \in [0, T_f]; \int_{[0,t] \times \mathsf{X}} \varrho(\log U_s(\omega, x) \bar{\mathrm{n}}^{\overrightarrow{q}}(\mathrm{d}x \mathrm{d}s)) \geqslant k \text{ or } \log U_t(\omega, \omega_t) \notin [-j, k] \right\} \,,$$

which coincides with the stopping time $\sigma_j^k$ when $\chi = \log U$. Denote $U^{\sigma_j^k} := \mathbb{1}_{[0,\sigma_j^k]} U$ and for simplicity, we write $U = U^{\sigma_j^k}$. By conditioning w.r.t. $\mathbf{X}_0$, we can assume without loss of generality that $\overrightarrow{R}_0 = \mathbb{P}_0$, *i.e.*, $\frac{\mathrm{d}\mathbb{P}_0}{\mathrm{d}\overrightarrow{R}_0}(\mathbf{X}_0) = 1$.

Applying Lemma F.14 for $\mathbb{P} \in \mathrm{MP}(U \overrightarrow{q})$ and $\chi = -\log U$, we have

$$\mathbb{Q}^{\tau_j^k} := Z_{T_f}^{-\log U, U \overrightarrow{q}} \mathbb{P}^{\tau_j^k} \in \mathrm{MP}(\mathbb{1}_{[0,\tau_j^k]} \mathrm{e}^{-\log U} U \overrightarrow{q}) = \mathrm{MP}(\mathbb{1}_{[0,\tau_j^k]} \overrightarrow{q}) \,. \tag{76}$$

Furthermore, $\mathbb{Q}_0^{\tau_j^k} = \mathbb{P}_0^{\tau_j^k} = \overrightarrow{R}_0^{\tau_j^k} = \gamma^d$, where $\gamma^d$ is the invariant distribution of $(\mathbf{X}_t)_{t \in [0, T_f]}$, which combined with the equation (76) imply $\mathbb{Q}^{\tau_j^k}$ is the invariant path measure, *i.e.*,

$$\mathbb{Q}^{\tau_j^k} = \overrightarrow{R}^{\tau_j^k} \,.$$

Now, applying first Lemma F.14 for $\overrightarrow{R} \in \mathrm{MP}(\overrightarrow{q})$ and $\chi = \log U$ yields

$$\tilde{\mathbb{P}}^{\tau_j^k} := Z_{T_f}^{\log U, \overrightarrow{q}} \overrightarrow{R}^{\tau_j^k} \in \mathrm{MP}(\mathbb{1}_{[0,\tau_j^k]} \mathrm{e}^{\log U} \overrightarrow{q}) = \mathrm{MP}(\mathbb{1}_{[0,\tau_j^k]} U \overrightarrow{q}) \ .$$

Secondly, applying Lemma F.14 with $\tilde{\mathbb{P}}^{\tau_j^k} \in \mathrm{MP}(\mathbb{1}_{[0,\tau_j^k]} U \overrightarrow{q})$ and $\chi = -\log U$ implies

$$\tilde{\mathbb{Q}}^{\tau_j^k} := Z_{T_f}^{-\log U, U\overrightarrow{q}} \tilde{\mathbb{P}}^{\tau_j^k} \in \mathrm{MP}(\mathbb{1}_{[0,\tau_j^k]} \mathrm{e}^{-\log U} U \overrightarrow{q}) = \mathrm{MP}(\mathbb{1}_{[0,\tau_j^k]} \overrightarrow{q}) \ .$$

Argue as before, the previous equation together with the initial condition $\tilde{\mathbb{Q}}_0^{\tau_j^k} = \tilde{\mathbb{P}}_0^{\tau_j^k} = \overrightarrow{R}_0^{\tau_j^k} = \gamma^d$ yield that $\tilde{\mathbb{Q}}^{\tau_j^k} = \overrightarrow{R}^{\tau_j^k}$. Combining it with $\mathbb{Q}^{\tau_j^k} = \overrightarrow{R}^{\tau_j^k}$ implies

$$\mathbb{Q}^{\tau_j^k} = \tilde{\mathbb{Q}}^{\tau_j^k} \ ,$$

which means that

$$Z_{T_f}^{-\log U, U\overrightarrow{q}} \mathbb{P}^{\tau_j^k} = Z_{T_f}^{-\log U, U\overrightarrow{q}} \tilde{\mathbb{P}}^{\tau_j^k} \ . \tag{77}$$

Now observe that $Z_{T_f}^{-\log U, U\overrightarrow{q}} > 0$, therefore, equation (77) implies $\mathbb{P}^{\tau_j^k} = \tilde{\mathbb{P}}^{\tau_j^k}$. Hence $\mathbb{P}^{\tau_j^k} = Z_{T_f}^{\log U, \overrightarrow{q}} \overrightarrow{R}^{\tau_j^k}$, i.e.,

$$\mathbb{1}_{[0,\tau_j^k \wedge T_f]} \frac{d\mathbb{P}}{d\overrightarrow{R}}((\mathbf{X}_t)_{t\in[0,T_f]}) = \mathbb{1}_{[0,\tau_j^k \wedge T_f]} \frac{d\mathbb{P}_0}{d\overrightarrow{R}_0}(\mathbf{X}_0)$$

$$\exp\left(\int_{[0,\tau_j^k \wedge T_f]\times\mathsf{X}} (\mathbb{1}_{[0,\tau_j^k \wedge T_f]} \log U) d\tilde{N}_{\mathbf{X}}^{\overrightarrow{q}} - \int_{[0,\tau_j^k \wedge T_f]\times\mathsf{X}} \varrho(\log U) d\bar{\mathrm{n}}^{\overrightarrow{q}}\right) \ .$$

Letting $k$ and $j$ tend to infinity, since $\tau := \lim_{k,j\to\infty} \tau_j^k = \infty$, we get

$$\frac{d\mathbb{P}}{d\overrightarrow{R}}((\mathbf{X}_t)_{t\in[0,T_f]}) = \frac{d\mathbb{P}_0}{d\overrightarrow{R}_0}(\mathbf{X}_0) \exp\left(\int_{[0,T_f]\times\mathsf{X}} \log U d\tilde{N}_{\mathbf{X}}^{\overrightarrow{q}} - \int_{[0,T_f]\times\mathsf{X}} \varrho(\log U) d\bar{\mathrm{n}}^{\overrightarrow{q}}\right) \ .$$

We now extend the result above to the case when $\mathbb{P}$ might not be equivalent to $\overrightarrow{R}$. The idea is to approximate $\mathbb{P}$ by a sequence $(\mathbb{P}_n)$, which satisfies $\mathbb{P}_n \sim \overrightarrow{R}$ for all $n \geqslant 1$. Denoting

$$\mathbb{P}_n = \left(1 - \frac{1}{n}\right)\mathbb{P} + \frac{\overrightarrow{R}}{n} \quad \text{for } n \geqslant 1 \ , \tag{78}$$

we have $\mathbb{P}_n \sim \overrightarrow{R}$ and $\lim_{n\to\infty} \mathrm{KL}(\mathbb{P}|\mathbb{P}_n) = 0$. For simplicity, we write $\chi = \log U$ and $\chi^n = \log U^n$, which are well-defined $\mathbb{P}$-a.s. From the variational representation given in (70) and using $\mathbb{P} \in \mathrm{MP}(U\overrightarrow{q})$ combined with Lemma F.13, we obtain

$$\mathrm{KL}(\mathbb{P}|\mathbb{P}_n) \geqslant \mathbb{E}_{\mathbb{P}}\left[\int_{[0,T_f]\times\mathsf{X}} (\chi - \chi^n) d\tilde{N}_{\mathbf{X}}^{U^n\overrightarrow{q}} - \int_{[0,T_f]\times\mathsf{X}} \varrho(\chi - \chi^n) d(U^n \bar{\mathrm{n}}^{\overrightarrow{q}})\right] \ .$$

By definition, we have

$$\tilde{N}_{\mathbf{X}}^{U^n\overrightarrow{q}} = N_{\mathbf{X}} - U^n \bar{\mathrm{n}}^{\overrightarrow{q}} = N_{\mathbf{X}} - U\bar{\mathrm{n}}^{\overrightarrow{q}} + (U - U^n)\bar{\mathrm{n}}^{\overrightarrow{q}} = \tilde{N}_{\mathbf{X}}^{U\overrightarrow{q}} + (U - U^n)\bar{\mathrm{n}}^{\overrightarrow{q}} \ ,$$

which yields

$$
\mathrm{KL}(\mathbb{P}|\mathbb{P}_n) \geq \mathbb{E}_{\mathbb{P}}\left[\int_{[0,T_f]\times \mathsf{X}}(\chi - \chi^n)\mathrm{d}(\tilde{N}_{\mathbf{X}}^{U\overrightarrow{q}} + \bar{\mathrm{n}}^{\overrightarrow{q}}(U - U^n)) - \int_{[0,T_f]\times \mathsf{X}}\left(\frac{U}{U^n} - \log\frac{U}{U^n} - 1\right)U^n\mathrm{d}\bar{\mathrm{n}}^{\overrightarrow{q}}\right]
$$

$$
= \mathbb{E}_{\mathbb{P}}\left[\int_{[0,T_f]\times \mathsf{X}}(\chi - \chi^n)\mathrm{d}\tilde{N}_{\mathbf{X}}^{U\overrightarrow{q}} + \int_{[0,T_f]\times \mathsf{X}}U\log\frac{U}{U^n}\mathrm{d}\bar{\mathrm{n}}^{\overrightarrow{q}} - \int_{[0,T_f]\times \mathsf{X}}\left(\frac{U}{U^n} - 1\right)U^n\mathrm{d}\bar{\mathrm{n}}^{\overrightarrow{q}}\right] .
$$

Since $\mathbb{P} \in \mathrm{MP}(U\overrightarrow{q})$, we deduce that the stochastic integral $\int_{[0,T_f]\times \mathsf{X}}(\chi - \chi^n)\mathrm{d}\tilde{N}_{\mathbf{X}}^{U\overrightarrow{q}}$ is a local $\mathbb{P}$-martingale. Therefore,

$$
\mathrm{KL}(\mathbb{P}|\mathbb{P}_n) \geq \mathbb{E}_{\mathbb{P}}\left[\int_{[0,T_f]\times \mathsf{X}}\left(U^n - U - U\log\frac{U^n}{U}\right)\mathrm{d}\bar{\mathrm{n}}^{\overrightarrow{q}}\right]
$$

$$
= \mathbb{E}_{\mathbb{P}}\left[\int_{[0,T_f]\times \mathsf{X}}\left(\frac{U^n}{U} - \log\frac{U^n}{U} - 1\right)U\mathrm{d}\bar{\mathrm{n}}^{\overrightarrow{q}}\right]
$$

$$
= \mathbb{E}_{\mathbb{P}}\left[\int_{[0,T_f]\times \mathsf{X}}\varrho(\chi^n - \chi)\mathrm{d}(U\bar{\mathrm{n}}^{\overrightarrow{q}})\right] .
$$

Since $\lim_{n\to\infty}\mathrm{KL}(\mathbb{P}|\mathbb{P}_n) = 0$, we obtain

$$
\lim_{n\to\infty}\mathbb{E}_{\mathbb{P}}\left[\int_{[0,T_f]\times \mathsf{X}}\varrho(\chi^n - \chi)\mathrm{d}(U\bar{\mathrm{n}}^{\overrightarrow{q}})\right] = 0 . \tag{79}
$$

On the other hand, the fact that $\mathbb{P}_n \sim \overrightarrow{R}$ yields

$$
\frac{\mathrm{d}\mathbb{P}_n}{\mathrm{d}\overrightarrow{R}}((\mathbf{X}_t)_{t\in[0,T_f]}) = \frac{\mathrm{d}\mathbb{P}_{n,0}}{\mathrm{d}\overrightarrow{R}_0}(\mathbf{X}_0)\exp\left(\int_{[0,T_f]\times \mathsf{X}}\chi^n\mathrm{d}\tilde{N}_{\mathbf{X}}^{\overrightarrow{q}} - \int_{[0,T_f]\times \mathsf{X}}\varrho(\chi^n)\mathrm{d}\bar{\mathrm{n}}^{\overrightarrow{q}}\right) . \tag{80}
$$

To obtain the desired expression for the Radon–Nikodym density $\frac{\mathrm{d}\mathbb{P}}{\mathrm{d}\overrightarrow{R}}$, we represent it as

$$
\frac{\mathrm{d}\mathbb{P}}{\mathrm{d}\overrightarrow{R}}((\mathbf{X}_t)_{t\in[0,T_f]}) = \left(\frac{\mathrm{d}\mathbb{P}}{\mathrm{d}\mathbb{P}_n}\cdot\frac{\mathrm{d}\mathbb{P}_n}{\mathrm{d}\overrightarrow{R}}\right)((\mathbf{X}_t)_{t\in[0,T_f]})
$$

$$
\overset{(80)}{=} \left(\frac{\mathrm{d}\mathbb{P}}{\mathrm{d}\mathbb{P}_n}\cdot\frac{\mathrm{d}\mathbb{P}_{n,0}}{\mathrm{d}\overrightarrow{R}_0}\right)(\mathbf{X}_0)\exp\left(\int_{[0,T_f]\times \mathsf{X}}\chi^n\mathrm{d}\tilde{N}_{\mathbf{X}}^{\overrightarrow{q}} - \int_{[0,T_f]\times \mathsf{X}}\varrho(\chi^n)\mathrm{d}\bar{\mathrm{n}}^{\overrightarrow{q}}\right)
$$

$$
= \left(\frac{\mathrm{d}\mathbb{P}}{\mathrm{d}\mathbb{P}_n}\cdot\frac{\mathrm{d}\mathbb{P}_{n,0}}{\mathrm{d}\mathbb{P}_0}\right)(\mathbf{X}_0)\frac{\mathrm{d}\mathbb{P}_0}{\mathrm{d}\overrightarrow{R}_0}(\mathbf{X}_0)\exp\left(\int_{[0,T_f]\times \mathsf{X}}\chi\mathrm{d}\tilde{N}_{\mathbf{X}}^{\overrightarrow{q}} - \int_{[0,T_f]\times \mathsf{X}}\varrho(\chi)\mathrm{d}\bar{\mathrm{n}}^{\overrightarrow{q}}\right)
$$

$$
\exp\left(\int_{[0,T_f]\times \mathsf{X}}(\chi^n - \chi)\mathrm{d}\tilde{N}_{\mathbf{X}}^{\overrightarrow{q}} - \int_{[0,T_f]\times \mathsf{X}}(\varrho(\chi^n) - \varrho(\chi))\mathrm{d}\bar{\mathrm{n}}^{\overrightarrow{q}}\right) , \mathbb{P}\text{-a.s.} \tag{81}
$$

The last part can be rewritten as follows using equation (73),

$$\exp\left(\int_{[0,T_f]\times\mathsf{X}}(\chi^n-\chi)\mathrm{d}\tilde{N}_\mathbf{X}^{U\overrightarrow{q}}+\int_{[0,T_f]\times\mathsf{X}}(\chi^n-\chi)(\mathrm{e}^\chi-1)\mathrm{d}\bar{\mathrm{n}}^{\overrightarrow{q}}-\int_{[0,T_f]\times\mathsf{X}}(\mathrm{e}^{\chi^n}-\chi^n-\mathrm{e}^\chi+\chi)\mathrm{d}\bar{\mathrm{n}}^{\overrightarrow{q}}\right)$$

$$=\exp\left(\int_{[0,T_f]\times\mathsf{X}}(\chi^n-\chi)\mathrm{d}\tilde{N}_\mathbf{X}^{U\overrightarrow{q}}-\int_{[0,T_f]\times\mathsf{X}}\varrho(\chi-\chi^n)\mathrm{d}(U\bar{\mathrm{n}}^{\overrightarrow{q}})\right)$$

We first handle the second integral above using (79) and the fact that $\varrho$ is a non-negative function,

$$\mathbb{E}_\mathbb{P}\left[\left|\int_{[0,T_f]\times\mathsf{X}}\varrho(\chi-\chi^n)\mathrm{d}(U\bar{\mathrm{n}}^{\overrightarrow{q}})\right|\right]\xrightarrow{n\to\infty}0\ .$$

This together with Markov's inequality lead to

$$\int_{[0,T_f]\times\mathsf{X}}\varrho(\chi-\chi^n)\mathrm{d}(U\bar{\mathrm{n}}^{\overrightarrow{q}})\xrightarrow{\mathbb{P}}0\ ,\tag{82}$$

and therefore, from Méliot (2025, Proposition 1.4), there is a subsequence $\chi^{n_k}$ such that

$$\int_{[0,T_f]\times\mathsf{X}}\varrho(\chi-\chi^{n_k})\mathrm{d}(U\bar{\mathrm{n}}^{\overrightarrow{q}})\xrightarrow{n\to\infty}0\ ,\quad\mathbb{P}\text{-a.s.}\tag{83}$$

Furthermore, recall that $\varrho(a)=\mathrm{e}^a-a-1\geqslant a^2/2$, hence (79) can be used to control the stochastic integral w.r.t. the $\mathbb{P}$-martingale $\tilde{N}_\mathbf{X}^{U\overrightarrow{q}}$ as follows

$$\mathbb{E}_\mathbb{P}\left[\left(\int_{[0,T_f]\times\mathsf{X}}(\chi^n-\chi)\mathrm{d}\tilde{N}_\mathbf{X}^{U\overrightarrow{q}}\right)^2\right]\overset{Corollary\ F.10}{=}\mathbb{E}_\mathbb{P}\left[\int_{[0,T_f]\times\mathsf{X}}(\chi^n-\chi)^2\mathrm{d}(U\bar{\mathrm{n}}^{\overrightarrow{q}})\right]$$

$$\leqslant 2\mathbb{E}_\mathbb{P}\left[\int_{[0,T_f]\times\mathsf{X}}\varrho(\chi-\chi^n)\mathrm{d}(U\bar{\mathrm{n}}^{\overrightarrow{q}})\right]\xrightarrow{n\to\infty}0\ .$$

This, along with Markov's inequality, results in

$$\int_{[0,T_f]\times\mathsf{X}}(\chi^n-\chi)\mathrm{d}\tilde{N}_\mathbf{X}^{U\overrightarrow{q}}\xrightarrow{\mathbb{P}}0\ .$$

Combining this with (82) implies that

$$\int_{[0,T_f]\times\mathsf{X}}(\chi^n-\chi)\mathrm{d}\tilde{N}_\mathbf{X}^{U\overrightarrow{q}}-\int_{[0,T_f]\times\mathsf{X}}\varrho(\chi-\chi^n)\mathrm{d}(U\bar{\mathrm{n}}^{\overrightarrow{q}})\xrightarrow{\mathbb{P}}0\ .$$

As a consequence, Méliot (2025, Proposition 1.4) asserts that there is a subsequence $(\chi^{n_k})$ such that

$$\left[\int_{[0,T_f]\times\mathsf{X}}(\chi^{n_k}-\chi)\mathrm{d}\tilde{N}_\mathbf{X}^{U\overrightarrow{q}}-\int_{[0,T_f]\times\mathsf{X}}\varrho(\chi-\chi^{n_k})\mathrm{d}(U\bar{\mathrm{n}}^{\overrightarrow{q}})\right]\xrightarrow{k\to\infty}0\ ,\quad\mathbb{P}\text{-a.s.}$$

It helps interpreting (81) as

$$\frac{\mathrm{d}\mathbb{P}}{\mathrm{d}\overrightarrow{R}}((\mathbf{X}_t)_{t\in[0,T_f]}) = \left(\frac{\mathrm{d}\mathbb{P}}{\mathrm{d}\mathbb{P}_{n_k}}\cdot\frac{\mathrm{d}\mathbb{P}_{n_k,0}}{\mathrm{d}\mathbb{P}_0}\right)(\mathbf{X}_0)\frac{\mathrm{d}\mathbb{P}_0}{\mathrm{d}\overrightarrow{R}_0}(\mathbf{X}_0)\exp\left(\int_{[0,T_f]\times\mathsf{X}}\chi\mathrm{d}\tilde{N}_{\mathbf{X}}^{\overrightarrow{q}} - \int_{[0,T_f]\times\mathsf{X}}\varrho(\chi)\mathrm{d}\bar{\mathrm{n}}^{\overrightarrow{q}}\right)$$

$$\exp\left(\int_{[0,T_f]\times\mathsf{X}}(\chi^{n_k}-\chi)\mathrm{d}\tilde{N}_{\mathbf{X}}^{\overrightarrow{q}} - \int_{[0,T_f]\times\mathsf{X}}(\varrho(\chi^{n_k})-\varrho(\chi))\mathrm{d}\bar{\mathrm{n}}^{\overrightarrow{q}}\right)$$

$$\xrightarrow[(78)]{k\to\infty}\frac{\mathrm{d}\mathbb{P}_0}{\mathrm{d}\overrightarrow{R}_0}(\mathbf{X}_0)\exp\left(\int_{[0,T_f]\times\mathsf{X}}\chi\mathrm{d}\tilde{N}_{\mathbf{X}}^{\overrightarrow{q}} - \int_{[0,T_f]\times\mathsf{X}}\varrho(\chi)\mathrm{d}\bar{\mathrm{n}}^{\overrightarrow{q}}\right) , \quad \mathbb{P}\text{-a.s.}$$

Replacing $\chi = \log U$, we arrive at our desired claim

$$\frac{\mathrm{d}\mathbb{P}}{\mathrm{d}\overrightarrow{R}}((\mathbf{X}_t)_{t\in[0,T_f]}) = \frac{\mathrm{d}\mathbb{P}_0}{\mathrm{d}\overrightarrow{R}_0}(\mathbf{X}_0)\exp\left(\int_{[0,T_f]\times\mathsf{X}}\log U\mathrm{d}\tilde{N}_{\mathbf{X}}^{\overrightarrow{q}} - \int_{[0,T_f]\times\mathsf{X}}\varrho(\log U)\mathrm{d}\bar{\mathrm{n}}^{\overrightarrow{q}}\right) , \quad \mathbb{P}\text{-a.s.}$$

Consequently, the KL divergence reads as

$$\mathrm{KL}(\mathbb{P}|\overrightarrow{R}) = \mathrm{KL}(\mathbb{P}_0|\overrightarrow{R}_0) + \mathbb{E}_{\mathbb{P}}\left[\int_{[0,T_f]\times\mathsf{X}}\log U\mathrm{d}\tilde{N}_{\mathbf{X}}^{\overrightarrow{q}} - \int_{[0,T_f]\times\mathsf{X}}\varrho(\log U)\mathrm{d}\bar{\mathrm{n}}^{\overrightarrow{q}}\right] .$$

Applying (74) to the function $\phi = \log U$, we get

$$\mathrm{KL}(\mathbb{P}|\overrightarrow{R}) = \mathrm{KL}(\mathbb{P}_0|\overrightarrow{R}_0) + \mathbb{E}_{\mathbb{P}}\left[\int_{[0,T_f]\times\mathsf{X}}(U-1)\log U\mathrm{d}\bar{\mathrm{n}}^{\overrightarrow{q}} - \int_{[0,T_f]\times\mathsf{X}}\varrho(\log U)d\bar{\mathrm{n}}^{\overrightarrow{q}}\right]$$

$$= \mathrm{KL}(\mathbb{P}_0|\overrightarrow{R}_0) + \mathbb{E}_{\mathbb{P}}\left[\int_{[0,T_f]\times\mathsf{X}}[(U-1)\log U - U + \log U + 1]\mathrm{d}\bar{\mathrm{n}}^{\overrightarrow{q}}\right]$$

$$= \mathrm{KL}(\mathbb{P}_0|\overrightarrow{R}_0) + \mathbb{E}_{\mathbb{P}}\left[\int_{[0,T_f]\times\mathsf{X}}h(U(t,x))\bar{\mathrm{n}}^{\overrightarrow{q}}(\mathrm{d}t\mathrm{d}x)\right] ,$$

with $h(a) := \varrho^*(a-1) = a\log a - a + 1$ for $a > 0$. The proof of Theorem F.12 is then finished. $\qquad\square$

In fact, we can simplify the KL expression above by replacing $\bar{\mathrm{n}}^{\overrightarrow{q}}(\mathrm{d}t\mathrm{d}x) = \sum_{y\in\mathsf{X}}\mathbb{1}_{\mathbf{X}_{t-}\neq y}\overrightarrow{q}(\mathbf{X}_{t-},y)\delta_y(\mathrm{d}x)\mathrm{d}t$ and $\overrightarrow{q}$ by the formula given in (59) to arrive at

$$\mathrm{KL}(\mathbb{P}|\overrightarrow{R}) = \mathrm{KL}(\mathbb{P}_0|\overrightarrow{R}_0) + \mathbb{E}_{\mathbb{P}}\int_{[0,T_f]\times\mathsf{X}}h(U(t,x))\sum_{y\in\mathsf{X}}\mathbb{1}_{\mathbf{X}_{t-}\neq y}\overrightarrow{q}(\mathbf{X}_{t-},y)\delta_y(\mathrm{d}x)\mathrm{d}t$$

$$= \mathrm{KL}(\mathbb{P}_0|\overrightarrow{R}_0) + \lambda\mathbb{E}_{\mathbb{P}}\int_{[0,T_f]}\sum_{\ell=1}^{d}h(U(t,\varphi^{(\ell)}(\mathbf{X}_{t-})))\mathrm{d}t .$$

### F.2.2. INTERPRETING THE TIME-REVERSED DYNAMIC AS A CONTROL-DRIVEN PROCESS

Let $\overrightarrow{R} \in \mathrm{MP}(\overrightarrow{q})$ be the stationary measure on the path space $\mathbb{D}_{T_f}$ introduced in the previous section. Then, the process $\overrightarrow{R}$ is showed to be reversible, meaning $\overleftarrow{R} = \overrightarrow{R}$, and corresponds to the invariant measure $\gamma^d = \mathrm{Uniform}(\mathsf{X})$. Let $\overrightarrow{\mathbb{P}}^{\mu^\star} = \mathrm{Law}((\overrightarrow{X}_t)_{t\in[0,T_f]})$ represent the forward probability measure on the interval $[0,T_f]$ starting from $\mu^\star$ and governed by the forward generator $\overrightarrow{q}$ given in (59). We denote the corresponding time-reversed probability measure ending at $\mu^\star$ by $\overleftarrow{\mathbb{P}}^{\mu^\star} = \mathrm{Law}((\overleftarrow{X}_t)_{t\in[0,T_f]})$. Let $\mu_t$ be the marginal density of the forward dynamic at time $t \in [0,T_f]$ and denote by $\tilde{\mu}_t := \mu_t/\gamma^d$ the corresponding relative density.

**Proposition F.15.** *The time reversal process $\overleftarrow{\mathbb{P}}^{\mu^\star}$ solves the Martingale Problem* $\mathrm{MP}(u\overrightarrow{q})$ *with the control $u$ given by: for $(t,x) \in [0,T_f] \times \mathsf{X}$,*

$$
u_t(x,y) = \begin{cases}
\tilde{\mu}_{T_f-t}(\varphi^{(\ell)}(x))/\tilde{\mu}_{T_f-t}(x) \,, & \text{if } y = \varphi^{(\ell)}(x) \text{ for some } \ell = 1,\dots,d \,, \\
\sum_{\ell=1}^{d} u_t(x,\varphi^{(\ell)}(x))/d \,, & \text{if } y = x \,, \\
1 \,, & \text{otherwise} \,.
\end{cases}
\tag{84}
$$

***Proof of Proposition F.15.*** Recall that the generator of the backward process $\overleftarrow{q}$ satisfies the following equation for $t \in [0,T_f]$ and $x \neq y$,

$$
\overleftarrow{q}_t(x,y) = \frac{\mu_{T_f-t}(y)}{\mu_{T_f-t}(x)} \overrightarrow{q}(y,x) \,.
$$

Using the fact that the stationary distribution $\gamma^d$ satisfies the following balance equation for $x,y \in \mathsf{X}$, $x \neq y$,

$$
\gamma^d(x)\overrightarrow{q}(x,y) = \gamma^d(y)\overrightarrow{q}(y,x) \,,
$$

we can express the backward generator as the perturbation of the forward one as follows

$$
\overleftarrow{q}_t(x,y) = \frac{\mu_{T_f-t}(y)}{\mu_{T_f-t}(x)} \overrightarrow{q}(y,x) = \frac{\mu_{T_f-t}(y)\gamma^d(x)}{\mu_{T_f-t}(x)\gamma^d(y)} \overrightarrow{q}(x,y) = \frac{\tilde{\mu}_{T_f-t}(y)}{\tilde{\mu}_{T_f-t}(x)} \overrightarrow{q}(x,y) \,.
$$

Note that $\overrightarrow{q}(x,y) = 0$ for $y \notin \{x, \varphi^{(\ell)}(x);\ \ell = 1,\dots,d\}$, thus we can define the control $u$ as: for $(t,x) \in [0,T_f] \times \mathsf{X}$,

$$
u_t(x,\varphi^{(\ell)}(x)) = \frac{\tilde{\mu}_{T_f-t}(\varphi^{(\ell)}(x))}{\tilde{\mu}_{T_f-t}(x)} \quad \text{for } \ell = 1,\dots,d \,, \quad \text{and} \quad u_t(x,y) = 1 \quad \text{for } y \notin \{x, \varphi^{(\ell)}(x);\ \ell = 1,\dots,d\} \,,
$$

to obtain the relation

$$
\overleftarrow{q}_t(x,y) = u_t(x,y)\overrightarrow{q}(x,y) \,, \quad \text{for } (t,x,y) \in [0,T_f] \times \mathsf{X}^2 \text{ and } y \neq x \,.
$$

Furthermore, under the convention

$$
u_t(x,x) = \frac{-\sum_{y \neq x} u_t(x,y)\overrightarrow{q}(x,y)}{\overrightarrow{q}(x,x)} = \frac{1}{d}\sum_{\ell=1}^{d} u_t(x,\varphi^{(\ell)}(x)) \,, \quad \text{for } (t,x) \in [0,T_f] \times \mathsf{X} \,,
$$

$u\overrightarrow{q}$ in fact forms a generator and satisfies $\overleftarrow{q} = u\overrightarrow{q}$, which implies that $\overleftarrow{\mathbb{P}}^{\mu^\star} \in \mathrm{MP}(u\overrightarrow{q})$ and we conclude the proof. $\square$

Proposition F.15 shows that the unique control $u$ associated with the backward dynamic $(\overleftarrow{X}_t)_{t \in [0,T_f]}$ in Girsanov's thereom F.12 is in fact Markovian. This enables expressing the Radon-Nikodym density $\mathrm{d}\overleftarrow{\mathbb{P}}^{\mu^\star}/\mathrm{d}\overrightarrow{R}$ as

$$
\frac{\mathrm{d}\overleftarrow{\mathbb{P}}^{\mu^\star}}{\mathrm{d}\overrightarrow{R}}((\overleftarrow{X}_t)_{t \in [0,T_f]}) = \frac{\mathrm{d}\overleftarrow{\mathbb{P}}_0^{\mu^\star}}{\mathrm{d}\overrightarrow{R}_0}(\overleftarrow{X}_0) \exp\left( \int_{[0,T_f]\times\mathsf{X}} \log u_t(\overleftarrow{X}_{t-},x)\mathrm{d}\tilde{N}_{\mathsf{X}}^{\overrightarrow{q}} - \int_{[0,T_f]\times\mathsf{X}} \varrho(\log u_t(\overleftarrow{X}_{t-},x))\mathrm{d}\bar{\mathrm{n}}^{\overrightarrow{q}} \right) \,,
$$

where $u$ is explicitly given in (84).

F.2.3. EVOLUTION OF THE CONTROL IN THE REVERSED-TIME SYSTEM

This section aims to characterize the control corresponding to the backward dynamics through the Hamilton–Jacobi–Bellman (HJB) equation. This characterization serves as the key ingredient for applying Itô's formula to analyze the evolution of the time-reversed process.

**Proposition F.16** (Hamilton–Jacobi–Bellman equation)**.** *The control $u$ given in* (84) *admits the following formula*

$$u_t(x,y) = e^{V(t,x)-V(t,y)} , \quad for \ (t,x,y) \in [0,T_f] \times \mathsf{X}^2 , \quad x \neq y ,$$

*where $V(t,x) = -\log \tilde{\mu}_{T_f-t}(x)$, which satisfies the following HJB equation*

$$\begin{cases} \partial_t V(t,x) = \lambda \sum_{\ell=1}^d [e^{V(t,x)-V(t,\varphi^{(\ell)}(x))} - 1] , \\ V(T_f,x) = g(x) = -\log \frac{d\mu^\star}{d\gamma^d}(x) , \end{cases} \quad for \ (t,x) \in [0,T_f) \times \mathsf{X} . \tag{85}$$

*Proof of Proposition F.16.* Denote $V(t,x) := -\log \tilde{\mu}_{T_f-t}(x)$, then the optimal control can be interpreted as

$$u_t(x,y) = \frac{\tilde{\mu}_{T_f-t}(y)}{\tilde{\mu}_{T_f-t}(x)} = \frac{e^{-V(t,y)}}{e^{-V(t,x)}} = e^{V(t,x)-V(t,y)} , \quad \text{for } (t,x,y) \in [0,T_f] \times \mathsf{X}^2 \text{ and } x \neq y .$$

In addition, the function $V$ fulfills the following equation: for $(t,x) \in [0,T) \times \mathsf{X}$,

$$\begin{aligned} \partial_t V(t,x) &= \frac{\partial_t \mu_{T_f-t}(x)}{\mu_{T_f-t}(x)} \\ &= \frac{\sum_{y\in\mathsf{X}} \mu_{T_f-t}(y) \overrightarrow{q}(y,x)}{\mu_{T_f-t}(x)} \quad \text{(by forward Kolmogorov equation)} \\ &= \frac{\sum_{y\in\mathsf{X}} \mu_{T_f-t}(y) \overrightarrow{q}(x,y)\gamma^d(x)/\gamma^d(y)}{\mu_{T_f-t}(x)} \quad \text{(by balance equation)} \\ &= \sum_{y\in\mathsf{X}} \frac{\tilde{\mu}_{T_f-t}(y)}{\tilde{\mu}_{T_f-t}(x)} \overrightarrow{q}(x,y) \\ &= \lambda \sum_{\ell=1}^d [e^{V(t,x)-V(t,\varphi^{(\ell)}(x))} - 1] . \end{aligned}$$

Moreover, $V$ also satisfies the final condition

$$V_{T_f}(x) = -\log \tilde{\mu}_0(x) = -\log \tilde{\mu}^\star(x) = g(x) , \quad \text{for } x \in \mathsf{X} ,$$

therefore $V$ solves the HJB equation (85) and thus concludes the proof of Proposition F.16. $\square$

To derive the convergence bound, it is essential to interpret the evolution of the control using the HJB equation first. In fact, the control above satisfies the following martingale and monotone property due to its characterization given in Proposition F.16.

**Proposition F.17.** *With all the notations above, $u_t(\overleftarrow{X}_t, \varphi^{(\ell)}(\overleftarrow{X}_t))$ is a $\overleftarrow{\mathbb{P}}^{\mu^\star}$-martingale for fixed $\ell = 1, \dots, d$. Consequently, $h(u_t(\overleftarrow{X}_t, \varphi^{(\ell)}(\overleftarrow{X}_t)))$ is a $\overleftarrow{\mathbb{P}}^{\mu^\star}$-submartingale and the monotonicity follows:*

$$\mathbb{E}_{\overleftarrow{\mathbb{P}}^{\mu^\star}}[h(u_s(\overleftarrow{X}_s, \varphi^{(\ell)}(\overleftarrow{X}_s)))] \leqslant \mathbb{E}_{\overleftarrow{\mathbb{P}}^{\mu^\star}}[h(u_t(\overleftarrow{X}_t, \varphi^{(\ell)}(\overleftarrow{X}_t)))] , \quad for \quad 0 \leqslant s \leqslant t \leqslant T_f .$$

*Proof of Proposition F.17.* Fix $t \in [0,T_f]$ and $\ell = 1, \dots, d$, applying Itô's formula on

$$f^\ell(t, \overleftarrow{X}_t) := u_t(\overleftarrow{X}_t, \varphi^{(\ell)}(\overleftarrow{X}_t)) = e^{V(t,\overleftarrow{X}_t)-V(t,\varphi^{(\ell)}(\overleftarrow{X}_t))} ,$$

and note that $\mathrm{Law}(\overleftarrow{X}_\cdot) = \overleftarrow{\mathbb{P}}^{\mu^\star}$ solves $\mathrm{MP}(u\overrightarrow{q})$ as well as $\overleftarrow{X}_t = \overleftarrow{X}_{t-}$ for Lebesgue almost all $t \in (0,T_f]$ (see Mozumder,

2009, Proposition 2.1), we obtain that the process

$$f^\ell(t, \overleftarrow{X}_t) - f^\ell(0, \overleftarrow{X}_0) - \int_0^t \left[\partial_s f^\ell(s, \overleftarrow{X}_s) + (u\overrightarrow{q})f_s^\ell(\overleftarrow{X}_s)\right] \mathrm{d}s$$

is a $\overleftarrow{\mathbb{P}}^{\mu^\star}$-martingale. Denote

$$b_s^\ell := \partial_s f^\ell(s, \overleftarrow{X}_s) + (u\overrightarrow{q})f_s^\ell(\overleftarrow{X}_s), \quad \text{for } s \in [0, t] .$$

We aim to prove that $b_s^\ell = 0$. Indeed, by the definition of $f^\ell$, $\overrightarrow{q}$ and the HJB equation (85), we get that

$$b_s^\ell = u_s(\overleftarrow{X}_s, \varphi^{(\ell)}(\overleftarrow{X}_s)) \left[\partial_s V(s, \overleftarrow{X}_s) - \partial_s V(s, \varphi^{(\ell)}(\overleftarrow{X}_s))\right]$$

$$+ \sum_{i=1}^d \left[u_s(\varphi^{(i)}(\overleftarrow{X}_s), \varphi^{(\ell)}(\varphi^{(i)}(\overleftarrow{X}_s))) - u_s(\overleftarrow{X}_s, \varphi^{(\ell)}(\overleftarrow{X}_s))\right] \lambda u_s(\overleftarrow{X}_s, \varphi^{(i)}(\overleftarrow{X}_s))$$

$$= \lambda u_s(\overleftarrow{X}_s, \varphi^{(\ell)}(\overleftarrow{X}_s)) \left[\sum_{i=1}^d u_s(\overleftarrow{X}_s, \varphi^{(i)}(\overleftarrow{X}_s)) - \sum_{i=1}^d u_s(\varphi^{(\ell)}(\overleftarrow{X}_s), \varphi^{(i)}(\varphi^{(\ell)}(\overleftarrow{X}_s)))\right]$$

$$+ \sum_{i=1}^d \left[u_s(\varphi^{(i)}(\overleftarrow{X}_s), \varphi^{(\ell)}(\varphi^{(i)}(\overleftarrow{X}_s))) - u_s(\overleftarrow{X}_s, \varphi^{(\ell)}(\overleftarrow{X}_s))\right] \lambda u_s(\overleftarrow{X}_s, \varphi^{(i)}(\overleftarrow{X}_s))$$

$$= \lambda \sum_{i=1}^d \left[u_s(\varphi^{(i)}(\overleftarrow{X}_s), \varphi^{(\ell)}(\varphi^{(i)}(\overleftarrow{X}_s))) u_s(\overleftarrow{X}_s, \varphi^{(i)}(\overleftarrow{X}_s))\right.$$

$$\left. - u_s(\varphi^{(\ell)}(\overleftarrow{X}_s), \varphi^{(i)}(\varphi^{(\ell)}(\overleftarrow{X}_s))) u_s(\overleftarrow{X}_s, \varphi^{(\ell)}(\overleftarrow{X}_s))\right] .$$

Using the identity $u_s(x, \varphi^{(i)}(x)) = \mathrm{e}^{V(s,x)-V(s,\varphi^{(i)}(x))}$ for $i = 1, 2, \ldots, d$ in Proposition F.22 yields

$$b_s^\ell = \lambda \sum_{i=1}^d \left[\mathrm{e}^{V(s,\varphi^{(i)}(\overleftarrow{X}_s))-V(s,\varphi^{(\ell)}(\varphi^{(i)}(\overleftarrow{X}_s)))+V(s,\overleftarrow{X}_s)-V(s,\varphi^{(i)}(\overleftarrow{X}_s))}\right.$$

$$\left. - \mathrm{e}^{V(s,\varphi^{(\ell)}(\overleftarrow{X}_s))-V(s,\varphi^{(i)}(\varphi^{(\ell)}(\overleftarrow{X}_s)))+V(s,\overleftarrow{X}_s)-V(s,\varphi^{(\ell)}(\overleftarrow{X}_s))}\right]$$

$$= \lambda \sum_{i=1}^d \left[\mathrm{e}^{-V(s,\varphi^{(\ell)}(\varphi^{(i)}(\overleftarrow{X}_s)))+V(s,\overleftarrow{X}_s)} - \mathrm{e}^{-V(s,\varphi^{(i)}(\varphi^{(\ell)}(\overleftarrow{X}_s)))+V(s,\overleftarrow{X}_s)}\right] = 0 ,$$

as $\varphi^{(\ell)}(\varphi^{(i)}(\overleftarrow{X}_s)) = \varphi^{(i)}(\varphi^{(\ell)}(\overleftarrow{X}_s))$ for any $\ell, i = 1, \ldots, d$. We thus conclude that $u_t(\overleftarrow{X}_t, \varphi^{(\ell)}(\overleftarrow{X}_t))$ is a $\overleftarrow{\mathbb{P}}^{\mu^\star}$-martingale. Furthermore, since $h$ is convex, it follows that $h(u_t(\overleftarrow{X}_t, \varphi^{(\ell)}(\overleftarrow{X}_t)))$ is a $\overleftarrow{\mathbb{P}}^{\mu^\star}$-submartingale, which implies the desired monotonicity for $\ell = 1, 2, \ldots, d$ and concludes the proof.

$\square$

### F.3. Connection between the transition matrix and canonical process point of view

As we see in previous sections, the time reversal process can be understood not only via the backward transition matrix but also via the process driven by the control. The transition matrix point of view provides an approximation of the score to simulate the backward process, which is very useful in practice. In parallel, the canonical process point of view gives us a better understanding of the evolution of the time reversal process, which allows us to show a theoretical guarantee on our algorithm. These two points of view in fact have a strong relation, which will be specified in the following Proposition.

**Proposition F.18.** *The control $u$ driving the backward process satisfies the following relation w.r.t. the score function*

*defined in* (17) *as*

$$u_t(x, \varphi^{(\ell)}(x)) = 1 - s_t^{\ell}(x) , \quad \text{with } \ell = 1, \ldots, d \text{ and } (t, x) \in [0, T_f) \times \mathsf{X} . \tag{86}$$

***Proof of Proposition F.18.*** The result follows directly from the definition of the score function $s$ in (17) and the form of the control $u$ in (84). This identity confirms the equivalence between the transition matrix viewpoint and the canonical process formulation. $\square$

### F.4. Optimal control perspective on the time-reversed process

This section presents an alternative perspective on deriving the HJB equation, viewing it through the lens of optimal control and leveraging the Dynamic Programming Principle. In particular, the predictable process $u$ defined in Theorem F.12 can be characterized not only via the backward generator but also as the solution to an optimal control problem involving the expression of relative entropy.

In the continuous case, Conforti et al. (2025) demonstrated that the time reversal process can be formulated as a solution to an optimal control problem. This characterization not only describes the dynamics of the process but also serves as a powerful framework for deriving the HJB equation, which can be obtained by leveraging Girsanov's theorem (Theorem F.12)

Let $\overrightarrow{R}$ be the stationary measure on the path space $\mathbb{D}_{T_f}$ introduced in the previous section. Then, the process $\overrightarrow{R}$ is reversible, meaning $\overleftarrow{R} = \overrightarrow{R}$, and corresponds to the invariant measure $\gamma^d = \mathrm{Uniform}(\mathsf{X})$. Let $\overrightarrow{\mathbb{P}}^{\mu^\star}$ represent the forward probability measure on the interval $[0, T_f]$ starting from $\mu^\star$ and governed by the forward generator $\overrightarrow{q}$ given in (59). We denote the corresponding backward probability measure ending at $\mu^\star$ by $\overleftarrow{\mathbb{P}}^{\mu^\star}$.

**Proposition F.19.** *The time reversal process $\overleftarrow{\mathbb{P}}^{\mu^\star}$ satisfies the following optimization problem*

$$\overleftarrow{\mathbb{P}}^{\mu^\star} = \underset{\mathbb{P} \in \mathcal{P}(\mathbb{D}_{T_f}):\ \mathrm{KL}(\mathbb{P}|\overrightarrow{R}) < \infty}{\arg\min} \left( \mathrm{KL}(\mathbb{P}|\overrightarrow{R}) + \int g \mathrm{d}\mathbb{P}_{T_f} \right), \quad \text{with } g = -\log \frac{\mathrm{d}\mu^\star}{\mathrm{d}\gamma^d} .$$

***Proof of Proposition F.19.*** For $\mathbb{P} = \mathrm{Law}((\mathbf{X}_t)_{t \in [0, T_f]}) \in \mathcal{P}(\mathbb{D}_{T_f})$ such that $\mathrm{KL}(\mathbb{P}|\overrightarrow{R}) < \infty$, the Donsker-Varadhan variational formulation of the KL implies that

$$\mathrm{KL}(\mathbb{P}|\overrightarrow{R}) = \underset{f \in L^1(\mathbb{P}) \text{ s.t. } \int \mathrm{e}^f \mathrm{d}\overrightarrow{R} < \infty}{\sup} \left( \int f \mathrm{d}\mathbb{P} - \log \int \mathrm{e}^f \mathrm{d}\overrightarrow{R} \right) .$$

Taking $f((\mathbf{X}_t)_{t \in [0, T_f]}) = -g(\mathbf{X}_{T_f})$ yields

$$\mathrm{KL}(\mathbb{P}|\overrightarrow{R}) \geqslant \int -g \mathrm{d}\mathbb{P}_{T_f} - \log \int \mathrm{e}^{-g} \mathrm{d}\overrightarrow{R}_{T_f} = -\int g \mathrm{d}\mathbb{P}_{T_f} - \log \int \mathrm{d}\mu^\star = -\int g \mathrm{d}\mathbb{P}_{T_f} ,$$

since $\int \mathrm{d}\mu^\star = 1$. On the other hand, since $\overleftarrow{\mathbb{P}}^{\mu^\star}$ is the backward process ended at $\mu^\star$ and $\overrightarrow{R}$ is a reversible path probability measure on $[0, T_f]$, *i.e.*, $\overrightarrow{R} = \overleftarrow{R}$, we have

$$\frac{\mathrm{d}\overleftarrow{\mathbb{P}}^{\mu^\star}}{\mathrm{d}\overrightarrow{R}} ((\overleftarrow{X}_t)_{t \in [0, T_f]}) = \frac{\mathrm{d}\mu^\star}{\mathrm{d}\gamma^d} (\overleftarrow{X}_{T_f}) = \mathrm{e}^{-g(\overleftarrow{X}_{T_f})} .$$

This implies

$$\mathrm{KL}(\overleftarrow{\mathbb{P}}^{\mu^\star}|\overrightarrow{R}) = \mathrm{KL}(\mu^\star|\gamma^d) = \int \log \frac{\mathrm{d}\mu^\star}{\mathrm{d}\gamma^d} \mathrm{d}\mu^\star = -\int g \mathrm{d}\mu^\star = -\int g \mathrm{d}\overleftarrow{\mathbb{P}}^{\mu^\star}_{T_f} .$$

Combining the previous results, we obtain that the time reversal $\overleftarrow{\mathbb{P}}^{\mu^\star}$ is the optimal solution to the following problem

$$\overleftarrow{\mathbb{P}}^{\mu^\star} = \underset{\mathbb{P}\in\mathcal{P}(\mathbb{D}_{T_f}):\, \mathrm{KL}(\mathbb{P}|\overrightarrow{R})<\infty}{\arg\min} \left( \mathrm{KL}(\mathbb{P}|\overrightarrow{R}) + \int g\mathrm{d}\mathbb{P}_{T_f} \right) ,$$

which is the desired conclusion. $\qquad\square$

Utilizing the expression for $\mathrm{KL}(\mathbb{P}|\overrightarrow{R})$ given by Girsanov's Theorem F.12, we can now frame the corresponding Optimal Control problem.

**Theorem F.20.** *Denote by $\mathcal{D}$ the set of all $U : \mathbb{D}_{T_f} \times [0,T_f] \times \mathsf{X} \to [0,\infty)$ satisfying the integrability condition*

$$\mathbb{E}_{\mathbb{P}} \left[ \int_{[0,T_f]} \sum_{\ell=1}^{d} \varrho^* (|U_t(\overleftarrow{X}^U_{[0,t)}, \varphi^{(\ell)}(\overleftarrow{X}^U_{t-})) - 1|)\mathrm{d}t \right] < \infty ,$$

*which is indeed equivalent to condition* (61). *Then $\overleftarrow{\mathbb{P}}^{\mu^\star}$ is the law of $\overleftarrow{X}^{u^*}$ with $u^*$ is the optimal solution to*

$$\inf_{U\in\mathcal{D}} \mathbb{E} \left[ \lambda \int_{[0,T_f]} \sum_{\ell=1}^{d} h(U_t(\overleftarrow{X}^U_{[0,t)}, \varphi^{(\ell)}(\overleftarrow{X}^U_{t-})))\mathrm{d}t + g(\overleftarrow{X}^U_{T_f}) \right] , \tag{87}$$

$$s.t. \ \mathrm{Law}((\overleftarrow{X}^U_t)_{t\in[0,T_f]}) \in \mathrm{MP}(U\overrightarrow{q}) \,, \text{ with } (U\overrightarrow{q})(\omega,t,x,y) := U_t(\omega_{[0,t)},y)\overrightarrow{q}(x,y) \text{ for } x\neq y \,.$$

***Proof of Theorem F.20.*** Theorem F.20 is a consequence of Theorem F.12 and Proposition F.19. $\qquad\square$

### F.4.1. HAMILTON–JACOBI–BELLMAN EQUATION

The goal of this section is to derive the Hamilton–Jacobi–Bellman (HJB) equation as in Proposition F.16 using the optimal control viewpoint. To this purpose, we first consider the generalization of the previous control problem. Let $J$ be the following cost

$$J(t,x,U) := \mathbb{E} \left[ \lambda \int_{[t,T_f]} \sum_{\ell=1}^{d} h(U_s(\overleftarrow{X}^{t,x,U}_{[t,s)}, \varphi^{(\ell)}(\overleftarrow{X}^{t,x,U}_{s-})))\mathrm{d}s + g(\overleftarrow{X}^{t,x,U}_{T_f}) \right] ,$$

$$s.t. \begin{cases} \mathrm{Law}((\overleftarrow{X}^{t,x,U}_t)_{t\in[0,T_f]}) \in \mathrm{MP}(U\overrightarrow{q}) \,, \\ \overleftarrow{X}^{t,x,U}_{t-} = x \,, \end{cases} \quad \text{for } (x,t,U)\in \mathsf{X}\times[0,T_f]\times\mathcal{D} \,.$$

Consider $V(t,x)$ to be the value function of the previous cost function, *i.e.*,

$$V(t,x) := \inf_{U\in\mathcal{D}} J(t,x,U) \,.$$

The following Dynamic Programming Principle is the main tool to derive the HJB equation.

**Lemma F.21.** *(Touzi, 2012, Theorem 3.3) For any stopping time $\kappa \in [t,T_f]$, the Dynamic Programming Principle (DPP) implies*

$$V(t,x) = \inf_{U\in\mathcal{D}} \mathbb{E} \left[ \lambda \int_{[t,\kappa]} \sum_{\ell=1}^{d} h(U_s(\overleftarrow{X}^{t,x,U}_{[t,s)}, \varphi^{(\ell)}(\overleftarrow{X}^{t,x,U}_{s-})))\mathrm{d}s + V(\kappa, \overleftarrow{X}^{t,x,U}_\kappa) \right] . \tag{88}$$

***Proof of Lemma F.21.*** Refer to Touzi (2012, Section 3.2). $\qquad\square$

The expression of $V$ given in Lemma F.21 leads us to the following HJB equation, that in fact coincides with the one derived in Proposition F.16, and is a characterization of the optimal control to the problem (87).

**Proposition F.22.** *Assume that $V$ is continuously differentiable in time. Then, the optimal control $u^*$ to the problem (87) is a Markovian, i.e., $u_t^*(\omega_{[0,t)}, x) = u_t^*(\omega_{t-}, x)$ for $t \in (0, T_f]$, and admits the following formula*

$$u_t^*(x, \varphi^{(\ell)}(x)) = e^{V(t,x) - V(t, \varphi^{(\ell)}(x))} \quad \text{for } \ell = 1, 2, \dots, d,$$

*with $V$ satisfies the following HJB equation*

$$\begin{cases} \partial_t V(t, x) - \lambda \sum_{\ell=1}^d e^{V(t,x) - V(t, \varphi^{(\ell)}(x))} = -\lambda d, \\ V(T_f, x) = g(x) = -\log \frac{d\mu^\star}{d\gamma^d}(x), \end{cases} \quad \text{for } (t, x) \in [0, T_f) \times \mathsf{X}. \tag{89}$$

***Proof of Proposition F.22.*** The proof is an adaptation of Proposition 3.5 in Touzi (2012). First, the DPP formula (88) for $t \in [0, T_f)$ and $\kappa = t + \alpha$ with $\alpha > 0$ leads to

$$\mathbb{E}\left[\lambda \int_{[t, t+\alpha]} \sum_{\ell=1}^d h(U_s(\overleftarrow{X}_{[t,s)}^{t,x,U}, \varphi^{(\ell)}(\overleftarrow{X}_{s-}^{t,x,U}))) ds + V(t + \alpha, \overleftarrow{X}_{t+\alpha}^{t,x,U}) - V(t, x)\right] \geqslant 0,$$

for any admissible control $U \in \mathcal{D}$. Using Itô's formula on the process $\overleftarrow{X}^{t,x,U}$ with the law $\overleftarrow{\mathbb{P}} \in \mathrm{MP}(U\overrightarrow{q})$, we get

$$\mathbb{E}\left[\int_{[t, t+\alpha]} \lambda \sum_{\ell=1}^d h(U_s(\overleftarrow{X}_{[t,s)}^{t,x,U}, \varphi^{(\ell)}(\overleftarrow{X}_{s-}^{t,x,U}))) ds + \int_{[t, t+\alpha]} (\partial_t V(s, \overleftarrow{X}_s^{t,x,U}) + (U_s \overrightarrow{q}) V_s(\overleftarrow{X}_s^{t,x,U})) ds\right] \geqslant 0.$$

Using the formula of $\overrightarrow{q}$ and multiplying the both hand sides by $\frac{1}{\alpha}$ and pushing $\alpha \to 0$, we arrive at

$$\lambda \sum_{\ell=1}^d h(U_t(x, \varphi^{(\ell)}(x))) + \partial_t V(t, x) + \lambda \sum_{\ell=1}^d \left[V(t, \varphi^{(\ell)}(x)) - V(t, x)\right] U_t(x, \varphi^{(\ell)}(x)) \geqslant 0,$$

for any $U \in \mathcal{D}$. Taking the infimum w.r.t. $U$, we get

$$\partial_t V(t, x) + \lambda \inf_{U \in \mathcal{D}} \sum_{\ell=1}^d \left[h(U_t(x, \varphi^{(\ell)}(x))) + [V(t, \varphi^{(\ell)}(x)) - V(t, x)] U_t(x, \varphi^{(\ell)}(x))\right] \geqslant 0, \quad \text{for } (t, x) \in [0, T_f) \times \mathsf{X}.$$

We prove next the equality by contradiction. Assume that there exists $(t_0, x_0) \in [0, T_f] \times \mathsf{X}$ such that

$$\partial_t V(t_0, x_0) + \lambda \inf_{U \in \mathcal{D}} \sum_{\ell=1}^d \left[h(U_{t_0}(x_0, \varphi^{(\ell)}(x_0))) + [V(t_0, \varphi^{(\ell)}(x_0)) - V(t_0, x_0)] U_{t_0}(x_0, \varphi^{(\ell)}(x_0))\right] > 0.$$

Denote $\Delta V(t_0, x_0, \varphi^{(\ell)}(x_0)) := V(t_0, \varphi^{(\ell)}(x_0)) - V(t_0, x_0)$. The previous inequality implies that there exists $\varepsilon > 0$ such that

$$\partial_t V(t_0, x_0) + \lambda \inf_{U \in \mathcal{D}} \sum_{\ell=1}^d \left[h(U) + U\Delta V\right](t_0, x_0, \varphi^{(\ell)}(x_0)) \geqslant \varepsilon > 0. \tag{90}$$

Take $\xi > 0$ small enough such that

$$\lambda \sum_{\ell=1}^d (e^{-\Delta V + \xi} - e^{-\Delta V})(t_0, x_0, \varphi^{(\ell)}(x_0)) < \frac{\varepsilon}{2}, \tag{91}$$

and define the function $f \leqslant V$ as

$$f(t, x) := V(t, x) - \xi\left[|t - t_0|^2 + \delta_{\{x_0\}}(x)\right], \quad \text{for } (t, x) \in [0, T_f] \times \mathsf{X}.$$

It is clear that

$$f(t_0, x_0) = V(t_0, x_0) , \quad \partial_t f(t_0, x_0) = \partial_t V(t_0, x_0) , \quad \text{and} \quad f(t_0, x) - V(t_0, x) = -\xi \text{ for } x \neq x_0 .$$

Therefore,

$$\partial_t f(t_0, x_0) + \lambda \inf_{U \in \mathcal{D}} \sum_{\ell=1}^{d} \left[ h(U_{t_0}(x_0, \varphi^{(\ell)}(x_0))) + [f(t_0, \varphi^{(\ell)}(x_0)) - f(t_0, x_0)]U_{t_0}(x_0, \varphi^{(\ell)}(x_0)) \right]$$

$$= \partial_t V(t_0, x_0) + \lambda \inf_{U \in \mathcal{D}} \sum_{\ell=1}^{d} \left[ h(U_{t_0}(x_0, \varphi^{(\ell)}(x_0))) + \left[ V(t_0, \varphi^{(\ell)}(x_0)) - V(t_0, x_0) - \xi \right] U_{t_0}(x_0, \varphi^{(\ell)}(x_0)) \right]$$

$$= \partial_t V(t_0, x_0) + \lambda \inf_{U \in \mathcal{D}} \sum_{\ell=1}^{d} \left[ h(U) + (\Delta V - \xi)U \right] (t_0, x_0, \varphi^{(\ell)}(x_0)) .$$

The minimum above is attained at $U$ such that $U(t_0, x_0, \varphi^{(\ell)}(x_0)) = \mathrm{e}^{-\Delta V + \xi}(t_0, x_0, \varphi^{(\ell)}(x_0))$ , thus

$$\partial_t f(t_0, x_0) + \lambda \inf_{U \in \mathcal{D}} \sum_{\ell=1}^{d} \left[ h(U_{t_0}(x_0, \varphi^{(\ell)}(x_0))) + [f(t_0, \varphi^{(\ell)}(x_0)) - f(t_0, x_0)]U_{t_0}(x_0, \varphi^{(\ell)}(x_0)) \right]$$

$$= \partial_t V(t_0, x_0) + \lambda \sum_{\ell=1}^{d} \left[ h(\mathrm{e}^{-\Delta V + \xi}) + (\Delta V - \xi)\mathrm{e}^{-\Delta V + \xi} \right] (t_0, x_0, \varphi^{(\ell)}(x_0))$$

$$= \partial_t V(t_0, x_0) + \lambda \sum_{\ell=1}^{d} (1 - \mathrm{e}^{-\Delta V + \xi})(t_0, x_0, \varphi^{(\ell)}(x_0))$$

$$= \partial_t V(t_0, x_0) + \lambda \sum_{\ell=1}^{d} (1 - \mathrm{e}^{-\Delta V})(t_0, x_0, \varphi^{(\ell)}(x_0)) + \lambda \sum_{\ell=1}^{d} (\mathrm{e}^{-\Delta V} - \mathrm{e}^{-\Delta V + \xi})(t_0, x_0, \varphi^{(\ell)}(x_0))$$

$$> \varepsilon - \frac{\varepsilon}{2} = \frac{\varepsilon}{2} > 0 ,$$

where the last inequality relies on (91) and (90) with $U = \mathrm{e}^{-\Delta V}$. Therefore, we obtain

$$\partial_t f(t_0, x_0) + \lambda \inf_{U \in \mathcal{D}} \sum_{\ell=1}^{d} \left[ h(U_{t_0}(x_0, \varphi^{(\ell)}(x_0))) + [f(t, \varphi^{(\ell)}(x_0)) - f(t_0, x_0)]U_{t_0}(x_0, \varphi^{(\ell)}(x_0)) \right] > 0 .$$

From the continuity in time of the Hamiltonian, the previous inequality yields that

$$\partial_t f(t, x) + \lambda \inf_{U \in \mathcal{D}} \sum_{\ell=1}^{d} \left[ h(U_t(x, \varphi^{(\ell)}(x))) + [f(t, \varphi^{(\ell)}(x)) - f(t, x)]U_t(x, \varphi^{(\ell)}(x)) \right] \geqslant 0 ,$$

$$\text{for } (t, x) \in (t_0 - r, t_0 + r) \times \{x_0\} , \tag{92}$$

for some $0 < r < 1$. Defining the stopping time $\kappa^U$ as

$$\kappa^U := \inf \left\{ t \in (t_0, T_f] : \overleftarrow{X}_{t-}^{t_0, x_0, U} \neq x_0 \right\} \wedge (t_0 + r) ,$$

for an arbitrary control $U$, we have

$$f(\kappa^U, \overleftarrow{X}_{\kappa^U}^{t_0, x_0, u}) = \begin{cases} f(t_0 + r, \overleftarrow{X}_{t_0 + r}^{t_0, x_0, U}) , & \text{if } \overleftarrow{X}_{\kappa^U-}^{t_0, x_0, U} = x_0 , \\ f(\kappa^U, \overleftarrow{X}_{\kappa^U}^{t_0, x_0, U}) , & \text{if } \overleftarrow{X}_{\kappa^U-}^{t_0, x_0, U} \neq x_0 . \end{cases}$$

This implies that

$$f(\kappa^U, \overleftarrow{X}^{t_0,x_0,u}_{\kappa^U_-}) - V(\kappa^U, \overleftarrow{X}^{t_0,x_0,u}_{\kappa^U}) = \begin{cases} -\xi r^2 \,, & \text{if} \quad \overleftarrow{X}^{t_0,x_0,u}_{\kappa^U_-} = x_0 \,, \\ -\xi(|\kappa^U - t_0|^2 + 1) \,, & \text{if} \quad \overleftarrow{X}^{t_0,x_0,u}_{\kappa^U_-} \neq x_0 \,, \end{cases}$$
$$\leqslant -\xi r^2 \,.$$

Note that for any $s \in [t_0, \kappa^U]$, not only $\overleftarrow{X}^{t_0,x_0,U}_{s-} = x_0$, thus $\overleftarrow{X}^{t_0,x_0,U}_{[t_0,s)} = x_0$. Therefore,

$$\mathbb{E}\left[\int_{[t_0,\kappa^U]} \lambda \sum_{\ell=1}^d h(U_s(\overleftarrow{X}^{t_0,x_0,U}_{[t_0,s)}, \varphi^{(\ell)}(\overleftarrow{X}^{t_0,x_0,U}_{s-}))) \mathrm{d}s + V(\kappa^U, \overleftarrow{X}^{t_0,x_0,U}_{\kappa^U})\right]$$

$$\geqslant \mathbb{E}\left[\int_{[t_0,\kappa^U]} \lambda \sum_{\ell=1}^d h(U_s(x_0, \varphi^{(\ell)}(x_0))) \mathrm{d}s + f(\kappa^U, \overleftarrow{X}^{t_0,x_0,U}_{\kappa^U}) + \xi r^2\right]$$

$$= \mathbb{E}\left[\int_{[t_0,\kappa^U]} \lambda \sum_{\ell=1}^d h(U_s(x_0, \varphi^{(\ell)}(x_0))) \mathrm{d}s + f(\kappa^U, \overleftarrow{X}^{t_0,x_0,U}_{\kappa^U}) - f(t_0, x_0)\right] + f(t_0, x_0) + \xi r^2 \,.$$

Using Itô's formula to compute the difference $f(\kappa^U, \overleftarrow{X}^{t_0,x_0,U}_{\kappa^U}) - f(t_0, x_0)$, and relies on the fact that $V(t_0, x_0) = f(t_0, x_0)$, the calculation follows

$$\mathbb{E}\left[\int_{[t_0,\kappa^U]} \lambda \sum_{\ell=1}^d h(U_s(\overleftarrow{X}^{t_0,x_0,U}_{[t_0,s)}, \varphi^{(\ell)}(\overleftarrow{X}^{t_0,x_0,U}_{s-}))) \mathrm{d}s + V(\kappa^U, \overleftarrow{X}^{t_0,x_0,U}_{\kappa^U})\right]$$

$$\geqslant \mathbb{E}\int_{[t_0,\kappa^U]} \left[\partial_t f(s, x_0) + \lambda \sum_{\ell=1}^d h(U_s(x_0, \varphi^{(\ell)}(x_0)))\right.$$
$$\left. + (f(s, \varphi^{(\ell)}(x_0)) - f(s, x_0))U_s(x_0, \varphi^{(\ell)}(x_0))\right] \mathrm{d}s + V(t_0, x_0) + \xi r^2$$

$$\overset{(92)}{\geqslant} V(t_0, x_0) + \xi r^2 \,.$$

Since the above control $U$ is arbitrary, the previous inequality is indeed a contradiction to DPP formula (88).

Consequently, we can deduce the following HJB equation satisfied by the value function for $(t, x) \in [0, T_f] \times \mathsf{X}$,

$$\begin{cases} \partial_t V(t, x) + \lambda \inf_{U \in \mathcal{D}} \sum_{\ell=1}^d \left[h(U_t(x, \varphi^{(\ell)}(x))) + [V(t, \varphi^{(\ell)}(x)) - V(t, x)]U_t(x, \varphi^{(\ell)}(x))\right] = 0 \,, \\ V(T_f, x) = g(x) \,. \end{cases} \tag{93}$$

Proceeding as before, we minimize (93) and obtain the optimal control

$$u^*_t(\overleftarrow{X}^{t,x,u^*}_{[0,t)}, \varphi^{(\ell)}(\overleftarrow{X}^{t,x,u^*}_{t-})) = u^*_t(\overleftarrow{X}^{t,x,u^*}_{t-}, \varphi^{(\ell)}(\overleftarrow{X}^{t,x,u^*}_{t-})) = u^*_t(x, \varphi^{(\ell)}(x)) = \mathrm{e}^{V(t,x)-V(t,\varphi^{(\ell)}(x))} \,, \quad \text{for } \ell = 1, \ldots, d \,,$$

that is Markovian. Replacing the formulation of $u^*$ into (93) boils down to

$$\begin{cases} \partial_t V(t, x) - \lambda \sum_{\ell=1}^d \mathrm{e}^{V(t,x)-V(t,\varphi^{(\ell)}(x))} = -\lambda d \,, \\ V(T_f, x) = g(x) = -\log \frac{\mathrm{d}\mu^\star}{\mathrm{d}\gamma^d}(x) \,, \end{cases} \quad \text{for } (t, x) \in [0, T_f] \times \mathsf{X} \,,$$

which in fact coincides with the equation in Proposition F.16 and concludes the proof of Theorem F.22. □

### F.5. Convergence of DMPMs

Based on the perspective of the canonical process, the backward evolution can be described as a control-driven dynamic. These tools enable us to establish the error bounds presented in Theorem 2.3 and Theorem 2.4.

F.5.1. PROOF OF THEOREM 2.3

We first prove the curvature–dimension inequality satisfied by our forward dynamics, which is associated with the stationary distribution $\gamma^d = \mathrm{Uniform}(\mathsf{X})$. This serves as the key estimate for deriving the entropy decay results later.

**Lemma F.23** (Curvature–dimension inequality). *The forward dynamic described above satisfies* curvature–dimension inequality $\mathrm{CD}(2\lambda, \infty)$, *i.e.*,

$$\Gamma_2(f) \geqslant 2\lambda\Gamma(f) \quad \textit{for any function } f \ ,$$

*where $\Gamma$ is the carré du champ operator and $\Gamma_2$ is the iterated carré du champ operator.*

*Proof of Lemma F.23.* Recall the formulations of $\Gamma$ and $\Gamma_2$ for functions $f$ and $g$, which are typically defined as follows:

$$\Gamma(f,g) = \frac{1}{2}\left[\overrightarrow{q}(fg) - f(\overrightarrow{q}g) - (\overrightarrow{q}f)g\right] \quad \text{and} \quad \Gamma(f,f) = \Gamma(f) = \frac{1}{2}\left[\overrightarrow{q}(f^2) - 2f\overrightarrow{q}f\right]$$

$$\Gamma_2(f) = \frac{1}{2}\left[\overrightarrow{q}\Gamma(f) - 2\Gamma(\overrightarrow{q}f,f)\right] \ ,$$

where $\overrightarrow{q}$ is the forward generator defined in (14). These quantities capture the interaction between the functions $f$ and $g$ under the generator $\overrightarrow{q}$ and play a crucial role in establishing results related to curvature–dimension inequalities and entropy decay. We now compute explicitly $\Gamma(f)(x)$ for $x \in \mathsf{X}$ as

$$\begin{aligned}
\Gamma(f)(x) &= \frac{1}{2}\left[\overrightarrow{q}(f^2) - 2f\overrightarrow{q}f\right] \\
&= \frac{\lambda}{2}\left[\sum_{\ell=1}^{d}\left(f^2(\varphi^{(\ell)}(x)) - f^2(x)\right) - 2f(x)\sum_{\ell=1}^{d}\left(f(\varphi^{(\ell)}(x)) - f(x)\right)\right] \\
&= \frac{\lambda}{2}\sum_{\ell=1}^{d}\left[f(\varphi^{(\ell)}(x)) - f(x)\right]^2 \ .
\end{aligned}$$

Regarding the iterated carré du champ $\Gamma_2(f)$, the first term can be calculated as

$$\begin{aligned}
\overrightarrow{q}\Gamma(f)(x) &= \lambda\sum_{i=1}^{d}\left[\Gamma(f)(\varphi^{(i)}(x)) - \Gamma(f)(x)\right] \\
&= \lambda\sum_{i=1}^{d}\left[\frac{\lambda}{2}\sum_{\ell=1}^{d}\left(f(\varphi^{(\ell)}(\varphi^{(i)}(x))) - f(\varphi^{(i)}(x))\right)^2 - \frac{\lambda}{2}\sum_{\ell=1}^{d}\left(f(\varphi^{(\ell)}(x)) - f(x)\right)^2\right] \\
&= \frac{\lambda^2}{2}\sum_{i=1}^{d}\sum_{\ell=1}^{d}\left[\left(f(\varphi^{(\ell)}(\varphi^{(i)}(x))) - f(\varphi^{(i)}(x))\right)^2 - \left(f(\varphi^{(\ell)}(x)) - f(x)\right)^2\right] \ .
\end{aligned} \tag{94}$$

To simplify the second term of $\Gamma_2$, we note that

$$\begin{aligned}
2\Gamma(f,g)(x) &= \overrightarrow{q}(fg) - f(\overrightarrow{q}g) - g(\overrightarrow{q}f) \\
&= \lambda\sum_{\ell=1}^{d}\left[f(\varphi^{(\ell)}(x))g(\varphi^{(\ell)}(x)) - f(x)g(\varphi^{(\ell)}(x)) + f(x)g(x) - g(x)f(\varphi^{(\ell)}(x))\right] \\
&= \lambda\sum_{\ell=1}^{d}\left[f(x) - f(\varphi^{(\ell)}(x))\right]\left[g(x) - g(\varphi^{(\ell)}(x))\right] \ .
\end{aligned}$$

Applying this to $g = \overrightarrow{q}f$ yields

$$2\Gamma(f, \overrightarrow{q} f)(x) = \lambda \sum_{\ell=1}^{d} \left[ f(x) - f(\varphi^{(\ell)}(x)) \right] \left[ \lambda \sum_{i=1}^{d} \left( f(\varphi^{(i)}(x)) - f(x) \right) - \lambda \sum_{i=1}^{d} \left( f(\varphi^{(\ell)}(\varphi^{(i)}(x))) - f(\varphi^{(\ell)}(x)) \right) \right]$$

$$= \lambda^2 \sum_{\ell=1}^{d} \sum_{i=1}^{d} \left[ f(x) - f(\varphi^{(\ell)}(x)) \right] \left[ f(\varphi^{(i)}(x)) - f(x) - f(\varphi^{(\ell)}(\varphi^{(i)}(x))) + f(\varphi^{(\ell)}(x)) \right] . \tag{95}$$

Plugging (94) and (95) into $\Gamma_2$ yields

$$\Gamma_2(f)(x) = \frac{1}{2} \left[ \overrightarrow{q} \Gamma(f) - 2\Gamma(\overrightarrow{q} f, f) \right]$$

$$= \frac{\lambda^2}{4} \sum_{\ell=1}^{d} \sum_{i=1}^{d} \left[ \left( f(\varphi^{(\ell)}(\varphi^{(i)}(x))) - f(\varphi^{(i)}(x)) \right)^2 - \left( f(\varphi^{(\ell)}(x)) - f(x) \right)^2 \right.$$

$$\left. + 2 \left( f(x) - f(\varphi^{(\ell)}(x)) \right)^2 - 2 \left( f(x) - f(\varphi^{(\ell)}(x)) \right) \left( f(\varphi^{(i)}(x)) - f(\varphi^{(\ell)}(\varphi^{(i)}(x))) \right) \right]$$

$$= \frac{\lambda^2}{4} \sum_{\ell=1}^{d} \sum_{i=1}^{d} \left[ f(\varphi^{(\ell)}(\varphi^{(i)}(x))) - f(\varphi^{(i)}(x)) - f(\varphi^{(\ell)}(x)) + f(x) \right]^2$$

$$\geqslant \frac{\lambda^2}{4} \sum_{\ell=1}^{d} \left[ f(x) - f(\varphi^{(\ell)}(x)) - f(\varphi^{(\ell)}(x)) + f(x) \right]^2$$

$$= \lambda^2 \sum_{\ell=1}^{d} \left[ f(\varphi^{(\ell)}(x)) - f(x) \right]^2 = 2\lambda \Gamma(f)(x) , \quad \text{for any } x \in \mathsf{X} .$$

Thus the desired inequality holds and we conclude the proof.

$\square$

We are now prepared to analyze the key distinguishing result of this paper.

***Proof of Theorem 2.3.*** We begin by establishing a bound on the "distance" between the backward path measure in continuous time, $\overleftarrow{\mathbb{P}}^{\mu^*}$, associated with the controlled process $(\overleftarrow{X}_t)_{t \in [0, T_f]}$, and the path measure $\overleftarrow{\mathbb{P}}^{\star}$ corresponding to the simulated backward process $(\overleftarrow{X}^{\star}_t)_{t \in [0, T_f]}$ generated in Algorithm 3. For brevity, we denote the backward path measure $\overleftarrow{\mathbb{P}}^{\mu^*}$ by $\overleftarrow{\mathbb{P}}$ throughout the remainder of the paper. By taking the stationary forward path measure $\overrightarrow{R} \in \mathrm{MP}(\overrightarrow{q})$ as the reference in Girsanov's Theorem F.12, we derive the Radon–Nikodym density of $\overleftarrow{\mathbb{P}} \in \mathrm{MP}(u\overrightarrow{q})$ with respect to $\overrightarrow{R}$, where the control $u$ is specified in (84), as follows

$$\frac{\mathrm{d}\overleftarrow{\mathbb{P}}}{\mathrm{d}\overrightarrow{R}}((\mathbf{X}_t)_{t \in [0, T_f]}) = \frac{\mathrm{d}\overleftarrow{\mathbb{P}}_0}{\mathrm{d}\overrightarrow{R}_0}(\mathbf{X}_0) \exp \left( \int_{[0, T_f] \times \mathsf{X}} \log u_t(\mathbf{X}_{t-}, x) \tilde{N}^{\overrightarrow{q}}_{\mathbf{X}}(\mathrm{d}t\mathrm{d}x) - \int_{[0, T_f] \times \mathsf{X}} \varrho(\log u_t(\mathbf{X}_{t-}, x)) \bar{\mathrm{n}}^{\overrightarrow{q}}(\mathrm{d}t\mathrm{d}x) \right) .$$

With a partition $0 = t_0 < ... < t_K = T_f$ for $K \geqslant 1$ of $[0, T_f]$ associated with the sequence of step-size $\{\tau_k\}_{k=1}^{K} : \tau_{k+1} = t_{k+1} - t_k$, the previous expression rewrites

$$
\frac{\mathrm{d}\overleftarrow{\mathbb{P}}}{\mathrm{d}\overrightarrow{R}}((\mathbf{X}_t)_{t\in[0,T_f]}) = \frac{\mathrm{d}\overleftarrow{\mathbb{P}}_0}{\mathrm{d}\overrightarrow{R}_0}(\mathbf{X}_0)\exp\sum_{k=0}^{K-1}\left(\int_{[t_k,t_{k+1})\times\mathsf{X}}\log u_t(\mathbf{X}_{t-},x)\tilde{N}_{\mathbf{X}}^{\overrightarrow{q}}(\mathrm{d}t\mathrm{d}x)\right.
$$

$$
\left. - \int_{[t_k,t_{k+1})\times\mathsf{X}}\varrho(\log u_t(\mathbf{X}_{t-},x))\bar{\mathrm{n}}^{\overrightarrow{q}}(\mathrm{d}t\mathrm{d}x)\right)
$$

$$
= \frac{\mathrm{d}\overleftarrow{\mathbb{P}}_0}{\mathrm{d}\overrightarrow{R}_0}(\mathbf{X}_0)\exp\sum_{k=0}^{K-1}\left(\int_{[t_k,t_{k+1})\times\mathsf{X}}\log u_t(\mathbf{X}_{t-},x)\tilde{N}_{\mathbf{X}}^{u\overrightarrow{q}}(\mathrm{d}t\mathrm{d}x)\right.
$$

$$
\left. + \int_{[t_k,t_{k+1})\times\mathsf{X}}\log u_t(\mathbf{X}_{t-},x)\bar{\mathrm{n}}^{(u-1)\overrightarrow{q}}(\mathrm{d}t\mathrm{d}x) - \int_{[t_k,t_{k+1})\times\mathsf{X}}\varrho(\log u_t(\mathbf{X}_{t-},x))\bar{\mathrm{n}}^{\overrightarrow{q}}(\mathrm{d}t\mathrm{d}x)\right)
$$

$$
= \frac{\mathrm{d}\overleftarrow{\mathbb{P}}_0}{\mathrm{d}\overrightarrow{R}_0}(\mathbf{X}_0)\exp\sum_{k=0}^{K-1}\left(\int_{[t_k,t_{k+1})\times\mathsf{X}}\log u_t(\mathbf{X}_{t-},x)\tilde{N}_{\mathbf{X}}^{u\overrightarrow{q}}(\mathrm{d}t\mathrm{d}x)\right.
$$

$$
+ \lambda\int_{[t_k,t_{k+1})}\sum_{\ell=1}^{d}(u_t-1)\log u_t(\mathbf{X}_{t-},\varphi^{(\ell)}(\mathbf{X}_{t-}))\mathrm{d}t
$$

$$
\left. - \lambda\int_{[t_k,t_{k+1})}\sum_{\ell=1}^{d}(u_t-\log u_t-1)(\mathbf{X}_{t-},\varphi^{(\ell)}(\mathbf{X}_{t-}))\mathrm{d}t\right)
$$

$$
= \frac{\mathrm{d}\overleftarrow{\mathbb{P}}_0}{\mathrm{d}\overrightarrow{R}_0}(\mathbf{X}_0)\exp\sum_{k=0}^{K-1}\left(\int_{[t_k,t_{k+1})\times\mathsf{X}}\log u_t(\mathbf{X}_{t-},x)\tilde{N}_{\mathbf{X}}^{u\overrightarrow{q}}(\mathrm{d}t\mathrm{d}x)\right.
$$

$$
\left. + \lambda\int_{[t_k,t_{k+1})}\sum_{\ell=1}^{d}(u_t\log u_t-u_t+1)(\mathbf{X}_t,\varphi^{(\ell)}(\mathbf{X}_t))\mathrm{d}t\right),
$$

as $\mathbf{X}_{t-}=\mathbf{X}_t$ for Lebesgue almost all $t\in[0,T_f]$. Applying Girsanov's theorem F.12 once more to the path measure $\overleftarrow{\mathbb{P}}^\star\in\mathrm{MP}(\hat{u}^{\theta^\star}\overrightarrow{q})$ associated with the process $(\overleftarrow{X}_t^\star)_{t\in[0,T_f]}$ generated by Algorithm 3, we obtain the following expression:

$$
\frac{\mathrm{d}\overleftarrow{\mathbb{P}}^\star}{\mathrm{d}\overrightarrow{R}}((\mathbf{X}_t)_{t\in[0,T_f]}) = \frac{\mathrm{d}\overleftarrow{\mathbb{P}}_0^\star}{\mathrm{d}\overrightarrow{R}_0}(\mathbf{X}_0)\exp\sum_{k=0}^{K-1}\left(\int_{[t_k,t_{k+1})\times\mathsf{X}}\log\hat{u}_t^{\theta^\star}(\mathbf{X}_{t-},x)\tilde{N}_{\mathbf{X}}^{\overrightarrow{q}}(\mathrm{d}t\mathrm{d}x)\right.
$$

$$
\left. - \int_{[t_k,t_{k+1})\times\mathsf{X}}\varrho(\log\hat{u}_t^{\theta^\star}(\mathbf{X}_{t-},x))\bar{\mathrm{n}}^{\overrightarrow{q}}(\mathrm{d}t\mathrm{d}x)\right)
$$

$$
= \frac{\mathrm{d}\overleftarrow{\mathbb{P}}_0^\star}{\mathrm{d}\overrightarrow{R}_0}(\mathbf{X}_0)\exp\sum_{k=0}^{K-1}\left(\int_{[t_k,t_{k+1})\times\mathsf{X}}\log\hat{u}_t^{\theta^\star}(\mathbf{X}_{t-},x)\tilde{N}_{\mathbf{X}}^{u\overrightarrow{q}}(\mathrm{d}t\mathrm{d}x)\right.
$$

$$
+ \int_{[t_k,t_{k+1})\times\mathsf{X}}\log\hat{u}_t^{\theta^\star}(\mathbf{X}_{t-},x)\bar{\mathrm{n}}^{(u-1)\overrightarrow{q}}(\mathrm{d}t\mathrm{d}x) - \int_{[t_k,t_{k+1})\times\mathsf{X}}\varrho(\log\hat{u}_t^{\theta^\star}(\mathbf{X}_{t-},x))\bar{\mathrm{n}}^{\overrightarrow{q}}(\mathrm{d}t\mathrm{d}x)\right)
$$

$$
= \frac{\mathrm{d}\overleftarrow{\mathbb{P}}_0^\star}{\mathrm{d}\overrightarrow{R}_0}(\mathbf{X}_0)\exp\sum_{k=0}^{K-1}\left(\int_{[t_k,t_{k+1})\times\mathsf{X}}\log\hat{u}_t^{\theta^\star}(\mathbf{X}_{t-},x)\tilde{N}_{\mathbf{X}}^{u\overrightarrow{q}}(\mathrm{d}t\mathrm{d}x)\right.
$$

$$
\left. + \lambda\int_{[t_k,t_{k+1})}\sum_{\ell=1}^{d}(u_t\log\hat{u}_t^{\theta^\star}-\hat{u}_t^{\theta^\star}+1)(\mathbf{X}_{t-},\varphi^{(\ell)}(\mathbf{X}_{t-}))\mathrm{d}t\right).
$$

Since $\mathbf{X}_t=\mathbf{X}_{t-}$ for Lebesgue almost all $t\in[0,T_f]$ and using the fact that $\hat{u}_t^{\theta^\star}(\mathbf{X}_t,\varphi^{(\ell)}(\mathbf{X}_t))=u_{t_k}^{\theta^\star}(\mathbf{X}_{t_k},\varphi^{(\ell)}(\mathbf{X}_{t_k}))$ for any $t\in[t_k,t_{k+1})$, $\ell=1,\dots,d$ and $(\mathbf{X}_t)_{t\in[0,T_f]}$, the calculation follows

$$\frac{\mathrm{d}\overleftarrow{\mathbb{P}}^{\star}}{\mathrm{d}\overrightarrow{R}}((\mathbf{X}_t)_{t\in[0,T_f]}) = \frac{\mathrm{d}\overleftarrow{\mathbb{P}}_0^{\star}}{\mathrm{d}\overrightarrow{R}_0}(\mathbf{X}_0)\exp\sum_{k=0}^{K-1}\left(\int_{[t_k,t_{k+1})\times\mathsf{X}}\log\hat{u}_t^{\theta^{\star}}(\mathbf{X}_{t-},x)\tilde{N}_{\mathbf{X}}^{u\overrightarrow{q}}(\mathrm{d}t\mathrm{d}x)\right.$$
$$\left.+\lambda\int_{[t_k,t_{k+1})}\sum_{\ell=1}^d(u_t(\mathbf{X}_t,\varphi^{(\ell)}(\mathbf{X}_t))\log u_{t_k}^{\theta^{\star}}(\mathbf{X}_{t_k},\varphi^{(\ell)}(\mathbf{X}_{t_k})) - u_{t_k}^{\theta^{\star}}(\mathbf{X}_{t_k},\varphi^{(\ell)}(\mathbf{X}_{t_k})) + 1)\mathrm{d}t\right).$$

Combining two Radon-Nikodym densities $\mathrm{d}\overleftarrow{\mathbb{P}}/\mathrm{d}\overrightarrow{R}$ and $\mathrm{d}\overleftarrow{\mathbb{P}}^{\star}/\mathrm{d}\overrightarrow{R}$, we deduce that

$$\frac{\mathrm{d}\overleftarrow{\mathbb{P}}}{\mathrm{d}\overleftarrow{\mathbb{P}}^{\star}}((\mathbf{X}_t)_{t\in[0,T_f]}) = \frac{\mathrm{d}\overleftarrow{\mathbb{P}}_0}{\mathrm{d}\overleftarrow{\mathbb{P}}_0^{\star}}(\mathbf{X}_0)\exp\sum_{k=0}^{K-1}\left(\int_{[t_k,t_{k+1})\times\mathsf{X}}(\log u_t(\mathbf{X}_{t-},x)-\log\hat{u}_t^{\theta^{\star}}(\mathbf{X}_{t-},x))\tilde{N}_{\mathbf{X}}^{u\overrightarrow{q}}(\mathrm{d}t\mathrm{d}x)\right.$$
$$+\lambda\int_{[t_k,t_{k+1})}\sum_{\ell=1}^d\left(u_t(\mathbf{X}_t,\varphi^{(\ell)}(\mathbf{X}_t))\log\frac{u_t(\mathbf{X}_t,\varphi^{(\ell)}(\mathbf{X}_t))}{u_{t_k}^{\theta^{\star}}(\mathbf{X}_{t_k},\varphi^{(\ell)}(\mathbf{X}_{t_k}))}\right.$$
$$\left.\left.- u_t(\mathbf{X}_t,\varphi^{(\ell)}(\mathbf{X}_t)) + u_{t_k}^{\theta^{\star}}(\mathbf{X}_{t_k},\varphi^{(\ell)}(\mathbf{X}_{t_k}))\right)\mathrm{d}t\right).$$

This leads to the following expression of the KL divergence

$$\mathrm{KL}(\overleftarrow{\mathbb{P}}\,|\,\overleftarrow{\mathbb{P}}^{\star}) = \mathrm{KL}(\mu_{T_f}|\gamma^d) + \sum_{k=0}^{K-1}\mathbb{E}_{\overleftarrow{\mathbb{P}}}\left[\int_{[t_k,t_{k+1})\times\mathsf{X}}(\log u_t(\overleftarrow{X}_{t-},x)-\log\hat{u}_t^{\theta^{\star}}(\overleftarrow{X}_{t-},x))\tilde{N}_{\overleftarrow{X}}^{u\overrightarrow{q}}(\mathrm{d}t\mathrm{d}x)\right.$$
$$+\lambda\int_{[t_k,t_{k+1})}\sum_{\ell=1}^d\left(u_t(\overleftarrow{X}_t,\varphi^{(\ell)}(\overleftarrow{X}_t))\log\frac{u_t(\overleftarrow{X}_t,\varphi^{(\ell)}(\overleftarrow{X}_t))}{u_{t_k}^{\theta^{\star}}(\overleftarrow{X}_{t_k},\varphi^{(\ell)}(\overleftarrow{X}_{t_k}))}\right.$$
$$\left.\left.- u_t(\overleftarrow{X}_t,\varphi^{(\ell)}(\overleftarrow{X}_t)) + u_{t_k}^{\theta^{\star}}(\overleftarrow{X}_{t_k},\varphi^{(\ell)}(\overleftarrow{X}_{t_k}))\right)\mathrm{d}t\right].$$

Note that $\overleftarrow{\mathbb{P}}\in\mathrm{MP}(u\overrightarrow{q})$, thus $\tilde{N}_{\mathbf{X}}^{u\overrightarrow{q}}$ is a $\overleftarrow{\mathbb{P}}$-martingale, which in turn allow us to reduce the first integral above. Hence we can simplify the expression for the KL divergence as follows

$$\mathrm{KL}(\overleftarrow{\mathbb{P}}\,|\,\overleftarrow{\mathbb{P}}^{\star}) = \mathrm{KL}(\mu_{T_f}|\gamma^d) + \sum_{k=0}^{K-1}\mathbb{E}_{\overleftarrow{\mathbb{P}}}\left[\lambda\int_{[t_k,t_{k+1})}\sum_{\ell=1}^d\left(u_t(\overleftarrow{X}_t,\varphi^{(\ell)}(\overleftarrow{X}_t))\log\frac{u_t(\overleftarrow{X}_t,\varphi^{(\ell)}(\overleftarrow{X}_t))}{u_{t_k}^{\theta^{\star}}(\overleftarrow{X}_{t_k},\varphi^{(\ell)}(\overleftarrow{X}_{t_k}))}\right.\right.$$
$$\left.\left.- u_t(\overleftarrow{X}_t,\varphi^{(\ell)}(\overleftarrow{X}_t)) + u_{t_k}^{\theta^{\star}}(\overleftarrow{X}_{t_k},\varphi^{(\ell)}(\overleftarrow{X}_{t_k}))\right)\mathrm{d}t\right].$$

For brevity, we denote $u_t(\overleftarrow{X}_t,\varphi^{(\ell)}(\overleftarrow{X}_t))$ as $u_t^{\ell}$ throughout the remainder of the proof. We rewrite $\mathrm{KL}(\overleftarrow{\mathbb{P}}\,|\,\overleftarrow{\mathbb{P}}^{\star})$ as

$$\mathrm{KL}(\overleftarrow{\mathbb{P}}\,|\,\overleftarrow{\mathbb{P}}^{\star}) = \mathrm{KL}(\mu_{T_f}|\gamma^d) + \lambda\sum_{k=0}^{K-1}\mathbb{E}_{\overleftarrow{\mathbb{P}}}\left[\int_{[t_k,t_{k+1})}\sum_{\ell=1}^d\left(u_t^{\ell}\log u_t^{\ell} - u_t^{\ell}\log u_{t_k}^{\theta^{\star},\ell} - u_t^{\ell} + u_{t_k}^{\theta^{\star},\ell}\right)\mathrm{d}t\right].$$

We separate this expression into three terms to control:

$$\mathrm{KL}(\overleftarrow{\mathbb{P}}|\overleftarrow{\mathbb{P}}^\star) = \underbrace{\mathrm{KL}(\mu_{T_f}|\gamma^d)}_{E_1} + \lambda \underbrace{\sum_{k=0}^{K-1} \mathbb{E}_{\overleftarrow{\mathbb{P}}}\left[ \int_{[t_k,t_{k+1})} \sum_{\ell=1}^{d} \left( u_{t_k}^\ell \log u_{t_k}^\ell - u_t^\ell \log u_{t_k}^{\theta^\star,\ell} - u_{t_k}^\ell + u_{t_k}^{\theta^\star,\ell} \right) \mathrm{d}t \right]}_{E_2}$$

$$+ \lambda \underbrace{\sum_{k=0}^{K-1} \mathbb{E}_{\overleftarrow{\mathbb{P}}}\left[ \int_{[t_k,t_{k+1})} \sum_{\ell=1}^{d} \left( u_t^\ell \log u_t^\ell - u_t^\ell - u_{t_k}^\ell \log u_{t_k}^\ell + u_{t_k}^\ell \right) \mathrm{d}t \right]}_{E_3} . \tag{96}$$

We begin by observing that the uniform distribution $\gamma^d$ is the invariant measure of the forward process, which satisfies the curvature–dimension condition $\mathrm{CD}(2\lambda, \infty)$ (see Lemma F.23). As a consequence, it satisfies a logarithmic Sobolev inequality by Bakry et al. (2014, Theorem 5.10). This, in turn, implies exponential decay of entropy over time by Bakry et al. (2014, Theorem 5.12), and thus we obtain:

$$E_1 = \mathrm{KL}(\mu_{T_f}|\gamma^d) \leqslant \mathrm{e}^{-4\lambda T_f} \mathrm{KL}(\mu^\star|\gamma^d) . \tag{97}$$

Next, we bound $E_2$ using the tower property, the martingale property established in Proposition F.17, and the approximation error assumption in Assumption 2.1,

$$E_2 = \sum_{k=0}^{K-1} \mathbb{E}_{\overleftarrow{\mathbb{P}}}\left[ \int_{[t_k,t_{k+1})} \sum_{\ell=1}^{d} \left( u_{t_k}^\ell \log u_{t_k}^\ell - \mathbb{E}_{\overleftarrow{\mathbb{P}}}\left[ u_t^\ell | \mathcal{F}_{t_k} \right] \log u_{t_k}^{\theta^\star,\ell} - u_{t_k}^\ell + u_{t_k}^{\theta^\star,\ell} \right) \mathrm{d}t \right]$$

$$\overset{Proposition\ F.17}{=} \sum_{k=0}^{K-1} \mathbb{E}_{\overleftarrow{\mathbb{P}}}\left[ \int_{[t_k,t_{k+1})} \sum_{\ell=1}^{d} \left( u_{t_k}^\ell \log u_{t_k}^\ell - u_{t_k}^\ell \log u_{t_k}^{\theta^\star,\ell} - u_{t_k}^\ell + u_{t_k}^{\theta^\star,\ell} \right) \mathrm{d}t \right]$$

$$= \sum_{k=0}^{K-1} (t_{k+1} - t_k)\mathbb{E}_{\overleftarrow{\mathbb{P}}}\left[ \sum_{\ell=1}^{d} \left( u_{t_k}^\ell \log \frac{u_{t_k}^\ell}{u_{t_k}^{\theta^\star,\ell}} - u_{t_k}^\ell + u_{t_k}^{\theta^\star,\ell} \right) \right]$$

$$= \sum_{k=0}^{K-1} (t_{k+1} - t_k)\mathbb{E}_{\overleftarrow{\mathbb{P}}}\left[ \sum_{\ell=1}^{d} u_{t_k}^{\theta^\star,\ell} h\left( \frac{u_{t_k}^\ell}{u_{t_k}^{\theta^\star,\ell}} \right) \right] ,$$

where $h(a) = a \log a - a + 1$ for $a > 0$ and $(\mathcal{F}_t)_{t\in[0,T_f]}$ denotes the right-continuous and complete natural filtration generated by the process $(\overleftarrow{X}_t)_{t\in[0,T_f]}$. Using Assumption 2.1, we can now bound the term $E_2$ as

$$E_2 \leqslant \epsilon \sum_{k=0}^{K-1} (t_{k+1} - t_k) = \epsilon(t_K - t_0) = \epsilon T_f , \tag{98}$$

since $\sum_{k=0}^{K-1}(t_{k+1} - t_k)$ is a telescoping sum. It remains to control $E_3$. To do so, we leverage the monotonicity of the function $h$ established in Proposition F.17,

$$E_3 = \sum_{k=0}^{K-1} \mathbb{E}_{\overleftarrow{\mathbb{P}}} \left[ \int_{[t_k, t_{k+1})} \sum_{\ell=1}^{d} \left( \mathbb{E}_{\overleftarrow{\mathbb{P}}} \left[ h(u_t^\ell) | \mathcal{F}_{t_k} \right] - h(u_{t_k}^\ell) \right) \mathrm{d}t \right]$$

$$\leqslant \sum_{k=0}^{K-1} \mathbb{E}_{\overleftarrow{\mathbb{P}}} \left[ \int_{[t_k, t_{k+1})} \sum_{\ell=1}^{d} \left( \mathbb{E}_{\overleftarrow{\mathbb{P}}} \left[ h(u_{t_{k+1}}^\ell) \Big| \mathcal{F}_{t_k} \right] - h(u_{t_k}^\ell) \right) \mathrm{d}t \right]$$

$$= \sum_{k=0}^{K-1} (t_{k+1} - t_k) \left( \mathbb{E}_{\overleftarrow{\mathbb{P}}} \left[ \sum_{\ell=1}^{d} h(u_{t_{k+1}}^\ell) \right] - \mathbb{E}_{\overleftarrow{\mathbb{P}}} \left[ \sum_{\ell=1}^{d} h(u_{t_k}^\ell) \right] \right)$$

Define $\tau = \max\{t_{k+1} - t_k : \ k = 0, \ldots K - 1\}$, the calculation follows

$$E_3 \leqslant \tau \sum_{k=0}^{K-1} \left( \mathbb{E}_{\overleftarrow{\mathbb{P}}} \left[ \sum_{\ell=1}^{d} h(u_{t_{k+1}}^\ell) \right] - \mathbb{E}_{\overleftarrow{\mathbb{P}}} \left[ \sum_{\ell=1}^{d} h(u_{t_k}^\ell) \right] \right)$$

$$= \tau \left( \mathbb{E}_{\overleftarrow{\mathbb{P}}} \left[ \sum_{\ell=1}^{d} h(u_{T_f}^\ell) \right] - \mathbb{E}_{\overleftarrow{\mathbb{P}}} \left[ \sum_{\ell=1}^{d} h(u_0^\ell) \right] \right) \leqslant \tau \beta_{\gamma^d}(\mu^\star) , \tag{99}$$

since $h$ is non-negative. Here, the Fisher-like information functional $\beta_{\gamma^d}$ of the data distribution $\mu^\star$ is defined as $\beta_{\gamma^d}(\mu^\star) := \mathbb{E}_{\overleftarrow{\mathbb{P}}} \left[ \sum_{\ell=1}^{d} h(u_{T_f}^\ell) \right]$. Combining (97), (98) and (99), we arrive at

$$\mathrm{KL}(\overleftarrow{\mathbb{P}} | \overleftarrow{\mathbb{P}}^\star) \leqslant \mathrm{e}^{-4\lambda T_f} \mathrm{KL}(\mu^\star | \gamma^d) + \lambda \tau \beta_{\gamma^d}(\mu^\star) + \lambda \epsilon T_f . \tag{100}$$

Finally, noting that $\mu^* = \mathrm{Law}(\overleftarrow{X}_{T_f}^{u^*})$, we obtain the relation

$$\mathrm{KL}(\mu^\star | \mathrm{Law}(\overleftarrow{X}_{T_f}^\star)) = \mathrm{KL}(\mathrm{Law}(\overleftarrow{X}_{T_f}) | \mathrm{Law}(\overleftarrow{X}_{T_f}^\star)) \leqslant \mathrm{KL}(\mathrm{Law}((\overleftarrow{X}_t)_{t \in [0, T_f]}) | \mathrm{Law}((\overleftarrow{X}_t^\star)_{t \in [0, T_f]})) = \mathrm{KL}(\overleftarrow{\mathbb{P}} | \overleftarrow{\mathbb{P}}^\star) ,$$

where the inequality is known as *Data processing* inequality for KL divergence (Nutz, 2021, Lemma 1.6). Combining this with (100), we conclude that

$$\mathrm{KL}(\mu^\star | \mathrm{Law}(\overleftarrow{X}_{T_f}^\star)) \leqslant \mathrm{e}^{-4\lambda T_f} \mathrm{KL}(\mu^\star | \gamma^d) + \lambda \tau \beta_{\gamma^d}(\mu^\star) + \lambda \epsilon T_f ,$$

and successfully provide a theoretical guarantee for our generative models. $\qquad \square$

### F.5.2. PROOF OF THEOREM 2.4

We demonstrate Theorem 2.4 through the following result:

**Lemma F.24.** *Denoting* $y_t = \mathbb{E}_{\overleftarrow{\mathbb{P}}} \left[ \sum_{\ell=1}^{d} h(u_t(\overleftarrow{X}_t, \varphi^{(\ell)}(\overleftarrow{X}_t)) \right]$ *then it holds for* $t \in [0, T_f)$,

$$y_t \lesssim \frac{d}{T_f - t} . \tag{101}$$

***Proof of Lemma F.24.*** Recall the definition of $h(a) = a \log a - a + 1$ for $a > 0$, and the connection between the optimal control $u_t$ and the score function $s_t$, as provided by Proposition F.18, is given by

$$u_t(x, \varphi^{(\ell)}(x)) = 1 - s_t^\ell(x) \quad \text{for } \ell = 1, \ldots, d \text{ and } (t, x) \in [0, T_f) \times \mathsf{X} ,$$

with the score function admitting a conditional expectation expression as given in (18),

$$s_t^\ell(x) = \mathbb{E}\left[\frac{2\alpha_{T_f-t}}{1+\alpha_{T_f-t}} - \frac{4\alpha_{T_f-t}(\overrightarrow{X}_{T_f-t}^\ell - \overrightarrow{X}_0^\ell)^2}{1-\alpha_{T_f-t}^2}\bigg|\overrightarrow{X}_{T_f-t} = x\right] , \quad \text{with } \alpha_t = \mathrm{e}^{-2t} .$$

Using the above formulation of the score function, we estimate $y_t$ as follows

$$
\begin{aligned}
y_t &= \mathbb{E}_{\overleftarrow{\mathbb{P}}}\left[\sum_{\ell=1}^d (1 - s_t^\ell(\overleftarrow{X}_t))\log(1 - s_t^\ell(\overleftarrow{X}_t)) + s_t^\ell(\overleftarrow{X}_t)\right] \\
&\leqslant \mathbb{E}_{\overleftarrow{\mathbb{P}}}\left[\sum_{\ell=1}^d (1 - s_t^\ell(\overleftarrow{X}_t))(1 - s_t^\ell(\overleftarrow{X}_t) - 1) + s_t^\ell(\overleftarrow{X}_t)\right] \quad (\text{since } \log a \leqslant a - 1) \\
&= \mathbb{E}_{\overleftarrow{\mathbb{P}}}\left[\sum_{\ell=1}^d (s_t^\ell(\overleftarrow{X}_t))^2\right] \\
&= \sum_{\ell=1}^d \mathbb{E}\left[\left(\mathbb{E}\left[\frac{2\alpha_{T_f-t}}{1+\alpha_{T_f-t}} - \frac{4\alpha_{T_f-t}(\overrightarrow{X}_{T_f-t}^\ell - \overrightarrow{X}_0^\ell)^2}{1-\alpha_{T_f-t}^2}\bigg|\overrightarrow{X}_{T_f-t} = x\right]\right)^2\right] \\
&\leqslant \sum_{\ell=1}^d \mathbb{E}\left[\mathbb{E}\left[\left(\frac{2\alpha_{T_f-t}}{1+\alpha_{T_f-t}} - \frac{4\alpha_{T_f-t}(\overrightarrow{X}_{T_f-t}^\ell - \overrightarrow{X}_0^\ell)^2}{1-\alpha_{T_f-t}^2}\right)^2\bigg|\overrightarrow{X}_{T_f-t} = x\right]\right] \\
&= \sum_{\ell=1}^d \mathbb{E}\left[\left(\frac{2\alpha_{T_f-t}}{1+\alpha_{T_f-t}} - \frac{4\alpha_{T_f-t}(\overrightarrow{X}_{T_f-t}^\ell - \overrightarrow{X}_0^\ell)^2}{1-\alpha_{T_f-t}^2}\right)^2\right] .
\end{aligned}
$$

Expanding the last quantity and noting that $(\overrightarrow{X}_{T_f-t}^\ell - \overrightarrow{X}_0^\ell)^2 = (\overrightarrow{X}_{T_f-t}^\ell - \overrightarrow{X}_0^\ell)^4$ since $(\overrightarrow{X}_{T_f-t}^\ell - \overrightarrow{X}_0^\ell) \in \{0, \pm 1\}$, we obtain that

$$
\begin{aligned}
y_t &\leqslant \sum_{\ell=1}^d \mathbb{E}\left[\frac{4\alpha_{T_f-t}^2}{(1+\alpha_{T_f-t})^2} + \frac{16\alpha_{T_f-t}^2(\overrightarrow{X}_{T_f-t}^\ell - \overrightarrow{X}_0^\ell)^2}{(1+\alpha_{T_f-t})(1-\alpha_{T_f-t}^2)}\left(-1 + \frac{1}{1-\alpha_{T_f-t}}\right)\right] \\
&= \sum_{\ell=1}^d \mathbb{E}\left[\frac{4\alpha_{T_f-t}^2}{(1+\alpha_{T_f-t})^2} + \frac{16\alpha_{T_f-t}^3(\overrightarrow{X}_{T_f-t}^\ell - \overrightarrow{X}_0^\ell)^2}{(1-\alpha_{T_f-t}^2)^2}\right] \\
&\leqslant \sum_{\ell=1}^d \left(\frac{4\alpha_{T_f-t}^2}{(1+\alpha_{T_f-t})^2} + \frac{16\alpha_{T_f-t}^3\mathbb{E}\left[(\overrightarrow{X}_{T_f-t}^\ell - \overrightarrow{X}_0^\ell)^2\right]}{(1-\alpha_{T_f-t}^2)^2}\right) .
\end{aligned}
$$

Note that

$$\mathbb{E}\left[(\overrightarrow{X}_{T_f-t}^\ell - \overrightarrow{X}_0^\ell)^2\right] = \mathbb{P}\left(\overrightarrow{X}_{T_f-t}^\ell \neq \overrightarrow{X}_0^\ell\right) = \frac{1}{2}(1 - \alpha_{T_f-t}) .$$

Thus the upper bound of $y_t$ is

$$y_t \leqslant \sum_{\ell=1}^{d} \left( \frac{4\alpha_{T_f-t}^2}{(1+\alpha_{T_f-t})^2} + \frac{8\alpha_{T_f-t}^3(1-\alpha_{T_f-t})}{(1-\alpha_{T_f-t}^2)^2} \right)$$

$$= \frac{4d\alpha_{T_f-t}^2}{(1+\alpha_{T_f-t})^2} \left[ 1 + \frac{2\alpha_{T_f-t}}{(1-\alpha_{T_f-t})} \right]$$

$$= \frac{4d\alpha_{T_f-t}^2}{1-\alpha_{T_f-t}^2} \lesssim \frac{d}{\mathrm{e}^{4(T_f-t)}-1} \lesssim \frac{d}{T_f-t} \ , \quad \text{for } t \in [0, T_f) \ ,$$

where the last estimate follows from the elementary inequality $\mathrm{e}^a \geqslant a+1$ for all $a \in \mathbb{R}$. Therefore, the bound in (101) holds for all $t \in [0, T_f)$.

$\square$

***Proof of Theorem 2.4.*** We follow the same strategy as in the proof of Theorem 2.3, the only difference being the way we handle the following term in (99)

$$E_3 := \sum_{k=0}^{K-1} (t_{k+1} - t_k) \left( \mathbb{E}_{\overleftarrow{\mathbb{P}}} \left[ \sum_{\ell=1}^{d} h(u_{t_{k+1}}) \right] - \mathbb{E}_{\overleftarrow{\mathbb{P}}} \left[ \sum_{\ell=1}^{d} h(u_{t_k}) \right] \right) = \sum_{k=0}^{K-1} \tau_{k+1} \left( y_{t_{k+1}} - y_{t_k} \right) \ .$$

Following precisely the argument structure in the proof of Theorem 3 from Conforti et al. (2025), and fixing $T_f$, $a$, and $c$, we choose the sequence of step-size as

$$\tau_{k+1} = \begin{cases} T_f - t_{N-1} \ , & k = N-1 \ , \\ ca \ , & k_0 + k_1 \leqslant k \leqslant k_0 + k_1 + k_2 - 1 \ , \\ c(T_f - t_k) \ , & k_0 \leqslant k \leqslant k_0 + k_1 - 1 \ , \\ c \ , & 0 \leqslant k \leqslant k_0 - 1 \ , \end{cases} \tag{102}$$

and set the number of iterations $K = k_0 + k_1 + k_2 + 1$, with

$$k_0 = \max \{k \geqslant 0 : T_f - t_k \geqslant 1\}, k_1 = \max \{k \geqslant 0 : T_f - t_{k_0+k} \geqslant a\} \text{ and } k_2 = \max \{k \geqslant 0 : T_f - t_{k_0+k_1+k} \geqslant 0\} \ . \tag{103}$$

It is shown in Conforti et al. (2025) that

$$k_0 = \lfloor c^{-1}(T_f - 1) \rfloor, \quad k_1 = \lfloor \log(a/(T_f - t_{k_0}))/\log(1-c) \rfloor \lesssim \log(1/a)/c \ ,$$
$$K - k_0 - k_1 = k_2 + 1 \lesssim 1/c \quad \text{and} \quad \tau_{k+1} = c(1-c)^{k-k_0}(T_f - t_{k_0}) \quad \text{for } k \in \{k_0, \ldots, k_0 + k_1 - 1\} \ . \tag{104}$$

Using (102) and the monotonicity of $y_t$ established in Proposition F.17, we can bound $E_3$ as

$$E_3 \leqslant \sum_{k=0}^{K-1} \tau_{k+1} \left( y_{t_{k+1}} - y_{t_k} \right)$$

$$= \tau_K y_{t_K} + \sum_{k=1}^{K-1} y_{t_k} (\tau_k - \tau_{k+1})$$

$$= \tau_K y_{t_K} + \sum_{k=1}^{k_0} y_{t_k} (\tau_k - \tau_{k+1}) + \sum_{k=k_0+1}^{k_0+k_1} y_{t_k} (\tau_k - \tau_{k+1})$$

$$+ \sum_{k=k_0+k_1+1}^{k_0+k_1+k_2-1} y_{t_k} (\tau_k - \tau_{k+1}) + y_{t_{k_0+k_1+k_2}} (\tau_{K-1} - \tau_K)$$

$$\lesssim \underbrace{y_{t_{k_0}} [c - c(T_f - t_{k_0})]}_{(1)} + \underbrace{c \sum_{k=k_0+1}^{k_0+k_1-1} y_{t_k} \tau_k}_{(2)}$$

$$+ \underbrace{c y_{t_{k_0+k_1}} (T_f - t_{k_0+k_1-1} - a)}_{(3)} + \underbrace{y_{t_K} \tau_{K-1}}_{(4)} .$$

We now bound $(1) - (2) - (3) - (4)$ one-by-one. We start with

$$(1) : y_{t_{k_0}} [c - c(T_f - t_{k_0})] \leqslant c y_{t_{k_0}} \overset{Lemma\ F.24}{\lesssim} \frac{cd}{T_f - t_{k_0}} \overset{(103)}{\leqslant} cd .$$

Next, we bound the second term

$$(2) : c \sum_{k=k_0+1}^{k_0+k_1-1} y_{t_k} \tau_k \overset{Lemma\ F.24}{\lesssim} c \sum_{k=k_0+1}^{k_0+k_1-1} \frac{d\tau_k}{T_f - t_k} = c^2 d \sum_{k=k_0+1}^{k_0+k_1-1} \frac{\tau_k}{\tau_{k+1}}$$

$$\overset{(104)}{=} c^2 d \sum_{k=k_0+1}^{k_0+k_1-1} \frac{c(1-c)^{k-k_0-1}(T_f - t_{k_0})}{c(1-c)^{k-k_0}(T_f - t_{k_0})}$$

$$\leqslant c^2 d \sum_{k=k_0+1}^{k_0+k_1-1} \frac{1}{1-c} \overset{c \leqslant 1/2}{\lesssim} c^2 d k_1 \lesssim cd \log(1/a) .$$

Recall that $T_f - t_{k_0+k_1-1} \leqslant 1$. As a result, we have

$$(3) : c y_{t_{k_0+k_1}} (T_f - t_{k_0+k_1-1} - a) \overset{Lemma\ F.24}{\lesssim} \frac{cd}{T_f - t_{k_0+k_1}} (T_f - t_{k_0+k_1-1}) = \frac{cd}{T_f - t_{k_0+k_1-1} - \tau_{k_0+k_1}} (T_f - t_{k_0+k_1-1})$$

$$\overset{(102)}{\leqslant} \frac{cd}{1-c} \overset{c \leqslant 1/2}{\lesssim} cd .$$

Finally, for the last term, we have by definition of $L = y_{T_f}/d$,

$$(4) : y_{t_K} h_{K-1} = y_{T_f} ca = cadL .$$

Plugging all the bounds of $(1) - (2) - (3) - (4)$ into $E_3$ gives

$$E_3 \lesssim cd[1 + \log(1/a) + aL] .$$

Moreover, choosing $a = 1/L$ yields the following bound on the sampling error

$$\mathrm{KL}(\mu^\star | \mathrm{Law}(\overleftarrow{X}^\star_{T_f})) \lesssim \mathrm{e}^{-4\lambda T_f} \mathrm{KL}(\mu^\star | \gamma^d) + \lambda c d[1 + \log(L)] + \lambda \epsilon T_f \, ,$$

and this concludes the proof of Theorem 2.4.

$\square$

### F.5.3. PROOF OF COROLLARY 2.5

**Proof of Corollary 2.5.** Applying Theorem 2.4 with $c, T_f$ chosen in (32) implies

$$\mathrm{KL}(\mu^\star | \mathrm{Law}(\overleftarrow{X}^\star_{T_f})) \lesssim \epsilon + \epsilon \log \frac{\mathrm{KL}(\mu^\star | \gamma^d)}{\epsilon} \, .$$

Therefore we obtain the approximation error $\tilde{O}(\epsilon \log(\mathrm{KL}(\mu^\star | \gamma^d)))$. In addition, the number of iterations is given by

$$
\begin{aligned}
K = k_0 + k_1 + K - k_0 - k_1 &\overset{(104)}{\lesssim} \frac{T_f - 1}{c} + \frac{\log(1/a)}{c} + \frac{1}{c} = \frac{T_f + \log(L)}{c} \\
&\overset{(32)}{=} \frac{\lambda d[1 + \log(L)][\log(\mathrm{KL}(\mu^\star | \gamma^d)/\epsilon)/\lambda + \log(L)]}{\epsilon} \\
&= \frac{d[1 + \log(L)][\log(\mathrm{KL}(\mu^\star | \gamma^d)/\epsilon) + \lambda \log(L)]}{\epsilon} \, ,
\end{aligned}
$$

which grows logarithmically rather than linearly with respect to the discrete Fisher information $\beta_{\gamma^d}(\mu^\star)$. We thus conclude that this step-size sequence offers improved performance compared to the constant step-size.

$\square$

## F.6. Convergence of DMPMs with early stopping strategy

### F.6.1. PROOF OF THEOREM 2.6

**Proof of Theorem 2.6.** Applying the proof of Theorem 2.4 with the target distribution $\mu_\eta$ in place of $\mu^*$ leads to the following analogous result:

$$\mathrm{KL}(\mu_\eta | \mathrm{Law}(\overleftarrow{X}^\star_{T_f - \eta})) \leqslant \mathrm{e}^{-4\lambda(T_f - \eta)} \mathrm{KL}(\mu_\eta | \gamma^d) + \lambda c d[1 + \log(L)] + \lambda \epsilon (T_f - \eta) \, . \tag{105}$$

Recall that the forward dynamic satisfies the curvature–dimension condition $\mathrm{CD}(2\lambda, \infty)$ (see Lemma F.23) and consequently, the logarithm Sobolev inequality holds (Bakry et al., 2014, Theorem 5.10):

$$\mathrm{KL}(\mu_\eta | \gamma^d) \leqslant \frac{1}{4\lambda} \mathcal{E}\left(\frac{\mu_\eta}{\gamma^d}, \log \frac{\mu_\eta}{\gamma^d}\right) \, . \tag{106}$$

As computed in Lemma F.23, we have

$$
\begin{aligned}
\mathcal{E}\left(\frac{\mu_\eta}{\gamma^d}, \log \frac{\mu_\eta}{\gamma^d}\right) &= \lambda \sum_{x \in \mathsf{X}} \sum_{\ell=1}^d \frac{\mu_\eta}{\gamma^d}(x) \left(\log \frac{\mu_\eta}{\gamma^d}(x) - \log \frac{\mu_\eta}{\gamma^d}(\varphi^{(\ell)}(x))\right) \gamma^d(x) \\
&= \lambda \mathbb{E}\left[\sum_{\ell=1}^d \left(\log \frac{\mu_\eta(\overrightarrow{X}_\eta)}{\mu_\eta(\varphi^{(\ell)}(\overrightarrow{X}_\eta))}\right)\right] \, . \tag{107}
\end{aligned}
$$

On the other hand, the discrete Fisher information of $\mu_\eta$ is given by

$$
\begin{aligned}
\beta_{\gamma^d}(\mu_\eta) &= \mathbb{E}\left[\sum_{\ell=1}^{d} h\left(e^{-\log\left(\frac{\mu_\eta}{\gamma^d}(\overrightarrow{X}_\eta)\right)+\log\left(\frac{\mu_\eta}{\gamma^d}(\varphi^{(\ell)}(\overrightarrow{X}_\eta))\right)}\right)\right] \\
&= \mathbb{E}\left[\sum_{\ell=1}^{d} h\left(e^{\log\left(\frac{\mu_\eta(\varphi^{(\ell)}(\overrightarrow{X}_\eta))}{\mu_\eta(\overrightarrow{X}_\eta)}\right)}\right)\right] \\
&= \mathbb{E}\left[\sum_{\ell=1}^{d} h\left(\frac{\mu_\eta(\varphi^{(\ell)}(\overrightarrow{X}_\eta))}{\mu_\eta(\overrightarrow{X}_\eta)}\right)\right] \\
&= \mathbb{E}\left[\sum_{\ell=1}^{d}\left(\frac{\mu_\eta(\varphi^{(\ell)}(\overrightarrow{X}_\eta))}{\mu_\eta(\overrightarrow{X}_\eta)}\log\frac{\mu_\eta(\varphi^{(\ell)}(\overrightarrow{X}_\eta))}{\mu_\eta(\overrightarrow{X}_\eta)} - \frac{\mu_\eta(\varphi^{(\ell)}(\overrightarrow{X}_\eta))}{\mu_\eta(\overrightarrow{X}_\eta)} + 1\right)\right] \\
&= \sum_{x\in\mathsf{X}}\sum_{\ell=1}^{d}\left(\mu_\eta(\varphi^{(\ell)}(x))\log\frac{\mu_\eta(\varphi^{(\ell)}(x))}{\mu_\eta(x)} - \mu_\eta(\varphi^{(\ell)}(x)) + \mu_\eta(x)\right) \\
&= \sum_{x\in\mathsf{X}}\sum_{\ell=1}^{d}\left(\mu_\eta(\varphi^{(\ell)}(x))\log\frac{\mu_\eta(\varphi^{(\ell)}(x))}{\mu_\eta(x)}\right) \\
&= \sum_{x\in\mathsf{X}}\sum_{\ell=1}^{d}\left(\mu_\eta(x)\log\frac{\mu_\eta(x)}{\mu_\eta(\varphi^{(\ell)}(x))}\right) \\
&= \mathbb{E}\left[\sum_{\ell=1}^{d}\left(\log\frac{\mu_\eta(\overrightarrow{X}_\eta)}{\mu_\eta(\varphi^{(\ell)}(\overrightarrow{X}_\eta))}\right)\right] \overset{(107)}{=} \frac{1}{\lambda}\mathcal{E}\left(\frac{\mu_\eta}{\gamma^d}, \log\frac{\mu_\eta}{\gamma^d}\right) .
\end{aligned}
$$

Plugging it into (106) implies that the discrete Fisher information dominates the KL divergence as

$$
\mathrm{KL}(\mu_\eta|\gamma^d) \leqslant \frac{1}{4}\beta_{\gamma^d}(\mu_\eta) .
$$

To complete the proof, it remains to bound the discrete Fisher information. To this end, we employ the elementary inequality $\log a \leqslant a - 1$ for $a > 0$, and proceed using the same reasoning as in Lemma F.24, as detailed below,

$$
\begin{aligned}
\beta_{\gamma^d}(\mu_\eta) &= \mathbb{E}\left[\sum_{\ell=1}^{d}\left(\frac{\mu_\eta(\varphi^{(\ell)}(\overrightarrow{X}_\eta))}{\mu_\eta(\overrightarrow{X}_\eta)}\log\frac{\mu_\eta(\varphi^{(\ell)}(\overrightarrow{X}_\eta))}{\mu_\eta(\overrightarrow{X}_\eta)} - \frac{\mu_\eta(\varphi^{(\ell)}(\overrightarrow{X}_\eta))}{\mu_\eta(\overrightarrow{X}_\eta)} + 1\right)\right] \\
&\leqslant \mathbb{E}\left[\sum_{\ell=1}^{d}\left(\frac{\mu_\eta(\varphi^{(\ell)}(\overrightarrow{X}_\eta))}{\mu_\eta(\overrightarrow{X}_\eta)}\left(\frac{\mu_\eta(\varphi^{(\ell)}(\overrightarrow{X}_\eta))}{\mu_\eta(\overrightarrow{X}_\eta)} - 1\right) - \frac{\mu_\eta(\varphi^{(\ell)}(\overrightarrow{X}_\eta))}{\mu_\eta(\overrightarrow{X}_\eta)} + 1\right)\right] \\
&= \mathbb{E}\left[\sum_{\ell=1}^{d}\left(\frac{\mu_\eta(\varphi^{(\ell)}(\overrightarrow{X}_\eta))}{\mu_\eta(\overrightarrow{X}_\eta)} - 1\right)^2\right] \\
&= \mathbb{E}\left[\sum_{\ell=1}^{d}\left(s_{T_f-\eta}^{\ell}(\overrightarrow{X}_\eta)\right)^2\right] \lesssim \frac{d}{T_f - (T_f - \eta)} = \frac{d}{\eta} .
\end{aligned}
$$

It follows that

$$
\mathrm{KL}(\mu_\eta|\gamma^d) \lesssim \frac{d}{\eta} \quad \text{and} \quad L = d^{-1}\beta_{\gamma^d}(\mu_\eta) \lesssim \eta^{-1} .
$$

Combined with (105), this leads to the desired conclusion:

$$\mathrm{KL}(\mu_\eta | \mathrm{Law}(\overleftarrow{X}^\star_{T_f - \eta})) \lesssim d\eta^{-1}\mathrm{e}^{-4\lambda(T_f - \eta)} + \lambda cd[1 + \log(\eta^{-1})] + \lambda\epsilon(T_f - \eta) .$$

$\square$

### F.6.2. PROOF OF PROPOSITION 2.7

***Proof of Proposition 2.7***. Recall that the total variation distance between $\mu_\eta$ and $\mu^*$ for $\eta \in (0, \max\{T_f, \frac{1}{\lambda}\})$ is defined as

$$\|\mu_\eta - \mu^\star\|_{\mathrm{TV}} = \sum_{x \in \mathsf{X}} |\mu_\eta(x) - \mu^\star(x)| .$$

By the triangle inequality, we obtain

$$\|\mu_\eta - \mu^\star\|_{\mathrm{TV}} \leqslant \sum_{x \in \mathsf{X}} |\mu_\eta(x) - \mu^\star(x)\overrightarrow{p}_\eta(x, x)| + |\mu^\star(x) - \mu^\star(x)\overrightarrow{p}_\eta(x, x)| ,$$

where the transition probability $\overrightarrow{p}_\eta$ is defined in (15). The two terms above are non-negative as

$$\mu_\eta(x) = \sum_{z \in \mathsf{X}} \mu^\star(z)\overrightarrow{p}_\eta(z, x) \geqslant \mu^\star(x)\overrightarrow{p}_\eta(x, x) ,$$

and the transition probability $\overrightarrow{p}_\eta(x, x) \leqslant 1$ for any $x \in \mathsf{X}$. This together with the formula of $\overrightarrow{p}_\eta$ in (15) yield

$$\|\mu_\eta - \mu^\star\|_{\mathrm{TV}} \leqslant \sum_{x \in \mathsf{X}} \left[ \mu_\eta(x) + \mu^\star(x) - 2\mu^\star(x)\left(\frac{1}{2} + \frac{1}{2}\mathrm{e}^{-2\lambda\eta}\right)^d \right]$$
$$\leqslant 2 - 2\left(\frac{1}{2} + \frac{1}{2}\mathrm{e}^{-2\lambda\eta}\right)^d .$$

To simplify this upper bound, we apply the exponential inequality $\mathrm{e}^a \geqslant a + 1$ to $\mathrm{e}^{-2\lambda\eta}$ and note that $1 - \lambda\eta > 0$:

$$\|\mu_\eta - \mu^\star\|_{\mathrm{TV}} \leqslant 2 - 2\left(\frac{1}{2} + \frac{1}{2}(-2\lambda\eta + 1)\right)^d = 2 - 2(1 - \lambda\eta)^d ,$$

and thus the proof is complete.

$\square$

### F.6.3. PROOF OF COROLLARY 2.8

***Proof of Corollary 2.8***. We observe the following by applying the triangle inequality and Pinsker's inequality,

$$\|\mu^\star - \mathrm{Law}(\overleftarrow{X}^\star_{T_f - \eta})\|_{\mathrm{TV}} \leqslant \|\mu^\star - \mu_\eta\|_{\mathrm{TV}} + \|\mu_\eta - \mathrm{Law}(\overleftarrow{X}^\star_{T_f - \eta})\|_{\mathrm{TV}}$$
$$\leqslant \|\mu^\star - \mu_\eta\|_{\mathrm{TV}} + \sqrt{2\mathrm{KL}(\mu_\eta | \mathrm{Law}(\overleftarrow{X}^\star_{T_f - \eta}))} .$$

Then Theorem 2.6 and Proposition 2.7 together imply

$$\|\mu^\star - \mathrm{Law}(\overleftarrow{X}^\star_{T_f - \eta})\|_{\mathrm{TV}} \lesssim 1 - (1 - \lambda\eta)^d + \sqrt{d\eta^{-1}\mathrm{e}^{-4\lambda(T_f - \eta)}} + \sqrt{\lambda cd[1 + \log(\eta^{-1})]} + \sqrt{\lambda\epsilon(T_f - \eta)} . \quad (108)$$

The choices of $\eta, c$ and $T_f$ in (35) lead to

$$1 - (1 - \lambda\eta)^d = \epsilon \quad \text{and} \quad d\eta^{-1}\mathrm{e}^{-4\lambda(T_f - \eta)} = \epsilon^2 \quad \text{and} \quad \lambda cd[1 + \log(\eta^{-1})] = \epsilon^2 . \quad (109)$$

Substituting (109) into (108) gives the desired upper bound

$$\|\mu^\star - \mathrm{Law}(\overleftarrow{X}^\star_{T_f - \eta})\|_{\mathrm{TV}} \lesssim \epsilon + \sqrt{\lambda \epsilon (T_f - \eta)} \,,$$

with the number of iterations is

$$K \lesssim \frac{T_f - \eta + \log(\eta^{-1})}{c} \overset{(35)}{=} \frac{\lambda d[1 + \log(\eta^{-1})][\log(d/\eta\epsilon^2)/\lambda + \log(\eta^{-1})]}{\epsilon^2}$$
$$= \frac{d[1 + \log(\eta^{-1})][\log(d/\eta\epsilon^2) + \lambda \log(\eta^{-1})]}{\epsilon^2}$$
$$= \frac{d[1 + \log(\eta^{-1})][\log(d/\epsilon^2) + (\lambda + 1) \log(\eta^{-1})]}{\epsilon^2} \,. \tag{110}$$

Finally, the term $\log \eta^{-1}$ can be bounded from above by Bernoulli's inequality $(1 - \epsilon)^{1/d} \leqslant 1 - \frac{\epsilon}{d}$ for $\epsilon \in (0, 1)$ and $d \geqslant 1$,

$$\log(\eta^{-1}) = \log\left(\frac{\lambda}{1 - (1 - \epsilon)^{1/d}}\right) \leqslant \log \frac{\lambda d}{\epsilon} \,.$$

Plugging it into (110) concludes the proof of Corollary 2.8.

$\square$

