# OpenReview forum: "Discrete Markov Probabilistic Models: An Improved Discrete Score-Based  Framework with sharp convergence bounds under minimal assumptions"
_ICML.cc/2025/Conference — ICML 2025 poster_

### Official Review · Reviewer_9yNo · 2025-03-10

**Overall Recommendation:** 3

**Summary:**

The authors propose the Discrete Markov Probabilistic Model (DMPM), a novel algorithm for generative modeling on the binary data space (bits). The generative model is based on a continuous-time Markov Chain (CTMC) framework with forward processes, which “noisses” the data, and backward processes, which “denoises” the data. The forward process is constructed as a homogeneous Poissonian process that flips the data bits and converges to uniform bits distribution at infinite time. The backward process is modeled as a reverse of the forward process, i.e., via a time-inhomogeneous Markov jump process, which is defined as a discrete analog of the score function.

During the training process, a discrete analog of the score function is learned by a neural network. The authors propose three loss functions and their linear combinations. Then, if the score function is properly learned, one can sample from the backward process to generate the data.

The authors analyze DMPM algorithm consistency by deriving bounds for KL and TV between initial data distribution and the resulting distribution of backward generation processes.

The proposed algorithm is being compared with state-of-the-art discrete state space Markov chain generative models, MD4 [1] and DFM [2]. The experiments on two datasets were shown: synthetic small dimensional Bernoulli distribution data and black-white MNIST dataset.

## Update after rebuttal
The authors mostly addressed my concerns. Since the method is focused on working with the binary data, I think that additional demonstration of its superiority on a real, naturally binary dataset would significantly strengthen the work. As well as the better text clarity to simplify understanding for the wider community.

**Claims And Evidence:**

The authors argue that the DMPM model matches or outperforms other state-of-the-art Markov chain discrete generative models. However, it seems that the experimental evidence is insufficient to draw such a conclusion.

The MNIST dataset is too simple and too low-dimensional to judge the generative model performance. Other models (MD4 and DFM) are mainly validated on the text data or much more complex image datasets such as CIFAR-10 or ImageNet 64x64, hence experiments presented in the paper are insufficient.

There do not appear to be widely recognized large-scale binary datasets that clearly necessitate the advanced CTMC-based approach. Hence, the need for such a powerful model for generative modeling on binary data is not obvious.

Theoretical claims are supported by the derivations in the Appendix.

**Essential References Not Discussed:**

Most of the essential literature is discussed with the exception of the MDLM [5] model.

**Experimental Designs Or Analyses:**

Other than the experimental problems that were already presented in the upper sections I think the paper lacks insufficient hyperparameters testing.

- Time horizon $T_f$ is tested only in the synthetic Bernoulli distribution experiment. In my opinion, more testing regarding time horizon $T_f$ is needed. One can just analyze the forward process resulting distribution and its closeness with uniform distribution without training a diffusion model.
- I have not found any analysis regarding forward process jump rate $\lambda$ and its impact on forward process convergence and performance of learned generative models.
- The visualization of the forward process with dependence on time horizon $T_f$ and jump rate $\lambda$ would be beneficial for the paper
- DMPM is the model that has uniform distribution as the resulting distribution of the forward process. Other tested competitors do have delta distribution as the resulting distribution of the forward process. It would be beneficial to compare with other state-of-the-art markov chain discrete generative models that have uniform distribution as the resulting distribution of the forward process, such as SEDD Uniform [4].

**Methods And Evaluation Criteria:**

As mentioned in the previous section, the datasets chosen for empirical evaluation suit the task solved but seems to be too simple.

The FID metric for MNIST is somewhat questionable since the Inception network (whose features are used to compute FID) was trained on ImageNet rather than on synthetic or binary data.

Competitor methods are suitable and are indeed state-of-the-art.

**Other Comments Or Suggestions:**

[1] Shi, J., Han, K., Wang, Z., Doucet, A., & Titsias, M. K. (2024). Simplified and Generalized Masked Diffusion for Discrete Data. arXiv preprint arXiv:2406.04329.

[2] Gat, I., Remez, T., Shaul, N., Kreuk, F., Chen, R. T., Synnaeve, G., ... & Lipman, Y. (2025). Discrete flow matching. Advances in Neural Information Processing Systems, 37, 133345-133385.

[3] Campbell, A., Benton, J., De Bortoli, V., Rainforth, T., Deligiannidis, G., & Doucet, A. (2022). A continuous time framework for discrete denoising models. Advances in Neural Information Processing Systems, 35, 28266-28279.

[4] Lou, A., Meng, C., & Ermon, S. Discrete Diffusion Modeling by Estimating the Ratios of the Data Distribution. In Forty-first International Conference on Machine Learning.

[5] Sahoo, S. S., Arriola, M., Gokaslan, A., Marroquin, E. M., Rush, A. M., Schiff, Y., ... & Kuleshov, V. Simple and Effective Masked Diffusion Language Models. In ICML 2024 Workshop on Efficient and Accessible Foundation Models for Biological Discovery.

**Other Strengths And Weaknesses:**

The paper could be written in a more comprehensive/clear way. Notation is overloaded, and mathematics is sometimes hard to understand. It seems that the Poissonian processes framework could be delivered in a more simple and clear way.

In addition, some parts are generally hard to understand. For example, lines 133-144 are hard to follow even after taking into account similar derivations in the Appendix.

It seems that there is a lot of content in the appendix, and it may be beneficial to move some of it to the main text.

**Questions For Authors:**

- It would be great if authors could address the limitations I mentioned in the “Experimental Designs Or Analyses” section.

**Relation To Broader Scientific Literature:**

The authors discuss related literature in a sufficient way.

Overall the proposed method can be viewed as a particular case of discrete state space CTMC [3, 4] generative model with uniform forward process resulting distribution. In that sense paper doesn't deliver that much of a novelty. In my opinion, there is hardly a need for such a powerful model for binary data.

Section 2 provides most of the theoretical novelty for the paper but in my opinion these theoretical results do not deliver impact sufficient  for the publication of the paper.

**Theoretical Claims:**

The derivations in Sections 1.1 and 1.2 have been checked through following the extended derivations in the appendix.

---

> ### Author Rebuttal · Authors · 2025-03-31
>
> We thank the reviewer for their thoughtful remarks and the opportunity to clarify the significance of our theoretical results. Our primary aim is to establish convergence guarantees for discrete generative models, analogous to those in continuous diffusion. This led us to a discrete score function with two key formulations (Propositions 1.1 and 1.2), from which our methodology naturally follows—culminating in strong convergence results, arguably our chief contribution.
>
> As emphasized in our paper, there is a notable lack of results concerning discrete diffusion models. The closest related work, which improves upon ours in several respects (as detailed in our response to Reviewer fUbX), has only recently been accepted to ICLR [1].
>
> While we include experiments to illustrate the method’s potential, we agree with the reviewer that they should be viewed as preliminary—particularly for large-scale or high-dimensional data. We will revise the paper’s conclusion to emphasize that these findings are intended to motivate and guide future empirical studies, not to be a conclusive benchmark.
>
> [1] Ren et al, ICLR 2025
>
>
> > [no] widely recognized large-scale binary datasets...
>
> We emphasize the hypercube setting’s relevance. First, many real-world large-scale datasets use binary data (e.g., millions of presence/absence edges in amazon products or research papers see [2]), enabling applications like link repair, synthetic-data generation for privacy, or also molecule design. Second, any discrete dataset can be binarized (e.g., 8-bit pixel encoding), and recent developments such as Differentiable Logic Gates Networks [3] show how bit-based methods can excel with neural networks, with the right engineering. We belive our framework provides a strong foundation for bit-based generative models.
>
> [2] Hu et al. 2020, Open Graph Benchmark, Neurips
>
> [3] Petersen et al. 2022, Neurips
>
> > [...] particular case of discrete state space CTMCs...
>
> Our method indeed fits within the class of discrete-state CTMCs. Its novelty stems from exploiting the hypercube in a more specialized way than the general framework in [4]. We can introduce efficient optimization strategies, mirroring approaches like MDLM with the SUBS parameterization or MD4, which leverage the structure of the absorbing kernel.
>
> Like SEDD [5], we parameterize a ratio of density $\mu_t(y) / \mu_t(x)$ rather than a reverse density $p_{0 |t}$, which is easier since it is a general value rather than a proper distribution. Furthermore, our objective function is expressed in continuous time, contrary to the loss in [4] which must be extended from discrete time and approximated.
>
> Contrary to SEDD, our backward process is characterized by a conditional expectation we exploit in a regression problem using an ${L}_2$ loss term. The discrete denoiser structure is also exploit with a cross-entropy loss term. In contrast, SEDD only derives our entropic regularization loss term (see [eq. (10), 5] and our eq. (16)).
>
> [4] Campbell et al. 2022, Neurips
>
> [5] Lou et al. 2024, ICML
>
> > the paper lacks insufficient hyperparameters testing...
>
> We first underline the symmetric choice of $T_f$ and $\lambda$, because of time reparameterization. Consider a time-dependent rate $\lambda_t$, with $Q_t = \lambda_t Q$, then $P_t^{\lambda} = \exp(Q\int_0^t \lambda_s  ds) = P_{\int_0^t \lambda_s ds}$. In practice, we fix $\lambda = 1$ and vary $T_f$, leaving more sophisticated noise schedules to future work.
>
> We did an initial grid search for $\lambda, T_f$, validating $T_f = 3$. We thought the results in Figure 1 were convincing. As suggested, we will include an expanded search on MNIST together with various plots in the camera-ready version, like convergence speed to uniform distribution (e.g., on MNIST, we can see the sliced Wasserstein distance between $p_{\text{uniform}}$ and $p_t$ plateau after $t \approx 2.5$).
>
> > The FID metric...
>
> The reviewer makes a fair point, but we note that FID has been successfully used for MNIST (e.g., [6]) and is reliable in our experiments. In addition to FID we report our $F_1^{dc}$ metric, see Appendix D.3. It relies on local geometry to evaluate generative quality/diversity. We remain open to other metrics as recommended, we are confident we will still observe improvements.
>
> [6] Xu et al. 2023, Normalizing flow neural networks by JKO scheme, Neurips
>
> > Beneficial to compare with [...] SEDD uniform
>
> We train SEDD Uniform with the same settings as the other models. In response to concerns about FID, we report the $F_1^{dc}$ metric. Again, we see clear improvements:
>
> **Table 1: $F_1^{\text{dc}}\uparrow$ for different methods**
>
> |steps|10|25|50|100|200|500|
> |-|-|-|-|-|-|-|
> |flip|0.64|0.92|0.93|0.93|0.93|0.92|
> |denoise|0.13|0.67|0.87|0.96|1.00|1.00|
> |SEDD|0.61|0.82|0.88|0.92|0.92|0.92|
>
> We hope that the previous discussion has satisfied the reviewer. If this is the case, we kindly ask them to consider raising their score, if they feel it is appropriate.

---

> > ### Comment · Reviewer_9yNo · 2025-04-04
> >
> > I thank the authors for taking the time to address my concerns, and I am mostly satisfied with the answers. It would be nice to see these changes reflected in the revised version.
> > The authors highlight the theoretical contributions—particularly those on algorithm convergence—as the core value of the paper. As my expertise is more aligned with the practical aspects, I may not be best positioned to fully assess the depth of the theoretical claims. Concerns about the notation and text clarity persist.
> >
> > Some of my practical concerns also remain only partially addressed. In particular, it would be very helpful to see an experiment on real-life data to better demonstrate the applicability of DMPM—for instance, on one of the proposed datasets such as link repair [2]. While I appreciate the point that all datasets can be binarized, the practical usefulness of such binarization in the context of the discrete diffusion models remains uncertain.
> >
> > I have updated my score.

---

> > > ### Author Response · Authors · 2025-04-05
> > >
> > > We thank the reviewer for reconsidering the scope of our paper.
> > >
> > > We also value their feedback and are committed to revising the notation and text to make the paper as accessible as possible, without compromising the rigor required for the theoretical developments.
> > >
> > > Regarding the practical concerns raised, as mentioned in our initial rebuttal, we will acknowledge that additional numerical experiments are needed to draw definitive practical conclusions about our method.

---

### Official Review · Reviewer_ymZ8 · 2025-03-11

**Overall Recommendation:** 4

**Summary:**

This paper introduces Discrete Markov Probabilistic Models (DMPMs), a novel framework for discrete data generation based on continuous-time Markov chains (CTMCs). The forward noising process follows a Poissonian clock that flips bits randomly, while the reverse process reconstructs the data via an estimated discrete score function.

**Claims And Evidence:**

Yes.

**Essential References Not Discussed:**

No.

**Experimental Designs Or Analyses:**

The design of experiments is well-structured, with appropriate comparisons to MD4 and DFM.

**Methods And Evaluation Criteria:**

Yes.

**Other Comments Or Suggestions:**

N/A.

**Other Strengths And Weaknesses:**

Strengths: The paper provides strong theoretical guarantees for convergence and avoids density ratio estimation, improving numerical stability.

Weaknesses: Some assumptions might be too restrictive, see questions below.

**Questions For Authors:**

1. The state space is {0,1}$^d$, and so I believe the proposed algorithm will have wide applications in physics. Then I wonder if the DMPM can be generalized to systems where each particle has more than two potential states. Can you modify the Poission clock to design a proper forward process? And can the backward process and convergence result be derived with similar approaches in this paper?

2. Assumption 2.2 seems a bit too restrictive. For example, in a quantum system, two fermions cannot occupy the same state and so for such a system, $\mu^*((0, 0)) = 0$. Is this assumption technical or essential to the proof of the convergence?

**Relation To Broader Scientific Literature:**

This work builds on continuous-time discrete diffusion models and avoids unstable density ratio estimation.

**Theoretical Claims:**

I briefly looked at the proofs, and they appear correct based on my reading, but I did not dig into all the details.

---

> ### Author Rebuttal · Authors · 2025-03-31
>
> We thank the reviewer for their very positive feedback, and for finding our paper an interesting contribution to discrete diffusion models from both theoretical and methodological perspectives.
>
> > I wonder if the DMPM can be generalized to systems where each particle has more than two potential states. Can you modify the Poission clock to design a proper forward process? And can the backward process and convergence result be derived with similar approaches in this paper?
>
>
> We thank the reviewer for the opportunity to address this question.
>
> First, it is entirely possible to stay within the hypercube setting for any discrete data by using binary encoding. For instance, colored images can be encoded in 8-bits RGB pixels. Then each image is encoded in {$0, 1$}$^{3 \times 8 \times 64 \times 64}$, if using images of size $64 \times 64$, for instance. The same approach can be adopted for particles with more than two possible states.
>
> Second, we believe it should be possible to extend our study to state-spaces like {$0, \cdots, N$}$^d$, defining the uniform random walk with unit increment as the noising process, and that the backward process and convergence results would be derived very similarly.
>
> However, we emphasize that our paper's philosophy is to stay within the binary hypercube, and benefit from various optimizations enabled by such a setting, both theoretically and methodologically. For instance, this yields our regression objective, with the ${L}_2$ loss term (eq. 15), and our discrete denoiser structure, leading to the cross-entropy loss term (eq. 17).
>
> > Assumption 2.2 seems a bit too restrictive [...]. Is this assumption technical or essential to the proof of the convergence?
>
> We thank the reviewer for the opportunity to clarify this point. We wish to reassure them as we relax Assumption 2.2 soon after introducing it.
>
> Initially, it is used in Theorem 2.3 to derive our simplest convergence bound, which involves computing a Fisher divergence. However, we subsequently employ an early-stopping argument (as detailed from Theorem 2.5 onward) to circumvent its necessity in practice.
>
> This mirrors what happens in continuous diffusion models, where the data may lie on a lower-dimensional manifold without admitting a density against the Lebesgue measure in the full ambient space. There, too, the same line of arguments involving early stopping are used.
>
> In practice, the learned model can still assign zero probability to impossible states. For example, on MNIST, we observe that the learned model faithfully recreates handwritten digits, and it is safe to assume that it assigns negligible or zero probability to meaningless states (such as a completely white image). Hence, the assumption does not pose a practical limitation either.
>
>
> We hope we have successfully addressed the reviewer's questions, and we would like to thank them again for their positive feedback.

---

### Official Review · Reviewer_fUbX · 2025-03-13

**Overall Recommendation:** 3

**Summary:**

Score-based diffusion models are becoming one of the most promising non-adversarial and easy-to-implement data distribution reconstruction techniques. The idea is to define a forward noise process that gradually degrades the training data until it is transformed into a simple distribution that is easy to sample from. The model learns to reverse this process by learning the log-gradient of the noisy marginal distribution, known as the score. While most noise removal models to date have dealt with continuous state spaces, in recent years, the benefits of noise removal frameworks have been highlighted for problems involving discrete data, such as text, segmentation maps, categorical features, discrete latent spaces, and direct 8-bit image representations, and research in this area has been developing very actively.

This paper presents theoretical and practical investigations of the case where the state transition probability follows the mechanism of Poisson clocks with state bits for a {0,1}^d state space. The model set up by the authors succeeds in obtaining a closed-form expression using score-based conditional probabilities of forward processes, whereas many diffusion models originally suffer from the difficulty of handling the representation of backward processes. As a result, the authors have succeeded in theoretically guaranteeing that the proposed discrete diffusion model has a tighter fitting bound than conventional models. Along with their theoretical results, the authors have also quantitatively evaluated the effectiveness of their proposed method using artificial data and MNIST binary images to demonstrate its practical performance.

**Claims And Evidence:**

This paper argues that imposing the condition that the state transition probability follows the mechanism of Poisson clocks on the state bits on the state space of the hypercube {0,1}^d for the discrete diffusion model can resolve the issues that the conventional discrete diffusion model has had (limitations on convergence, increased computational cost for high dimensions, and the need for extremely strong assumptions in theoretical analysis).
Section 1 describes the discrete diffusion process assumed by the authors. Section 2 evaluates the usefulness of this model theoretically in the form of a bound on the convergence. Section 3 empirically and experimentally verifies this.

**Essential References Not Discussed:**

This paper refers to a wealth of literature on the discrete diffusion process, which has been the subject of particularly active research in recent years.
(I checked the list of references, for example, which cites several important papers [Campbell+, NeurIPS2022], and confirmed that this paper cites them appropriately.)

**Experimental Designs Or Analyses:**

The authors compare the proposed method with the latest discrete diffusion process for artificial data (Bernoulli simulation) and MNIST (the original 28x28 is upsampled to 32x32). While the Hellinger distance and KL divergence between distributions are generally used as quantitative comparison criteria, this paper uses the sliced wasserstein distance for evaluation because it has a theory that can handle high-dimensional data that previous studies could not handle.

I think this paper has produced some impressive results in terms of theoretical contributions. However, the theoretical results in Section 2 only give a bound on the closeness between the data and the model distribution, so it would certainly be very interesting from a practitioner's perspective to see how it behaves on practical data. I think the authors' experimental results (albeit not on a large and diverse set of settings) give the reader some intuitive insight into the effectiveness of the proposed method.

**Methods And Evaluation Criteria:**

My comment on this term is merged into Claims part.

**Other Comments Or Suggestions:**

Line 581: ’’masked’’ -> ``masked’’ would be better.

I do not know why it is necessary to have \propto_{I} in equation 10. Is this a typo? I can certainly see why \propto is needed in equation 9.


The current draft of this paper seems to be a rather esoteric explanation aimed at experts in discrete diffusion processes or those who are very familiar with the latest developments in them. As a result, newcomers to the topic or general readers interested in the surrounding fields may not find it easy to properly understand the true value of this paper. In order to better convey the value of this paper to a more diverse audience, we would like to make the following recommendations.

> addressing the issues raised in prior work. (Page 1, right column)
> However, many of these methods still lack rigorous error bounds or scale poorly in high dimensions. (Page 7, left column)
> However, their results rely on strong assumptions in contrast to our results. Besides, our bounds are simpler and better, in particular with respect to the time horizon. (Appendix A)

The high-level sketched commentary to these existing studies is no doubt very intuitive and meaningful. However, the current draft does not specify what model assumptions individual previous studies use and what theoretical results they have, and readers will have to individually check the original papers themselves. To be honest, I cannot write down in a positive way what improvements this paper has from some of the previous studies. Therefore, my request is that the authors compile the “model assumptions” and “theoretical results (error bounds)” for each literature in the form of a table so that we can see at a glance the value of this paper when compared to existing studies. In fact, the authors' investigation of the previous studies is so deep and comprehensive that it would be of great value to summarize the results of the previous studies as a unified and improved paper on them.

**Other Strengths And Weaknesses:**

[Strengths]

- This paper insightfully discovers a special class of discrete diffusion processes in which the backward process can be expressed in a closed-form, easy-to-handle form.

- The authors theoretically guarantee the importance of this discovery in the form of a bound on the closeness of the data distribution and sample distribution. This is a very impressive result that scales more naturally to higher dimensions than conventional discrete diffusion processes.

- The authors demonstrate the practical effectiveness of the proposed method not only with theoretical guarantees, but also with medium-sized data.

[Weaknesses]

- My main concern is that the class of discrete diffusion processes considered in this paper (discrete states on the hypercube, with state transition probabilities being the direct product of each dimension) is a much stronger situation than the assumptions made in previous papers. My main concern is that the class of discrete diffusion processes considered in this paper (discrete states on the hypercube, with state transition probabilities being the Cartesian product of each dimension) is a much stronger situation than the assumptions made in previous papers. I think that setting very strong assumptions in order to achieve a breakthrough in theoretical analysis is very meaningful in itself. However, because this paper does not discuss this in depth, it is somewhat unclear where it stands in relation to recent developments in discrete diffusion processes. In particular, the description of the paper's position in relation to the latest methods (Section 0 and Appendix A) is limited to high-level, abstract explanations, so a very deep prior knowledge of the field is required to correctly understand the paper's true value.

**Questions For Authors:**

I have no additional questions.

**Relation To Broader Scientific Literature:**

With the spectacular success of continuous state diffusion processes, discrete state diffusion processes are expected to play an increasingly important role in a wide range of applications, including text and biological information. I think this paper provides some very important insights into this development.

On the other hand, this paper seems to make the strong assumption that the state space of the discrete diffusion model is a hypercube on {0,1}^d, so it seems non-trivial whether this approach can be applied straightforwardly to natural language or genetic information.

**Theoretical Claims:**

I find the theoretical contribution of this paper very impressive. I would like to ask the authors a question to properly understand its value. I am trying to understand this paper by contrasting it with the theoretical work of literature [Campbell+, NeurIPS2022] and [Ren+, ICLR2025] (which seems to be same as the reference arXiv2410.03601 cited in this paper) as important examples. For eacmple, Reference [Ren+, ICLR2025] seems to consider state transition matrices in the general discrete state space as state transitions in discrete diffusion processes (Section 2 and Appendix B). As a result, since the state transition matrix of a Markov chain is combinatorial in dimension and large in size, it seems that, for example, in its mixing time analysis, the spectral gap (the order of the inverse of the mixing time) is evaluated as in the standard MCMC procedure for mixing time analysis. On the other hand, this paper, very interestingly, seems to achieve avoiding any mixing time analysis focusing on the eigenvalues of this kind of state transition matrix. In fact, the proof in Appendix F.4 shows a very concise argument for state transitions, where only a reference to self-transitions for an arbitrary time interval\eta is sufficient. So, my question is, does the fact that we have made a stronger assumption on the state transition matrix than before contribute to this concise argument? As I understand it, the assumption seems to be placed that the state transition probabilities for high-dimensional binary data are represented by the direct product of independent binary state transitions in each dimension, as shown in Equation 9. Is this understanding (i.e., that it restricts the state transition dynamics more than previous studies) correct?

---

> ### Author Rebuttal · Authors · 2025-03-31
>
> We thank the reviewer for their comments, which will help improve the presentation of our contributions. We hope that we have addressed all of their questions in the responses below. Given their positive feedback—particularly their appreciation of the "very impressive" theoretical contributions—we would kindly ask the reviewer to consider raising their score, if they feel it is appropriate.
>
> > I wanted to know whether the breakthrough [...] briefly outline Appendix F (Proof of Theoretical Results)
>
> We thank the reviewer for their remark, which will help us clarify the key contributions of our paper. To this end, we provide a brief sketch of the proof for **Theorem 2.3**, which we will expand upon and include in the revised version of the paper.
>
> The first step in the proof is to decompose the relative entropy
> between the backward process and our generative algorithm into three terms:
> $$
> \mathrm{KL}(\overleftarrow{P} \| P^*) \leq \mathrm{KL}(\mu_{T_f} \| \gamma^d) + II + III.
> $$
>
> - **The first term** on the right-hand side arises from the fact that the generative process does not start from the marginal at time $ T_f $, but rather from the stationary distribution $ \gamma^d $. Using standard arguments for the convergence of random walks on $ \lbrace 0,1\rbrace^d $, this term is exponentially small in $ T_f $.
>
> - **The second term** $ II $ corresponds to the score approximation error. By **Assumption 2.1**, we can bound it by $ T_f \varepsilon $.
>
> - **The third term**, $ III $, corresponds to the time discretization error of the generative process. It is given by
> $$
> \sum_k \mathbb{E} [ \int_{[t_k, t_{k+1}]} \sum_{\ell} u^*_{t_k}(\ell) \cdot h(u^*_{t}(\ell)/u^*_{t_k}(\ell)) dt ],
> $$
> where $u_t^*(\ell) = 1 - s_t^{\ell}(\overleftarrow{X}_{t})$, and $(\overleftarrow{X}_t)_t$ denotes the time-reversed process.
>
> To bound this term, we rely critically on **Proposition F.11**, which serves as the main technical tool in our analysis. Specifically, the proposition shows that:
>
> 1. The process $ (\sum_{\ell} u^*_t( \ell) )_t$ is a **martingale**, and
> 2. The process $ (\sum_{\ell} h(u^*_t(\ell)))_t $ is a **submartingale**.
>
> Thanks to these two properties, along with **Assumption 2.2** and an application of Jensen’s inequality, we are able to bound the third term $ III $.
>
>
> > does the fact that we have made a stronger assumption on the state transition matrix than before contribute to this concise argument?
>
> The reviewer is correct in noting that the arguments we developed—and summarized in our previous response—crucially rely on using the random walk on {$0,1$}$^d$ as the noising process. Nevertheless, we view our work as a foundational step toward extending the framework to broader classes of discrete generative models.
>
> In fact, as part of our future work, we aim to formalize our reasoning in a more general setting that could encompass a wide family of discrete models, including masked diffusion processes.
>
>  > My main concern [...] is a much stronger situation than the assumptions made in previous papers.[...]
>
>  We thank the reviewer for this comment, which helps to clarify the implications of our work.
>
> First, as this appears to be the reviewer’s most critical concern, we acknowledge that—unlike most previous works—both the state space and the noising process we consider are fixed. These choices are indeed crucial for establishing our theoretical results (as detailed further above) under minimal assumptions on the data distribution. We will emphasize on this point in the revised version of the paper.
>
> However, we would like to make two points in response. First, although working on the hypercube may initially appear restrictive, we note that any discrete space can, in principle, be embedded into the hypercube via binary encoding. This means that our algorithm can, in theory, be applied to any discrete data structures.
>
> Second, we view the theoretical analysis presented in this paper as a first important step to demonstrate that convergence guarantees for a general class of discrete score-based generative models can be achieved under minimal assumptions on the data distribution.
> As we emphasized in our first response, we are confident that the techniques developed here have the potential to extend to other discrete structures as well. However, since the exact scope of this generalization is not yet fully understood, we refrain from making definitive claims at this stage.
>
> >  In particular, the description of the paper's position in relation to the latest methods [...] is limited to high-level, abstract explanations.
>
> We will follow the reviewer’s suggestion and, in particular, include a comparison table in the camera-ready version of our paper. We also plan to add a detailed discussion of the assumptions made in previous works—specifically those concerning the data distribution, as well as the density and score of the noising process—which our approach is able to relax.

---

### Official Review · Reviewer_5ek8 · 2025-03-16

**Overall Recommendation:** 3

**Summary:**

This paper introduces the Discrete Markov Probabilistic Model (DMPM), a uniform noising/denoising algorithm for discrete data generation. The authors establish theoretical convergence bounds under minimal assumptions and validate the effectiveness of their method empirically on both low-dimensional Bernoulli datasets and a relatively high-dimensional binary MNIST data. By bridging rigorous theoretical foundations with practical performance, this work makes a valuable contribution in the emerging area of discrete generative modeling.

### update after rebuttal
The reviewer thanks the authors for the detailed response and for positively answering the raised concerns. Overall, the reviewer thinks that the theoretical results in this paper adds value to the better understanding of discrete diffusion models. Although experiments are not the main focus of this paper, the submitted version had a wrong experiment as acknowledged by the authors at the end of the rebuttal below. Besides, the paper makes a claim against the standard practice in large-scale experiments, which is "absorbing kernels perform better than uniform". However, there is not enough evidence to prove or disprove this claim. Given the strength of the paper in theoretical analysis, the reviewer recommends accepting the paper.

**Claims And Evidence:**

The paper provides strong theoretical results supported by rigorous proofs; however, the empirical evidence presented appears somewhat limited. Specifically, the claim that uniform noising kernels offer provable theoretical guarantees and superior performance compared to masked approaches like MD4 remains inconclusive. The main concern is that uniform noising kernels typically perform poorly in high-dimensional scenarios compared to absorbing kernels. Given this context, additional large-scale experiments or high-dimensional evaluations are necessary to convincingly support the claim that uniform noising kernels are practically superior to masked methods like MD4 or MDLM.

**Essential References Not Discussed:**

Some essential references have not been included. Please see the weaknesses for details.

**Experimental Designs Or Analyses:**

Yes, the conducted experiments look okay to me. But large-scale experiments are missing for a thorough evaluation.

**Methods And Evaluation Criteria:**

The evaluation is currently done at a relatively smaller scale (toy dataset and MNIST). Large-scale evaluations are essential for conclusive evaluation of the proposed claims.

**Other Comments Or Suggestions:**

Please see weaknesses.

**Other Strengths And Weaknesses:**

### Strengths
The choice of transition kernel is quite ambiguous because there exists a mixed set of papers that claim absorbing kernels are better than uniform noising kernels or vice-versa. Indeed, a proper theoretical understanding of any of these two commonly used kernels or new approaches is a valuable contribution to the community.

### Weaknesses

1. Line 25 (right col): “In this paper, we introduce Discrete Markov Probabilistic Models (DMPMs), a new class of generative models for discrete data that bridges these gaps.” This is not entirely new and dates back to Sohl-Dickstein et al. 2025 and popularized by Austin et al. 2021 with many other recent follow-up works.

2. The notation in Line 66 for $|d\mu/dR-d\nu/R|dR$ seems squeezed in one line. It could be better presented as $|\frac{d\mu}{dR} - \frac{d\nu}{dR}|dR$.

3. Lines 99-103: “We derive closed-form expressions for the backward transition rates, which involve conditional expectations over the forward process. This enables, for the first time, an efficient training procedure based on regression in the discrete setting.” What is new here? What is done for the first time? The notion of conditional expectation of $X^{rev}_T$ given $X^{rev}_0$ is the key contribution in prior works, such as D3PM (Austin et al. 2021) or MDLM (Sahoo et al. 2024).

3. The transition probabilities derived in Equation (2) and Appendix B.1 can be seen in the introduction to CTMC (Chapter 2, Liggett). Again, the originality and innovation of the analysis is limited in my opinion.The authors are encouraged to highlight (maybe informally) the original theoretical contributions early on in the paper for the ease of assessing the major contributions.

4. The key contribution of the paper is in Equation (6) with a more generalized version in Proposition 1.1, where the discrete equivalent of score is written as an affine transformation of $X^{fwd}_0$ and $X^{fwd}_t$. A slightly modified version of this result has previously appeared in DFM (Campbell et al. 2024).

5. The regression loss used in Equation (15) is quite similar to the DFM loss or other discrete diffusion losses as in MDLM. The reviewer is finding it difficult to understand what this theoretical result is prescribing here. Again, the loss in Line 225 is similar to the subs-parameterization introduced in MDLM (Sahoo et al. 2024). The authors are encouraged to clarify these similarities and highlight the innovations of this paper.

6. What is the relation between Assumption 2.2 and the irreducibility property of CTMC? It’d be better to connect this assumption with CTMC for better understanding and consistency with prior discussion.

7. Theorem 2 (Chen et al. https://openreview.net/pdf?id=zyLVMgsZ0U_) has a similar form as that of Theorem 2.3 (also Theorem 2.5) in this paper. While the first term in the upper bound seems unavoidable, the second term can be avoided as shown in Theorem 2.5. The reviewer is wondering about the dependence on $\epsilon$ and $T_f$ in the third term, which seems like a coarse discretization error as compared to Chen et al. Is this the optimal characterization of the score approximation in the full trajectory? Intuitively, if the approximation error is of $O(\epsilon)$ and then we take $T_f$ denoising steps, then the total accumulated error would be of $O(\epsilon T_f)$. Isn’t there a tight characterization of this error?

8. The authors are encouraged to write some implications of Corollary 2.7 at the end of Section 2. The transition from Section 2 to Section 3 seems abrupt.

9. Table 1: While other methods show a consistent behavior, DMPM (flip) seems inconsistent as the number of reverse steps increases. The authors are encouraged to check the numerical simulations of this sampler or provide a better justification for this inconsistent behavior.

10. Experiments on MNIST are not quite high-dimensional. The paper would significantly benefit from even higher dimensional modeling, especially because uniform noising is known to perform suboptimally compared to absorbing diffusion in higher dimension.

11. How would the analysis change for absorbing kernels since that has been very successful in large-scale experiments (see MDLM and other follow up works).

**Questions For Authors:**

Please see weaknesses.

**Relation To Broader Scientific Literature:**

The choice of transition kernel is quite ambiguous because there exists a mixed set of papers that claim absorbing kernels are better than uniform noising kernels or vice-versa. Indeed, a proper theoretical understanding of any of these two commonly used kernels or new approaches is a valuable contribution to the community.

**Theoretical Claims:**

Theorem 2 (Chen et al. https://openreview.net/pdf?id=zyLVMgsZ0U_) has a similar form as that of Theorem 2.3 (also Theorem 2.5) in this paper. While the first term in the upper bound seems unavoidable, the second term can be avoided as shown in Theorem 2.5. The reviewer is wondering about the dependence on $\epsilon$ and $T_f$ in the third term, which seems like a coarse discretization error as compared to Chen et al. Is this the optimal characterization of the score approximation in the full trajectory? Intuitively, if the approximation error is of $O(\epsilon)$ and then we take $T_f$ denoising steps, then the total accumulated error would be of $O(\epsilon T_f)$. Isn’t there a tight characterization of this error?

---

> ### Author Rebuttal · Authors · 2025-03-31
>
> We thank the reviewer for their positive feedback, especially regarding the strength of our theoretical contributions. Due to space constraints, we address only the most critical concerns, but we welcome further discussion if any important points remain unaddressed.
>
> > Additional large-scale experiments or high-dimensional evaluations are necessary [...]
>
> We acknowledge that our experiments are limited, and we agree that more extensive large-scale and high-dimensional evaluations are necessary to fully support the use of uniform noising kernels in comparison to masked ones. Nevertheless, as the reviewer noted, the core contributions of our paper are theoretical.
>
> Our main objective is to derive convergence guarantees for discrete generative models that match those known for continuous diffusion models. As emphasized in our paper, there is a notable lack of results concerning discrete diffusion models. The closest related work, upon which ours significantly improves (as detailed in our response to Rev. fUbX), has only recently been accepted to ICLR [1].
>
> In pursuing our goal, we obtained a representation of the score as a conditional expectation (Prop 1.1 and 1.2), which is the foundation of our methodology.
>
> While limited, we believe our numerical experiments add value by showing the practical potential of our approach. That said, we agree that strong conclusions should not be drawn from these preliminary results, especially in large-scale or high-dimensional scenarios. To address this, we will revise the conclusion to clarify that our findings are meant to motivate future empirical work rather than provide definitive evidence.
>
> [1] Ren et al, ICLR 2025
>
>  > Lines 99-103 [...] The notion of conditional expectation of [...]   is the key contribution in prior works, such as D3PM  or MDLM).
>
> From our understanding, prior works—including those mentioned by the reviewer— aim to estimate the generator associated with the backward dynamics of the backward process  relying on estimating the conditional distributions  $p_{0|t}$ of $X_0$ given $X_t$ for $t \in [0,T_f]$.
>
> In contrast, we first demonstrate that estimating the generator of the time-reversed process is equivalent to estimating the discrete score function introduced. Second, we show that (1) this score function can be expressed as a conditional expectation, and (2) it is associated with a discrete denoising structure.
>
> Therefore estimating the backward generator reduces to solving either a regression or a classification problem—tasks that are often more tractable than learning full conditional distributions. While prior work seeks to model an entire conditional distribution, we focus on estimate a function that solves a regression or classification task.
>
> We regret any confusion caused by the original formulation and will revise the paper to include this discussion in order to clarify our claims.
>
> > Theorem 2 in [2] [...] Isn’t there a tight characterization of this error?
>
> The third term $\epsilon T_f$ has the same meaning and significance of the corresponding term in Theorem 2 of [2], appearing as such. Indeed, it accounts for the score approximation error that we do not investigate in our submission similarly to [2].
>
> [2] Chen et al, ICLR 22
>
>  > The transition probabilities [...] can be seen in [...] (Chapter 2, Liggett)...
>
> As stated in the paper, these computations are well known, and we thank the referee for pointing out the reference in Liggett’s book, which we will include in the revised version. As the referee suggests, we will emphasize this point more clearly early in the paper.
>
> > The regression loss used in Equation (15) is quite similar to the DFM loss...
>
> First, we would like to emphasize the following points based on our understanding:
> - (1) The SUBS parameterization is applicable only to absorbing kernels;
> - (2) The approach in [3] focuses on estimating conditional probabilities $p_{s|t}$, which is fundamentally different from our methodology, as previously discussed.
>
> Concerning the DFM loss, we were not able to find a similar result as ours in [4]. It seems to us they use a flow matching construction with a conditional rate matrix built ad-hoc (Section 3.2). This seems fundamentally different from the way we derive our reverse process (see (4)).
>
> [3] Sahoo S. et al, 2024, Neurips
> [4] Campbell et al 2022, Neurips
>
>  > What is the relation between Assumption 2.2 and the irreducibility property of CTMC?
>
> Ass. 2.2 only states that $\mu^\star$ has full support and does not impose any conditions on the CTMC considered in our analysis.
>
> > Table 1: DMPM (flip) seems inconsistent
>
> We thank the reviewer for spotting this, indeed we loaded the wrong model checkpoint for the evaluation run at 500 steps, here is the result of a new run:
>
> **Table 1: FID$\downarrow$ and F$_1^{dc}\uparrow$ for DMPM methods at 500 reverse steps.**
>
> | |FID|F$_1^{dc}$|
> |---|---|---|
> |flip|6.98|0.94|
> |denoise|2.52|1.00|

---

> > ### Comment · Reviewer_5ek8 · 2025-04-04
> >
> > Thank you for the rebuttal.
> >
> > Some of my concerns have been addressed. I will update my review in light of this response as necessary. Regarding the characterization of approximation error, $O(\epsilon T_f)$ seems trivial. A tight characterization would strengthen the contribution of the paper.
> >
> > **As stated in the paper, these computations are well known, and we thank the referee for pointing out the reference in Liggett’s book, which we will include in the revised version. As the referee suggests, we will emphasize this point more clearly early in the paper.**
> > Perhaps I missed, could the authors point out where it is stated in the paper?
> >
> > Missing response to previous Q12. Would the current analysis break for absorbing kernel?

---

> > > ### Author Response · Authors · 2025-04-07
> > >
> > > We are grateful to the reviewer for their continued engagement and thoughtful input.
> > >
> > > > A tight characterization would strengthen the contribution of the paper.
> > >
> > >  While we acknowledge that developing sample complexity bounds is undoubtedly an important and valuable direction for future research, we believe, as initially stated, that it falls outside the scope of the present paper.
> > >
> > > It is worth noting that in the context of continuous diffusion models, the literature typically begins by establishing bounds on the discretization error (e.g., [1,2,3]), before later works (e.g., [4,5,6]) build on those results to derive sample complexity bounds. While incorporating such bounds would certainly strengthen our paper, doing so would require substantial additional theoretical developments, which would significantly increase the length of the manuscript.
> > >
> > > As it stands, our paper already totals 47 pages including the supplementary material. Based on analogous analyses for continuous diffusion models, we estimate that including a thorough sample complexity analysis would add at least 30 pages, bringing the total to around 80 pages, which can be considered excessive.
> > >
> > > [1] S. Chen, S. Chewi, J. Li, Y. Li, A. Salim, and A. R. Zhang. Sampling is as easy as learning the score: theory for diffusion models with minimal data assumptions, 2023. arXiv: 2209.11215 \
> > > [2] Hongrui Chen, Holden Lee, and Jianfeng Lu. Improved analysis of score-based generative modeling:
> > > User-friendly bounds under minimal smoothness assumptions. In International Conference on Machine
> > > Learning, pages 4735–4763. PMLR, 2023. \
> > > [3] Holden Lee, Jianfeng Lu, and Yixin Tan. Convergence of score-based generative modeling for general data distributions. In International Conference on Algorithmic Learning Theory, pages 946–985. PMLR, 2023. \
> > > [4] Oko, K., Akiyama, S., & Suzuki, T. (2023, July). Diffusion models are minimax optimal distribution estimators. In International Conference on Machine Learning (pp. 26517-26582). PMLR. \
> > > [5] Azangulov, Iskander, George Deligiannidis, and Judith Rousseau. "Convergence of diffusion models under the manifold hypothesis in high-dimensions." arXiv preprint arXiv:2409.18804 (2024). \
> > > [6] Yakovlev, Konstantin, and Nikita Puchkin. "Generalization error bound for denoising score matching under relaxed manifold assumption." arXiv preprint arXiv:2502.13662 (2025).\
> > > > Perhaps I missed, could the authors point out where it is stated in the paper?
> > >
> > > Line 70-71 second column, it is written "The transition probability matrix [...] is **known** to be"
> > >
> > > As mentioned previously, we will emphasize this point more clearly in the paper.
> > >
> > > > Missing response to previous Q12. Would the current analysis break for absorbing kernel?
> > >
> > >
> > > We view the theoretical analysis carried out in this paper as a proof of concept for the fact that guarantees of convergence for discrete score based generative models can be obtained with the same degree of precision as for continuous diffusion models. Building on the results of this work, we are  confident that the techniques developed here can lead to strong guarantees of convergence for other noising kernels. In particular most of the theoretical analysis carried out here would work for  absorbing kernels as well, with the important exception of the score monotonicity estimates of Appendix F. Our current research efforts go in the direction of developing a general convergence theory for discrete diffusion models which encompasses absorbing kernels as a special case. At the moment of writing, we do not fully foresee the range of applicability of the ideas explored in this paper, and thus abstain from making further claims.

---

### Decision · Program_Chairs · 2025-05-01

**Decision:**

Accept (poster)

**Comment:**

**Summary.**

The topic of this work is discrete space generative modeling. Specifically, the authors introduce the new class of generative models, the so-called Discrete Markov Probabilistic Models (henceforth DMPMs).

They construct the forward and derive the backward processes (continuous time MCs on the hypercube), which is amenable to efficient training (overcoming a limitation of previous methods).

They obtain theoretical convergence guarantees for DMPMs.

They demonstrate the performance of their approach by experiments (MNIST) against state-of-the-art discrete generative models.

**Strengths.**

* The proposed framework of DMPMs is new.
* Theoretical error guarantees.

**Weaknesses.**

* Performance compared to masked approaches (e.g. MD4, MDLM) is not convincing (5ek8).
* The scale of the experiments (MNIST) is not large enough (5ek8, 9yNo).
* The technical novelty was deemed limited by some (5ek8, 9yNo) and the difference between elements taken from prior work and contributions should be better emphasized (refer to the comments of 5ek8).
* There should be a clearer comparison of the assumptions made in this submission and in prior work (fUbX).
* The clarity of the manuscript could be improved (9yNo).

**Discussion and reviewer consensus.**

There was originally a disparity in the reviewers’ opinion but after the author response, the reviewers were unanimously in agreement that the paper could be (weakly) accepted.

**Additional remarks.**

There was a concern that the state space of the proposed discrete diffusion model is restricted to the hypercube (fUbX, ymZ8, 9yNo). The authors partially address this concern by pointing out that arbitrary discrete data can be handled through binary encoding.

**Overall evaluation.**

I recommend this paper for acceptance if there is room in the program.